# Transient ligand contacts of the intrinsically disordered N-terminus of neuropeptide Y$_2$ receptor regulate arrestin-3 recruitment

Anette Kaiser [1,2] ✉, Juan C. Rojas Echeverri[3,4], Asat Baischew [3,4], Maik Pankonin [5], Karl D. Leitner [1], Claudio Iacobucci[3,4,6], Davide Sala[7], Christian Ihling[3,4], Ronny Müller[2], Rok Ferenc[5], Annette G. Beck-Sickinger [2], Peter Schmidt[5], Jens Meiler [7], Peter W. Hildebrand [5] & Andrea Sinz [3,4] ✉

Previous efforts in delineating molecular mechanisms of G protein-coupled receptor (GPCR) activation have focused on transmembrane regions and ligand-receptor contacts of the extracellular loops. The role of the highly flexible N-termini of rhodopsin-like GPCRs have not been well characterized to date. We hypothesize that transient contacts between the peptide ligand and the intrinsically disordered N-terminus (NT) of the neuropeptide Y (NPY) receptor Y$_2$ (Y$_2$R) will affect receptor signaling. We employ cross-linking mass spectrometry to capture ligand-receptor contacts including transient binding modes. A photo-reactive NPY analogue allows mapping the interaction between NPY and Y$_2$R NT resulting in a total number of 40 cross-links. The cross-links provide distance constraints for deriving structural models of the interaction. Molecular dynamics simulations highlight the structural flexibility and rapid interconversion of ligand-receptor contacts. Mutagenesis of Y$_2$R and functional characterization suggest that the cross-linking hotspots in the NT electrostatically control its conformational ensemble. The NT engages in transient contacts to the peptide and prolongs ligand residence time, which is required for efficient interaction of Y$_2$R with arrestin-3, but not G$_i$. We delineate structure-function relationships for the intrinsically disordered Y$_2$R NT and propose a functional role for transient binding modes involving the NT of a peptide-binding receptor.

G protein-coupled receptors (GPCRs) are physiologically versatile molecular transducers of extracellular signals into the cell. Due to their role as drug targets, GPCRs are of outstanding clinical importance and approximately 30% of all currently available drugs target these receptors[1,2]. Potential ligands range from small-molecule neurotransmitters to large and flexible peptides and proteins. In the past two decades, much work has been devoted to the molecular mechanisms of GPCR activation. Most of the work has focused on elucidating molecular details of ligand binding to the extracellular loops[3–5] and on re-wiring of inter-transmembrane (TM) contacts that

[1]Department of Anesthesiology and Intensive Care, University of Leipzig Medical Center, Leipzig, Germany. [2]Institute of Biochemistry, Faculty of Life Sciences, Leipzig University, Leipzig, Germany. [3]Department of Pharmaceutical Chemistry and Bioanalytics, Institute of Pharmacy, Martin Luther University Halle-Wittenberg, Halle, Saale, Germany. [4]Center for Structural Mass Spectrometry, Martin Luther University Halle-Wittenberg, Halle, Saale, Germany. [5]Institute of Medical Physics and Biophysics, Medical Faculty, Leipzig University, Leipzig, Germany. [6]Department of Physical and Chemical Sciences, University of L'Aquila, Via Vetoio, L'Aquila, Italy. [7]Institute for Drug Discovery, Medical Faculty, Leipzig University, Leipzig, Germany. ✉e-mail: anette.kaiser@medizin.uni-leipzig.de; andrea.sinz@pharmazie.uni-halle.de

enable the interaction with different cellular transducer proteins[6–11]. The structural and functional modulation of signaling by the more variable GPCR N-termini (NT) has remained largely enigmatic, particularly for rhodopsin-like GPCRs. In the glutamate, secretin, and adhesion families, the NT are large and form well-structured domains[12]. In contrast, in the rhodopsin-like family, the NT are typically shorter and often not resolved in high-resolution structures, suggesting that they are highly dynamic. Most likely, the NT are intrinsically disordered regions (IDRs) that might be involved in interactions with different extracellular partners and adopt context-dependent conformations. While it is clear that NT can contain export signals for protein processing and sorting[13], proof of direct contributions to receptor signaling is scarce. Notable exceptions are a few receptors that carry tethered ligands in their NT, such as protease-activated receptors PAR1-PAR4[14] or the melanocortin receptor 4 (MC4R)[15]. A large-scale analysis of transcriptome data has recently shown that, also for many other GPCRs, N- and C-termini can affect signaling and drug responses, as evidenced by different signaling outcomes of GPCR isoforms[16]. For example, long and short isoforms of the rhodopsin-like GPR35 NT alter G-protein and arrestin-recruitment preferences[16,17]. This might be mediated by altering receptor expression, direct ligand contacts or allosteric modulation.

In particular, for rhodopsin-like GPCRs with larger peptide or small protein ligands, such as the neuropeptide Y (NPY) family, the potential interaction surface with the putatively disordered NT is large. The NPY family consists of three homologous peptide ligands (NPY, PYY, PP) and four rhodopsin-like receptor subtypes in humans[18,19]. Their numerous biological functions, among others, anorexigenic signals mediated by $Y_2R$ and $Y_4R$[20,21], make this family a promising pharmaceutical target[22–24] and have spurred functional and structural investigations. High-resolution structures by X-ray crystallography and cryo-electron microscopy (cryo-EM) of the antagonist-bound $Y_2R$[25] and NPY-bound $Y_2R$ (along with $Y_1R$ and $Y_4R$) in complex with $G_i$ proteins have been determined[26,27]. This has provided molecular insights into receptor-ligand selectivity and signal transduction to $G_i$. The structures show an extended interaction surface between NPY and the extracellular loop 2 (ECL2), in agreement with earlier mutational and NMR studies[28], but the low local resolution of the ECL2 and extracellular parts of the helix of NPY indicate a certain flexibility in this region. In addition, the N-terminus of the $Y_2R$ was not structurally resolved. We therefore hypothesize that transient ligand contacts will occur in this putative IDR.

Cross-linking mass spectrometry (XL-MS) allows characterizing IDRs and intrinsically disordered proteins (IDPs) by covalently linking reactive sites in proteins. XL-MS has been successfully applied to IDPs, such as the tumor suppressor p53[29], AUX/IAAs transcriptional repressors[30], and α-synuclein[31], identifying binding topologies with interacting proteins and characterizing conformational ensembles. XL-MS offers detailed molecular insights into IDPs/IDRs at low protein concentrations and is also applicable to membrane proteins[32] and GPCRs[33]. The frequently used photo-reactive diazirine moiety undergoes a cross-linking reaction on a µs timescale[34] upon activation by UV-A irradiation (365 nm). Diazirine cross-links form adducts with all 20 amino acids, with a preference for acidic amino acids[35,36], thereby providing distance constraints (10–15 Å) for subsequent computational modeling and molecular dynamics (MD) simulations.

Here, we characterize the interaction of the intrinsically disordered NT of $Y_2R$ with NPY employing XL-MS, receptor mutagenesis analysis, Rosetta modeling and MD simulations. Our data show the high structural flexibility and rapid interconversion of ligand-receptor contacts and provide evidence that transient contacts of $Y_2R$ NT to NPY modulate recruitment of arrestin-3 to the receptor by prolonging ligand residence time.

## Results

### XL-MS identifies direct interactions between NPY and $Y_2R$ NT

Recent cryo-EM structures of NPY bound to the $Y_2R$ lack a defined structure for the NT[26,27]. This suggests a high structural flexibility and the existence of an ensemble of conformations that might be linked to specific biological functions, such as peptide recognition or transducer coupling. This is consistent with bioinformatic predictions by IUPred2A[37], flDPnn[38] or ESpritz[39] that consistently show a high probability of intrinsic disorder in the $Y_2R$ NT (Supplementary Fig. 1). To capture potential NPY-$Y_2R$ interactions by XL-MS, we site-specifically labeled NPY with diazirine-containing photo-leucines, replacing leucine residues at positions 17, 24, and 30, and employed lipid-reconstituted $Y_2R$. To this end, a cysteine-deficient $Y_2R$, which only contained the two cysteine residues that form the disulfide bridge ($C^{123}$ and $C^{203}$), was recombinantly expressed in *E. coli* as inclusion bodies using high-density fed-batch fermentation[40]. Inclusion bodies were solubilized in SDS, folded in vitro, and reconstituted into DMPC/DHPC bicelles. Receptor functionality after refolding was confirmed using a fluorescence-based ligand binding assay (Supplementary Fig. 2). This $Y_2R$ in vitro system has provided important molecular insights[28,41,42], for example, unfolding of NPY's C-terminal helix upon receptor binding[28] even before the cryo-EM structure had become available. Upon irradiation of the diazirine in the peptide with UV-A light, linear diazo and carbene intermediates are formed, which can insert into X-H bonds in proximity (10–15 Å; Fig. 1a). The reaction occurs on the ns to µs timescale, enabling trapping short-lived binding states. We identified 40 peptide-receptor cross-links by liquid chromatography coupled to trapped ion mobility tandem mass spectrometry (LC-TIMS-MS/MS)[43], 34 of which correspond to the extracellular regions of $Y_2R$. These map 16 contact points to $Y_2R$ NT and three contacts to ECL2 (Fig. 1B and Supplementary Data 1). Raw data of annotated mass spectra are found in ProteomeXchange with identifier (PXD051865), and select MS/MS spectra of the cross-linked fragments are presented in Supplementary Fig. 3. As expected, cross-links are mainly located at the extracellular surface of the receptor. The cross-links between $L30^{NPY}$-$E210^{ECL2}$ and $L30^{NPY}$-$E211^{ECL2}$ are in good agreement with the cryo-EM structure[26] ($C_\beta$-$C_\beta$ distance 8.8–12.1 Å). $L24^{NPY}$ is in spatial proximity to E210 and $E211^{ECL2}$ in the cryo-EM structure with $C_\beta$-$C_\beta$ distances of 16.7 and 19.8 Å, assuming a high flexibility in these regions. In addition to these expected ECL2 cross-links, a large number of cross-links were mapped to the $Y_2R$ NT (Fig. 1b). We identified several clusters of proximity, in which the diazirine-substituted leucine analogues in NPY at positions L24 and L30 react with multiple sites in the $Y_2R$ NT. Most cross-links were found for the acidic residues in region $E15^{NT}$–$E20^{NT}$ and $D35^{NT}$–$E39^{NT}$. Furthermore, spatial proximity of $L24^{NPY}$ to $Y24^{NT}$ as well as $L30^{NPY}$ to $D7^{NT}$, $E9^{NT}$, $E10^{NT}$, and $Y22^{NT}$ were detected. To probe the specificity of the cross-links, we performed competition experiments using an excess of either the $Y_2R$ antagonist JNJ-31020028 (ref. 44) or unlabeled NPY. Both unlabeled ligands essentially abolished all cross-links and hence validate the specificity of the photoreactions. The corresponding mass spectra are also available in the ProteomeXchange accession.

To further confirm the cross-links between NPY and $Y_2R$ NT, we prepared an isolated peptide of the $Y_2R$ NT (aa 1–45) by solid-phase peptide synthesis and conducted an analogous set of XL-MS experiments with the isolated $Y_2R$ NT sequence instead of the full-length receptor. These experiments confirmed all cross-links that had been found for full-length $Y_2R$ in the lipid bicelle. On top of that, an additional cross-link of $L24^{NPY}$ to $E16^{NT}$ was exclusively detected in the XL-MS experiment with the isolated $Y_2R$ NT (aa 1–45) (Supplementary Data 1). Furthermore, we monitored binding of site-specifically $^{13}C/^{15}N$-labeled NPY variants[28] to the $Y_2R$ NT peptide by NMR spectroscopy (Supplementary Fig. 4). NMR is complementary to the XL-MS technique as it is biased towards the very mobile conformations that

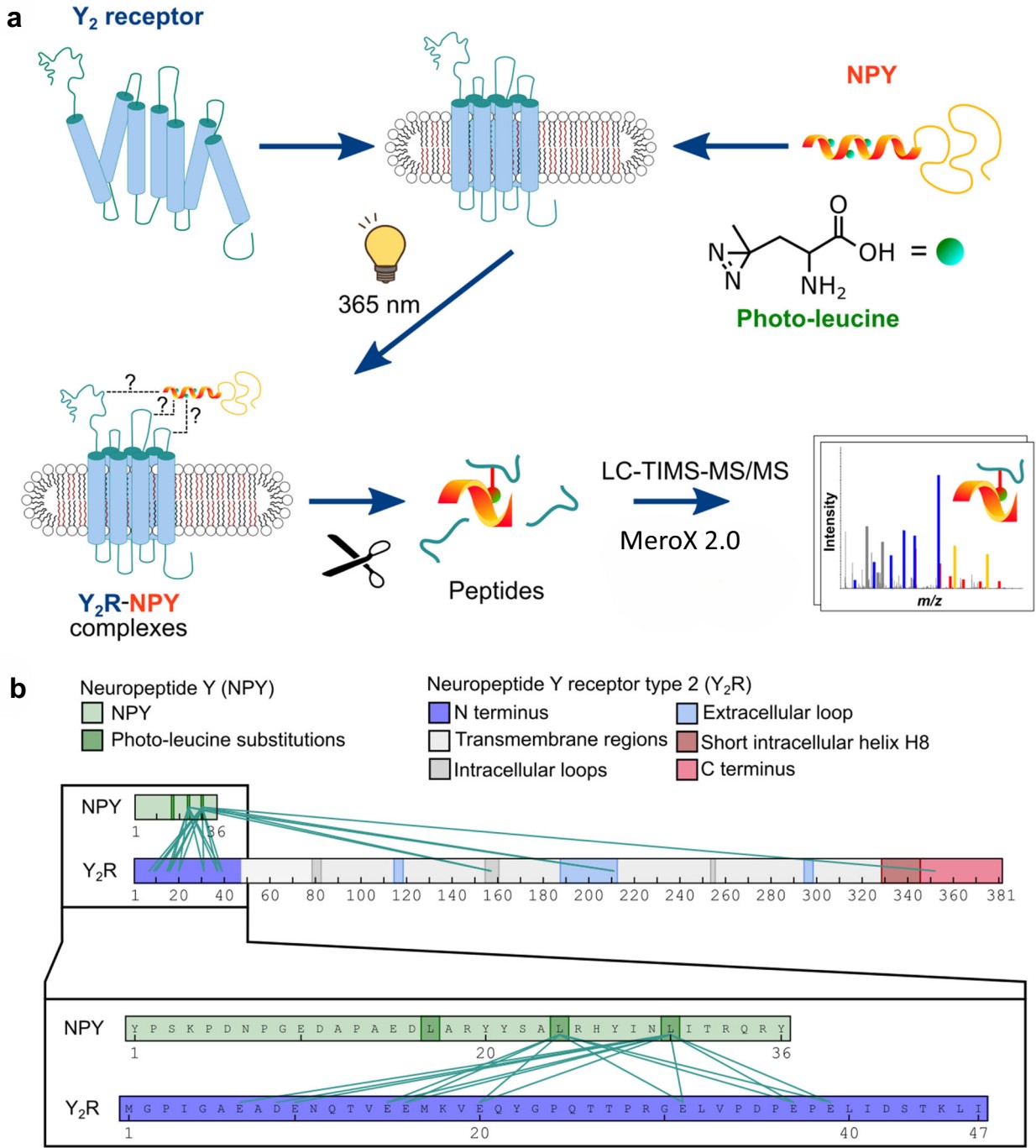

**Fig. 1 | XL-MS between NPY and Y₂R. a** Scheme of photoaffinity labeling. NPY containing photo-leucines at positions 17, 24, and 30 was cross-linked to bicelle-reconstituted Y₂R preparations by UV-A light. The cross-linked complex is enzymatically digested, peptides are analyzed by liquid chromatography and trapped ion mobility spectroscopy tandem mass spectrometry (LC-TIMS-MS/MS) using the MeroX software. **b** Overview of XL-MS results. Each cross-link is represented by a green line. Inset: The majority of cross-links were found between L24$^{NPY}$ and L30$^{NPY}$ to Y₂R NT. Cross-linking experiments were conducted at least three times independently. Related to Supplementary Data 1, listing all observed cross-links.

can be driven apart by thermal energy anytime, and the method has high sensitivity for short-range interactions. Chemical shift changes are induced by conformational changes in the peptide complex upon binding, which alter the magnetic environment of a given nucleus. Indeed, weak changes in the chemical shifts of the labeled amino acids E15$^{NPY}$, L24$^{NPY}$, and R35$^{NPY}$ are seen upon binding to Y₂R NT, which confirm interactions between NPY and Y₂R NT that are strongest within the central helical portion of NPY.

## Conformational ensembles of the disordered Y₂R NT

We explored the NPY-Y₂R structure in greater detail using the XL-MS data as distance constraints for Rosetta modeling and MD simulations to sample the conformational space of Y₂R NT. Different algorithms consistently predict Y₂R NT to be intrinsically disordered (Supplementary Fig. 1), suggesting that no stable secondary structures for the isolated Y₂R NT exist. Nevertheless, it is tempting to speculate that in the presence of the peptide ligand, the conformational space of the NT

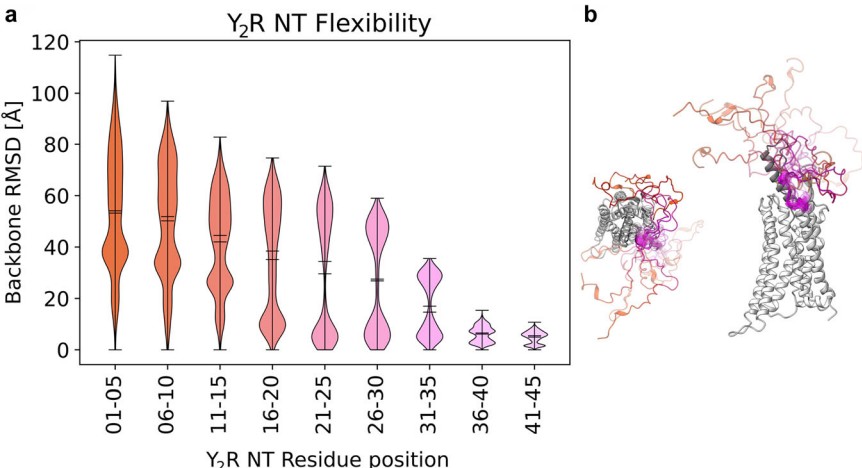

**Fig. 2 | Structural flexibility of the Y₂R NT in microsecond MD simulations.**
**a** Violin plot of the backbone-RMSD values for each residue of the NT during the MD simulations, relative to the initial frame, grouped in sets of five. **b** Example conformations sampled during MD simulations. Residue coloring matches the groups in the violin plot with a color gradient from orange to pink from distal to membrane-proximal regions and opacity increasing over simulation time.

is confined by transient interactions resulting in diverse conformational ensembles. We employed the AlphaFold2[45] and Rosetta[46] algorithms to create 3D-structural models of full-length Y₂R in the presence of NPY. In particular, we generated an ensemble of active-like state Y₂R-NPY models in the presence of the disordered NT surrounding the NPY peptide. Then, we included all cross-links identified by XL-MS in our Rosetta refinement protocol (Supplementary Fig. 5). Indeed, the resulting model shows a combination of structural features: All distance constraints are fulfilled within a $C_\beta$-$C_\beta$ distance of 13 Å (Supplementary Fig. 6), the last L40$^{NT}$ residue that was structurally resolved in the cryo-EM structure is correctly positioned, and the NT turns away from the ligand and wraps around the outside of ECL2 (Supplementary Fig. 6).

Using the hybrid model as a starting point, we performed MD simulations to monitor the dynamics of the NPY-Y₂R complex. Given the flexibility of the NT, we performed 10 independent simulations of 1 μs each to sample the conformational space of the modeled Y₂R NT. During the simulation, the Y₂R NT samples a large conformational space, which is in agreement with its proposed disordered character (Fig. 2). We observe increasing flexibility towards the NT, raising from an average RMSD value of 5–10 Å to 50 Å. In our simulations, the proximal region of the NT (aa 21–45) remains in close proximity to the upper part of TM7 and ECL3, while the distal region (aa 1–20) fluctuates freely (Fig. 2). Within the Y₂R NT, the two acidic patches E15–E20$^{NT}$ and D35–E39$^{NT}$ fluctuate around a mean distance of 24 Å. Accordingly, we speculate that the negatively charged patches from these acidic residues - due to their repulsive forces – might keep the ligand binding pocket of the Y₂R NT in an 'open' conformation.

Residues from the proximal region of the NT are frequently in contact distance of 4.5 Å with residues from the TM region and the peptide ligand (Supplementary Fig. 7). The most frequent contacts towards the TM region were accounted for by T44$^{NT}$-Q50$^{TM1}$ (74%), D42$^{NT}$-K304$^{7.32}$ (62%), K45$^{NT}$-Q50$^{TM1}$ (53%), and T44$^{NT}$-K304$^{7.32}$ (51%) (Supplementary Fig. 7). In addition, we observed highly frequent contacts of the NT to NPY, such as E39$^{NT}$-R25$^{NPY}$ (75%), L40$^{NT}$-R25$^{NPY}$ (74%), L40$^{NT}$-L24$^{NPY}$ (73%), L40$^{NT}$-Y21$^{NPY}$ (73%), and L40$^{NT}$-I28$^{NPY}$ (61%) (Supplementary Fig. 7). Typical for an IDR, the contacts of the Y₂R NT with the peptide ligand interchanged frequently during our simulations. Nonetheless, the acidic patches E15–E20$^{NT}$ and D35–E39$^{NT}$ tend to remain in contact with NPY during the simulation time, visualized by high line opacity in the flare plot (Fig. 3). While the membrane-proximal region D35–E39$^{NT}$ mainly contacts the central part of the NPY helix around residues Y21–N29$^{NPY}$, the more distal NT residues

Q12–E20$^{NT}$ interact with both the central part of NPY (Y27$^{NPY}$) and the N-terminus of NPY (Y1–K4$^{NPY}$). In agreement with the MD analysis of native contacts, L24 and L30 of NPY remain in putative cross-linking distance to the membrane-proximal receptor region (e.g., L24$^{NPY}$-E37$^{NT}$, L24$^{NPY}$-E39$^{NT}$, L30$^{NPY}$-E37$^{NT}$, L30$^{NPY}$-E39$^{NT}$ (Supplementary Fig. 8). This observation is in agreement with the contact pattern observed in the initial Rosetta model from which the simulations were started (cf. Supplementary Fig. 6). In summary, the MD simulations reveal the disordered character of the Y₂R NT. Moreover, they suggest that the huge conformational space adopted by the NT is restricted by electrostatic forces, specifically by some frequently observed interactions between the flexible NT and the peptide ligand.

### Y₂R NT modulates arrestin-3 binding, but not G$_i$ activation

Intrigued by our findings of the XL-MS experiments and MD simulations, we next interrogated how the Y₂R NT might modulate receptor function. Therefore, we created a series of receptor mutants by mutating the primary cross-linking sites around the acidic clusters E15–E20$^{NT}$ and D35–E39$^{NT}$ to neutral asparagine/glutamine and positively charged lysine residues, or by deleting the entire flexible part of the NT (Δ2–41). Furthermore, we mutated a stretch of three polar residues D42-S43-T44$^{NT}$ to alanine, which appears to stabilize a turn-like structure in the structurally well-resolved membrane-proximal part of the NT (PDB 7DDZ, 7X9B, 7YON, refs. 25–27) and mediate contacts to the ECL3 (Fig. 4a).

All receptor variants were exported to the plasma membrane in transiently transfected HEK293 cells, and showed an expression level of ≥ 50% compared to the wild-type receptor (Fig. 4b, c). We first investigated the signaling properties of Y₂R NT variants towards the canonical G$_i$-pathway (Fig. 4d, e) and recruitment of arrestin-3 (Fig. 4f, g). Wild-type Y₂R activated G$_{i1}$ in a direct BRET-based readout with an EC$_{50}$ of 0.4 nM. This was hardly affected by mutations in the acidic clusters, with the exception of the charge reversal at D/E35-39K (acidic patch 2), which displayed a very subtle, but statistically significant two-fold shifted EC$_{50}$ value (Supplementary Table 1). In line with these results, deletion of the flexible part of Y₂R in the Δ2–41 variant also displayed a moderate three to four-fold decreased EC$_{50}$, while the D42A-S43A-T44A variant was twelve-fold less potent to activate G$_{i1}$-proteins. We measured very similar effects for the activation of the neuronal G$_{oA}$ subtype in the same BRET-based setting, and in classic second messenger assays using a chimeric G$_{qi}$ protein that is very well established for NPY receptors[26,47,48] (Supplementary Table 1 and

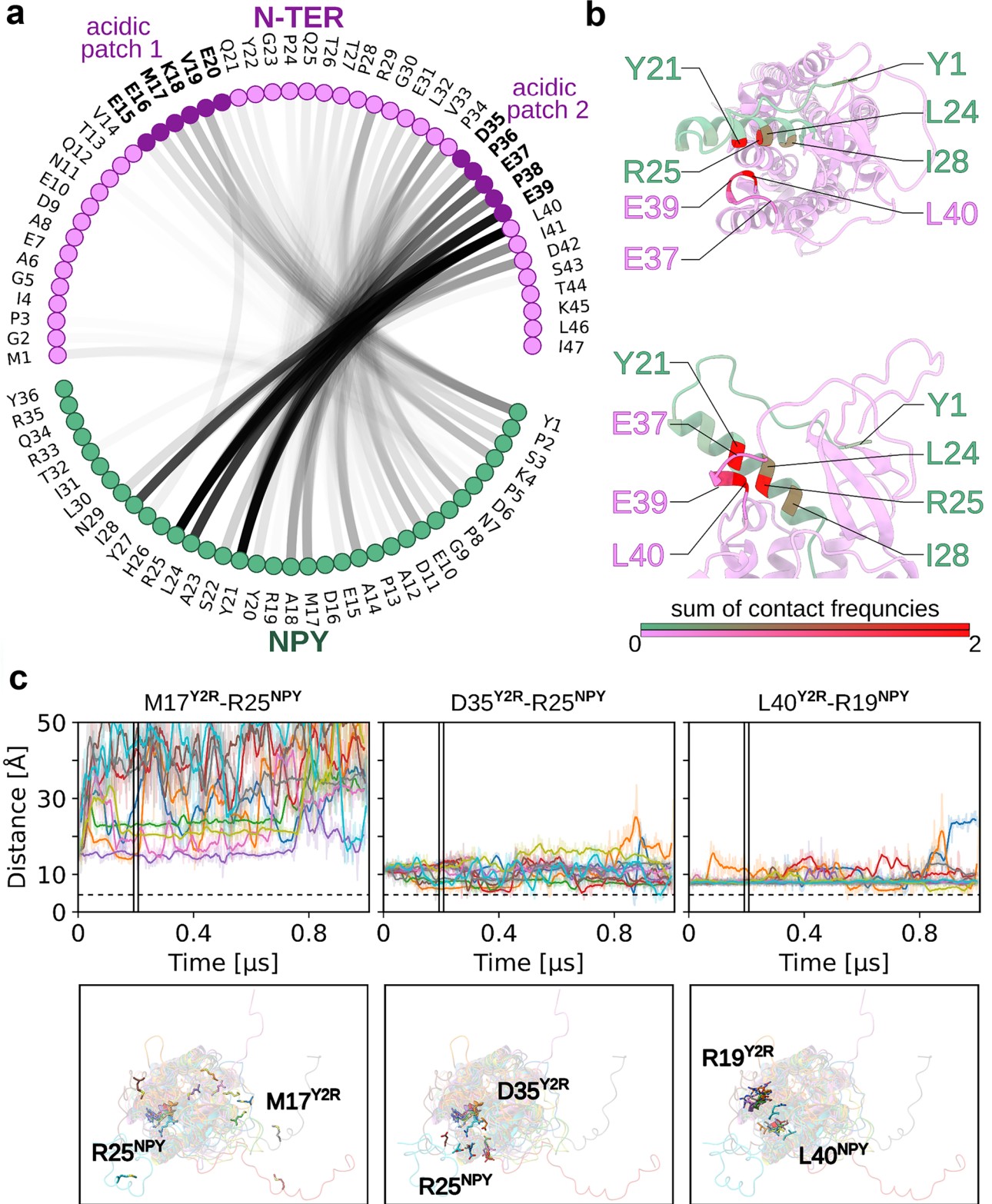

**Fig. 3 | Residue Contacts between Y₂R NT and NPY. a** Flare plot showing the contact frequency between residues of Y₂R NT and NPY from microsecond MD simulations. Contacts are defined as residues within a distance of 4.5 Å (excluding direct neighbors) and are represented by black lines. Line opacity indicates contact frequencies, with higher opacity reflecting more frequent contacts. The acidic patches E15-E20 and D35-E39 of the NT are highlighted. **b** Visualization of overall contact frequency between residues of NPY and Y₂R NT. Contact frequencies are shown by red hue intensity, with higher frequencies indicated by darker shades. Residues with high contact frequencies are labeled for clarity. **c** Distance plots for selected residue pairs M17$^{Y2R}$-R25$^{NPY}$, D35$^{Y2R}$-R25$^{NPY}$, and L40$^{Y2R}$-R19$^{NPY}$. Different colors represent different simulation runs. Runs were smoothed by averaging over ten frames per point. The original, unsmoothed time trace is shown in the same color with higher transparency. Corresponding structural snapshots at 0.2 μs are shown below the plots using the same color scheme.

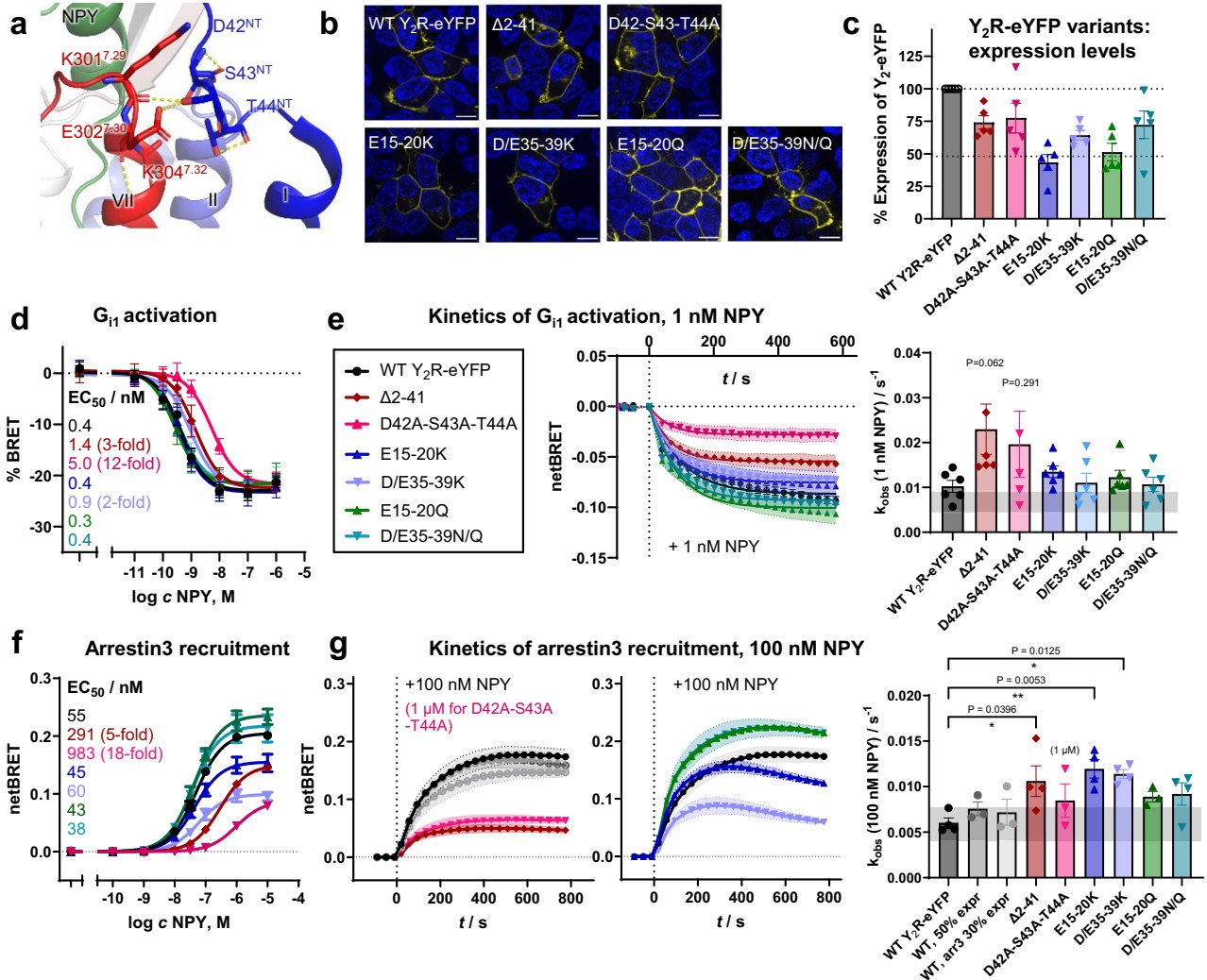

**Fig. 4 | Y₂R NT mutations differentially modulate G protein activation and recruitment of arrestin-3. a** Close-up view to membrane proximal Y₂R NT from cryo-EM (PDB 7X9B) highlighting interactions of D42-S43-T44[NT]. **b** Live-cell fluorescence microscopy shows that Y₂R NT variants are transported to the plasma membrane like the wild-type. Y₂R-eYFP variants are shown in yellow, cell nuclei are stained by H33342 and shown in blue, scale bar equals 10 μm. All pictures were acquired with identical light exposure and picture processing. **c** Quantification of cellular receptor expression is based on eYFP fluorescence in a plate reader, normalized to wild-type Y₂R. **d** Activity of Y₂R variants towards $G_{i1}$ proteins as measured by a direct BRET assay ($G_{i1}$-CASE) 10 min after ligand stimulation. All variants show full activation; D42A-S43A-T44A has the strongest shift in potency, while all others show mild or no effects. **e** Kinetics of $G_{i1}$ activation after stimulation with 1 nM NPY is overall similar for most variants except Δ2–41 and D42A-S43A-T44A, which have a tendency for faster apparent rate constants ($k_{obs}$). 95% CI of wild-type

Y₂R $k_{obs}$ is given as a gray rectangle for comparison. **f** Recruitment of arrestin-3 to Y₂R variants as measured by BRET 10 min after NPY stimulation. D42A-S43A-T44A and Δ2–41 variants reduce recruitment of arrestin-3 to the receptor. Neutral (in green) and charge-inverted (in blue) variants of the acidic patch in Y₂R NT lead to distinct behavior, with only charge reversal impairing arrestin-3 recruitment.
**g** Kinetic analysis of arrestin-3 recruitment after stimulation with 100 nM/1 μM NPY. Charge reversal in acidic patches 1 and 2 leads to faster initial recruitment, but also early signal decay. The bar plot shows quantification of the initial rate of arrestin-3 recruitment to receptor variants. Apparent rate constants ($k_{obs}$) are plotted in comparison to wild-type Y₂R-eYFP and its 95% CI (gray rectangle). * $P < 0.05$, ** $P < 0.01$ in one-way ANOVA, with Dunnett's post-hoc test corrected for multiple comparisons against wild-type Y₂R. For the color legend, please see panel (**e**). Data in (**c**–**g**) are the mean ± SEM of $n = 5$ (**c**), $n = 4$-9 (**d**), $n = 5$-6 (**e**), $n = 3$ (**f**), $n = 3$-4 (**g**) independent experiments, each performed in technical triplicate.

Supplementary Fig. 9). This suggests that the structurally well-resolved membrane proximal region around D42-S43-T44[NT] is critical for high-potency activation of G proteins, while the more distal areas, including the acidic patches, show smaller effects on G protein activation. Interestingly, however, there was a trend towards faster $G_{i1}$ activation in the Δ2–41 and D42A-S43A-T44A mutant (Fig. 4e), which might indicate faster ligand access.

Next, we investigated the effects of NT modifications on the recruitment of arrestin-3 (Fig. 4f, g). In contrast to G protein activation, the Y₂R NT strongly and differentially modulated the recruitment of arrestin-3 to the receptor (Fig. 4f, g). Charge reversal of the acidic clusters by mutation to lysine reduced the BRET_max to 49% for

D35–E39[NT], and 77% for E15–E20[NT] (Supplementary Table 1). The apparent ligand potency remained wild-type-like. In contrast, mutation of E15–E20[NT] or D35–E39[NT] to neutral asparagine/glutamine residues did not impair arrestin-3 recruitment to the receptor. For comparison, deletion of the entire flexible part of the NT (Δ2–41) reduced the recruitment of arrestin-3 to Y₂R to 75% with a five-fold reduced apparent potency of NPY. Arrestin-3 recruitment was further reduced to 44% in the D42A-S43A-T44A variant and required 18-fold higher NPY concentration, underlining a strong functional contribution of the membrane-proximal region. We validated the reduced recruitment of arrestin-3 by monitoring the internalization of all receptor variants upon NPY stimulation by live-cell fluorescence microscopy

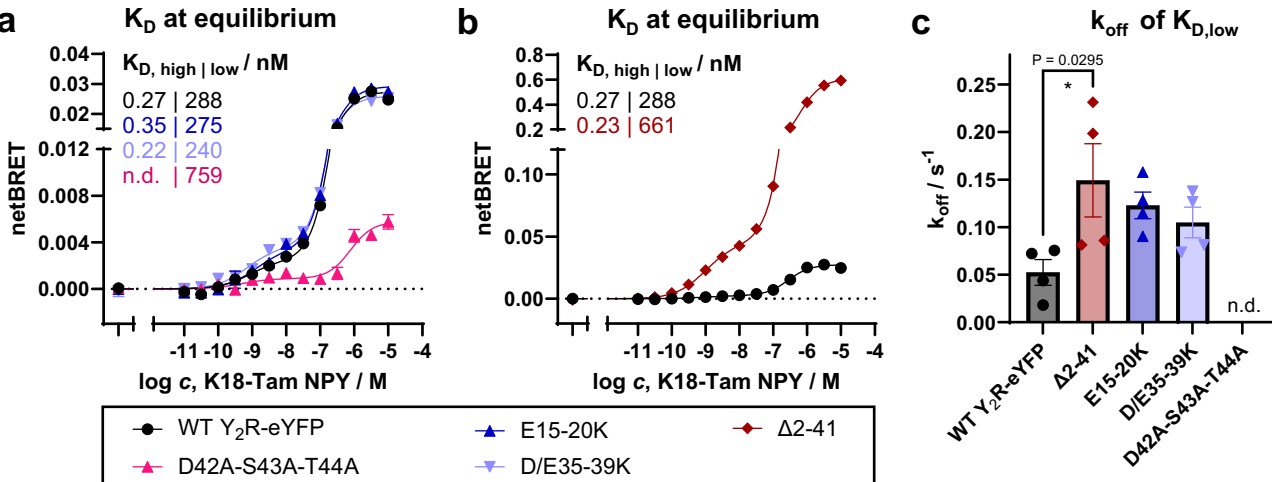

**Fig. 5 | Ligand binding at $Y_2R$ NT variants. a** Equilibrium binding between K18-Tamra-NPY and Nluc-$Y_2R$-eYFP variants. Charge reversal in the acidic patches retains wild-type-like equilibrium binding properties, while D42A-S43A-T44A displays a significantly smaller BRET window and about three-fold reduced low-affinity binding compared to wild-type $Y_2R$. **b** N-terminal deletion in $\Delta2$–41 $Y_2R$ increases overall BRET window as expected from the reduced distance, and the low-affinity state has about two-fold reduced affinity compared to wild-type $Y_2R$. Please note that the $Y$-axis scaling is different between panels (**a** and **b**). **c** Ligand dissociation from the low-affinity state (bound to 300 nM K18-Tamra-NPY; re-binding blocked by 50 μM antagonist) is faster for $\Delta2$–41 and acidic patch charge reversal mutants. Data are the mean ± SEM of $n = 3$-4 (**a**, **b**) or $n = 4$ (**c**) independent experiments, performed in technical duplicate or triplicate. * $P < 0.05$ in one-way ANOVA, with Dunnett's post-hoc test corrected for multiple comparisons against wild type $Y_2R$. Related to Supplementary Fig. 11: Expression and activity of Nluc-$Y_2R$-eYFP variants used for NanoBRET binding.

(Supplementary Fig. 10), which uses unmodified arrestin at endogenous expression levels. As a limiting factor, the internalization of $Y_2R$ contains arrestin-3-dependent and –independent components[49], which overlay in this type of analysis and are therefore expected to weaken the apparent effects of the arrestin-dependent pathway. Nonetheless, in agreement with the BRET results, the D42A-S43A-T44A variant, which shows the weakest arrestin-3 recruitment and lowest potency, displayed severely delayed internalization with ~100% (86–133%) of receptors still residing in the plasma membrane after 10 min stimulation with 100 nM NPY, compared to 32% for wild-type $Y_2R$ (Supplementary Fig. 10B). Internalization of the $\Delta2$–41 variant was also reduced to 58% (50–71%) receptors residing in the membrane after 10 min, although this remained a trend that did not reach statistical significance.

Kinetic analysis of the arrestin-3 interaction at a fixed NPY concentration of 100 nM (1 μM NPY for D42A-S43A-T44A because of the strongly shifted $EC_{50}$) by BRET revealed that modifications of the receptor NT generally accelerate the on-rate of arrestin-3, rather than slowing down complex formation (Fig. 4g and Supplementary Table 1). The $\Delta2$–41 variant showed a statistically significant 1.8-fold faster $k_{obs}$ than the wild-type despite its lower $EC_{50}$, and there was a similar trend for D42A-S43A-T44A. Mutating the acidic clusters E15–E20[NT] or D35–E39[NT] to neutral asparagine/glutamine residues also showed a tendency for faster arrestin-3 recruitment, which was even more enhanced for charge reversal to lysine in these acidic clusters, which accelerated early recruitment of arrestin-3 to the receptor by two-fold (Fig. 4g and Supplementary Table 1). Interestingly, however, for these two variants the BRET signal clearly decayed faster, starting after about 4 min, suggesting an overall destabilized complex for the E15–E20K and D35–E39K variants. To ensure that the kinetic differences observed are a result of the N-terminal variations rather than differences in expression level, we performed control experiments titrating the amounts of wild-type $Y_2R$ and/or arrestin-3 down to 50% and 30%. This did neither change the signal window nor the $k_{obs}$ rate (Fig. 4g).

To get more detailed insights into the modulation of arrestin-3 recruitment by $Y_2R$ NT variants, we next measured ligand binding using a NanoBRET readout in HEK293 membranes (Fig. 5). This provides access to affinity and relative orientation of a tetramethylrhodamine(Tamra)-labeled peptide (K18-Tamra-NPY) towards the NT of the receptor, which is genetically labeled with a Nanoluciferase (Nluc)[26]. The N-terminal addition of Nluc does not interfere with receptor function (ref. 26 and Supplementary Fig. 11). Wild-type $Y_2R$ displays a characteristic biphasic binding with a G-protein-dependent high-affinity state with subnanomolar $K_{D,high}$ and a G-protein-independent low-affinity state with $K_{D,low}$ ~ 300 nM (ref. 26). Both states overlap structurally and can be blocked with the small molecule antagonist JNJ-31020028[26]. Upon mutations in the membrane-proximal polar stretch in the D42A-S43A-T44A[NT] variants, the BRET window decreased drastically. A high-affinity state is hardly detectable due to the small measuring window, and its affinity cannot be determined with confidence, while the affinity of the low-affinity state is reduced by about three-fold compared to wild-type $Y_2R$ (Fig. 5a and Supplementary Table 2). This would be in agreement with a changed orientation of the NT pointing away from the binding pocket due to lack of interactions with ECL3/TM7, which weakens low- and potentially high-affinity binding of NPY. Vice versa, deletion of the flexible part of the NT including the acidic patches ($\Delta2$–41) increased the NanoBRET window as expected from the reduced distance between NT and labeled peptide ligand. High-affinity binding was intact and occurred to a similar proportion as in the wild-type receptor (Fig. 5b and Supplementary Table 2). Interestingly, however, the about two-fold reduced low-affinity binding constant of the $\Delta2$–41 variant supports a contribution of this region for NPY low affinity binding. When only the acidic patches were mutated (E15–E20K and D35–E39K), we did not detect alterations in NPY binding at equilibrium. There were neither changes in the high- or low-affinity state nor in the BRET window compared to wild-type $Y_2R$, suggesting unchanged affinity at equilibrium and overall highly similar binding orientation (Fig. 5a and Supplementary Table 2).

We further looked into the kinetics of ligand dissociation as a potential mechanism of the observed signaling bias of the $Y_2R$ NT variants. Ligand residence times can be one mechanism to regulate arrestin-3 interactions[50–52], as its recruitment to the receptor is a rather slow, multi-step process, while activation of G proteins is usually faster. This is also reflected in the apparent rates of $G_{i1}$ activation compared to the recruitment of arrestin-3 that we measured in our BRET settings for

$Y_2R$ variants (cf. Fig. 4e, g), where $G_{i1}$ activation was about two times faster when measured near the $EC_{50}$, even though separation of the heterotrimer needs to be preceded by nucleotide exchange and is therefore not the most immediate readout. Indeed, deletion of the flexible part of the NT in the $\Delta 2–41$ variant or charge reversal in the acidic patches increased ligand dissociation by two to three-fold (Fig. 5c and Supplementary Table 2), providing a molecular basis for the reduced arrestin-3 interaction despite virtually unchanged equilibrium binding properties.

## Discussion

The structural and functional contribution of the NT of most rhodopsin-like GPCRs has remained enigmatic to date. Due to their typical high flexibility, N- as well as C-termini of receptors were often removed for structural studies or even replaced by well-structured T4 lysozyme or similar proteins to aid crystallization[53,54]. In case the native termini were present in the protein sample, these regions were usually not resolved in the high-resolution 3D structures, such as the recent cryo-EM structure[26] of the human $Y_2R$ bound to its endogenous ligand NPY and $G_{i1}$. It has been predicted that the NT of rhodopsin-like GPCRs are IDRs[55] that adopt context-specific conformational ensembles to fulfill specific functions. We speculate that the high structural flexibility is vital for GPCR function. Notwithstanding, our understanding of the different interaction partners or even interaction modes of IDRs in GPCRs is limited[13,55,56] and the high diversity of sequences among different receptors makes functional predictions even more difficult. However, the field has recently gained momentum with a few studies on the intrinsically disordered C-terminus and ICL3 of the $\beta_2AR$ modulating receptor signaling[57–60].

We applied XL-MS to capture the conformational ensembles of NPY binding to $Y_2R$ with a special focus on $Y_2R$'s disordered NT. So far, photo-cross-linkers have been incorporated site-specifically into GPCRs by amber stop codon suppression, and cross-links were read out by Western blot analyses. This approach has provided impressive footprints of peptide ligands in their respective binding pockets[61–67]. However, suitable photo-reactive cross-linkers for amber stop codon suppression, such as $p$-benzoyl-Phe (Bpa), are bulky and hydrophobic. Diazirines might be incorporated into proteins as photo-leucine or photo-methionine, which closely resemble their natural counterparts and are accepted by the endogenous tRNA synthetase[68]. Hence, photo-leucine or photo-methionine residues are well suited for global labeling in proteins or can be introduced site-specifically by chemical synthesis in peptides. A major advantage of diazirines is their fast reaction time in the ns to µs timescale[34], allowing to capture transient conformations. By incorporating diazirine in photo-leucine residues site-specifically into positions L17, L24 and L30 of NPY by solid-phase peptide synthesis, we gained an unbiased view of interactions in the µs timescale. In addition to cross-links to the ECL regions of the receptor that recapitulated the cryo-EM structure, we identified a number of previously unknown contacts of $L24^{NPY}$ and $L30^{NPY}$ to $Y_2R$ NT. In addition to XL-MS of full-length $Y_2R$, potential contacts of NPY to $Y_2R$ NT were independently confirmed by XL-MS and NMR studies using a peptide fragment (aa 1–45) of $Y_2R$ NT. XL-MS and NMR are complementary techniques, with NMR being more sensitive towards highly mobile conformations and short-range interactions, as thermal energy can drive molecules apart anytime. NMR experiments confirmed interactions between NPY and $Y_2R$ NT that are strongest within the central helical portion of NPY. Compared to NMR experiments with full length $Y_2R$[28], the pattern of chemical shifts appeared different, indicating a modulation of NPY binding by the TM region as expected by the tight interactions of NPY's C-terminus with the transmembrane bundle (PDB 7X9B, ref. 26). This is reflected most dramatically for R35 in NPY's C-terminus, which is tightly engaged when bound to full length $Y_2R$, while its changes in chemical shift were much weaker when only the isolated $Y_2R$ NT is present (Supplementary Fig. 4).

The majority of cross-links between the peptide ligand and $Y_2R$ NT was concentrated on two acidic clusters around $E15–E20^{NT}$ and $D35–E39^{NT}$ of the receptor. In our MD simulations, these acidic patches in $Y_2R$ NT fluctuated around a distance of 24 Å, and hence they shape an electrostatic network on the extracellular surface of the receptor. Since diazirines have a preference for cross-linking with acidic amino acids[35,36], we interrogated whether these predominant interaction sites are a mere consequence of photo-crosslinking chemistry, or if electrostatic interactions between NPY and $Y_2R$ NT are the basis for forming cross-links. Overall, despite frequent interchanging of contacts during the 1 µs long MD simulations, the experimentally derived cross-links remained enriched during the simulations, strongly supporting specific interactions between $Y_2R$ NT and the central helix of NPY. Electrostatic interactions are the most long-range and strongest non-covalent interactions, and it appears likely that an acidic receptor NT is involved in ligand recognition of the arginine-rich peptide ligand, in particular given the large potential interaction surface. NPY contains one lysine and four arginine residues, among them $R33^{NPY}$ and $R35^{NPY}$ in the very C-terminus that are indispensable for biological function and are well-coordinated in the TM binding pocket in the active $G_i$-bound receptor state[26]. $R19^{NPY}$ and $R25^{NPY}$ are located in the central helix of the peptide and are more exposed to the extracellular part of $Y_2R$ in the bound state. Although charge complementarity seems to be an important factor for the interaction pattern, the neighboring regions also contributed. This is reflected by the enrichment of $R25^{NPY}$, but not $R19^{NPY}$, in the contact plots and the observation that exchange of acidic residues to neutral, but polar residues, is functionally accepted, as discussed below.

Mutating the acidic patches around $E15–E20^{NT}$ and $D35–E39^{NT}$ to neutral asparagine or glutamine residues or reverting the negative charge to positively charged lysine stretches still allowed for high-affinity NPY binding and activation of the canonical $G_i$ pathway. This is also reflected by the $\Delta 2–41$ variant, which lacks the flexible part of $Y_2R$ NT including both acidic patches, and only displayed a moderately three- to six-fold reduced potency in different $G_{i/o}$-based readouts (Fig. 4d, Supplementary Fig. 9 and Supplementary Table 1) in agreement with a previous study[47] using the same deletion (termed '$Y_2R$ ($\Delta N + 8$)', 2-fold reduced $EC_{50}$ compared to wild-type $Y_2R$). The lack of significant effects of $Y_2R$ NT on high-affinity binding and G protein activation can be explained by a very strong allosteric enhancement of ligand affinity by $G_{i/o}$ proteins[26]. In the absence of $G_i$, the affinity of NPY is rather low, with ~ 300 nM, but the NPY-$Y_2R$-$G_i$ ternary complex displays a high-affinity interaction with sub-nanomolar $K_D$, which is also reflected in the high functional affinity in G protein activation assays. This strong gain in affinity is probably mediated by a contraction of TM2 and TM6 compared to the inactive $Y_2R$ bound to an antagonist (PDB 7DDZ, ref. 25), and enables high-affinity interactions of ligand and receptor in the transmembrane pocket. Accordingly, minor changes in the extracellular contacts have a negligible impact on the affinity of the G-protein-bound complex.

In contrast to the mild effects on G protein activation, the $Y_2R$ NT strongly modulated the recruitment of arrestin-3 to the receptor. This interaction occurs with significantly lower potency of around 50 nM, and hence, does not seem to be allosterically stabilized by the intracellular partner and changes in the extracellular interactions should become more visible. This is consistent with the observed two- to three-fold reduction in affinity of the G-protein-independent low-affinity state when deleting the flexible part of $Y_2R$'s NT in the $\Delta 2–41$ variant or dislodging the entire NT by D42A-S43A-T44A mutation in the membrane proximal region (see below) (Fig. 5a, b and Supplementary Table 2). Moreover, the potency to recruit arrestin-3 was further reduced for these variants compared to wild-type $Y_2R$ (Fig. 4f and Supplementary Table 2). Mutations in the acidic patches of $Y_2R$ NT had differential effects on the recruitment of arrestin-3 depending on the charge state. Neutral, but polar substitution of $E15–E20^{NT}$ or

D35–E39[NT] to glutamine or asparagine residues preserved arrestin-3 recruitment and even showed a tendency for increased interaction. In contrast, substitution to positive charges in the Y$_2$R E15K–E20K[NT] and Y$_2$R D35K–E39K[NT] variants showed a strongly reduced recruitment of arrestin-3. This apparently originates from reduced stability of the complex as reflected by a BRET decay already after 4–5 min at a sub-saturating ligand concentration of 100 nM, while the signal of the wild-type Y$_2$R-arrestin-3 complex remained stable for ≥15 min under the experimental conditions. This was underlined by increased k$_{off}$ rates and hence reduced ligand residence time in ligand binding experiments for the charge reversal variants of the acidic patches and the Δ2–41 deletion, even though we could not detect significant changes in equilibrium binding for the E15–E20K and D/E35–E39K variants (Fig. 5 and Supplementary Table 2). We speculate that an increased k$_{on}$ rate of the ligand at least partly compensates the increased off-rate, such that equilibrium binding is only minimally affected. This is supported by the observed increased apparent rates (k$_{obs}$) of arrestin-3 recruitment and G$_{i1}$ activation for the Δ2–41 deletion, and partly for the charge reversal mutants (Fig. 4e, g and Supplementary Table 2).

Our data suggest that the flexible Y$_2$R NT, including its negatively charged acidic patches, controls an electrostatic network within the NT and seems to have a dual role. On the one hand, transient contacts between the Y$_2$R NT and the peptide ligand contribute to the low-affinity binding state and prolong ligand residence time, which is required for efficient interactions of Y$_2$R with arrestin-3, but not G$_i$. On the other hand, we suggest that the NT kinetically limits ligand access, and the observed on-rate is accelerated upon deletion of this stretch or electrostatic perturbation. In this context, exchange of acidic to polar residues still permits ligand contacts and enables participation in the dynamic interaction network. Activation of G$_i$ is largely unaffected by changes in the disordered N-terminus, as is experiences strong allosteric enhancement of ligand binding from the transmembrane area in the ternary complex, and full activation of this pathway might be additionally enhanced by G$_i$ pre-assembly[69]. In contrast, perturbations in extracellular binding and the associated weakening of the low-affinity state, as well as increased off-rate, have greater effects on the binding of arrestin-3, which is recruited in a slower, multistep process.

In addition to the role of the dynamic interaction network in the disordered NT of Y$_2$R, our data also show an important contribution of the more ordered membrane-proximal part of the NT for the integrity of the Y$_2$R binding pocket. The polar stretch D42-S43-T44[NT], which was left intact in the Δ2–41 mutant, proved essential for receptor functionality, not only for interactions with arrestin-3, but also pertaining high-affinity ligand binding and activation of G$_i$. This region forms a small loop and has multiple polar contacts to the ECL3 in the cryo-EM structures[25–27], thereby orienting the NT away from the membrane towards the extracellular space. In the D42A-S43A-T44A variant, these contacts are lost, and the NT has an overall increased distance to the TM binding pocket as reflected by a five-fold reduced window in nanoBRET-based ligand binding assays. We note that modification of this region alters the relative orientation of the entire N-terminus and will also affect conformation and interactions of the disordered distal part. Therefore, the functional effects should be interpreted as the sum of direct effects from changes in the membrane proximal area plus impairment of the interactions of the disordered distal Y$_2$R NT. Accordingly, the D42A-S43A-T44A variant displayed a three-fold reduction in the K$_D$ of the low-affinity state. Due to the very small measuring window, we were unable to determine the affinity of the high-affinity state with reasonable accuracy. Activation of G$_i$ was still possible, but occurred with 18-fold reduced potency. Similarly, arrestin-3 recruitment to this receptor variant occurred with low potency and the maximal receptor-arrestin-3-complexes were reduced to ~40% compared to wild-type Y$_2$R. We confirmed that these severe functional effects originate from loss of contacts to ECL3/TM7 by mutagenesis of the interacting residues in ECL3 (Supplementary Fig. 12

and Supplementary Table 3). The point mutation E302[7.30]A reduced the potency to activate G$_{i1}$ by three-fold and reduced recruitment of arrestin-3 to 65% with four-fold reduced potency, thus resembling the effects of the mutation in the N-terminus. Similarly, K304[7.32]A reduced arrestin-3 recruitment to 66%. While those single mutants remained relatively mild in their effects (similar to single exchanges in the membrane proximal NT, cf. Supplementary Fig. 11C), introducing multiple alanine exchanges in this stretch in ECL3/TM7 severely decreased receptor expression (Supplementary Fig. 12B, C), thus precluding complete inhibition of hydrogen bonding from this side.

Such a stabilizing role of the membrane-proximal stretch might be a more common feature among GPCRs. Within the NPY family, a similar contact between the membrane-proximal NT and ECL3 is reinforced by a disulfide bridge in the Y$_1$R (C33[NT] – C296[ECL3]) and a homologous disulfide is also likely for the Y$_4$R (C34[NT] – C298[ECL3])[26]. Similarly, in the angiotensin II receptor 1 (AT1R)[70] and endothelin receptors[71] a disulfide bridge anchors the membrane-proximal NT to the ECL3. Consistent with this hypothesis of a structural anchor in the proximal NT of Y$_2$R, residues I41[NT] through L46[NT] showed limited movement in the MD simulations as reflected by the smallest average RMSD per five residues (Fig. 2), and a very high contact frequency between D42/S43/T44[NT] and E302/K304[ECL3] (Supplementary Fig. 7). In contrast, the more distal NT appeared more flexible with RMSDs ≥40 Å, in particular for residues 1–5 and 6–10. The high flexibility towards the N-terminal amino acid apparently creates a 'shield' over the transmembrane binding pocket, which controls ligand access, but also contributes to ligand binding and prolongs ligand residence time.

In conclusion, we combined highly sensitive XL-MS with state-of-the-art computational modeling and MD simulations to investigate the highly dynamic conformational ensemble of the intrinsically disordered Y$_2$R NT and the interactions with its endogenous peptide ligand NPY. We identified a transient interaction network between two acidic clusters in the Y$_2$R NT with the central helix of NPY that had not been captured by traditional structural biology approaches. These interactions contribute to the low-affinity binding site of the Y$_2$R and prolong the ligand residence time, which is required for efficient recruitment of arrestin-3 to the Y$_2$R, while G$_i$ activation does not depend on the receptor NT.

## Methods

### Peptide synthesis

Fluorenylmethoxycarbonyl (Fmoc)-protected amino acid building blocks and coupling reagents were purchased from Iris Biotech (Marktredwitz, Germany) unless stated otherwise. NPY peptides were synthesized by combined automated/manual synthesis on solid support using an Fmoc/*tert*-butyl strategy, following established protocols[72,73]. Briefly, 15 µmol of Rink Amide aminomethyl or Tentagel Rink amide resin (NovaSyn TGR R resin; Merck KGaA, Darmstadt, Germany) were used, and the peptide sequence was assembled from C- to N-terminus using repeated cycles of Fmoc deprotection, COOH activation and coupling of the consecutive amino acid. Fmoc was cleaved off in the last cycle to obtain a free N-terminus before the peptides were cleaved off the resin. Automated synthesis of the NPY-derivatives was performed in a SYRO I synthesis robot (MultiSyn Tech, Bochum, Germany) using N,N-dimethylformamide (DMF) as solvent. Fmoc deprotection was performed using 40% piperidine/DMF solution for 3 min, followed by 20% piperidine/DMF solution for 10 min. Amino acid coupling was performed as double couplings using 8 eq. amino acid, in situ activated with ethyl cyanohydroxyiminoacetate (Oxyma), and N,N'-diisopropylcarbodiimide (DIC) for 40 min per coupling. Photo-leucine was coupled manually using 3 eq. of Fmoc-photoLeu-OH, DIC, and HOBt. The coupling was carried out twice for 2 h and 16 h. K[18]-Tamra-NPY was synthesized as described[25].

The N-terminal sequence of Y$_2$R (Y$_2$R_1-45) was synthesized by using microwave-assisted automated synthesis in a LibertyBlue robot

(CEM, Kamp-Lintfort, Germany) on Rink Amide ProTide resin in 100 μmol scale. All couplings were carried out as single couplings with 5 eq. of amino acid, Oxyma and DIC each, with the exception of Fmoc-L-Arg(Pbf)-OH, which was coupled twice. All couplings were performed for 15 s at 140 W with a target temperature of 75 °C, followed by 110 s at 30 W with a target temperature of 100 °C. Fmoc deprotection was performed with 20% piperidine solution with the addition of 0.1 M HOBt at 75 °C for 15 s at 175 W, followed by 90 °C for 50 s at 30 W.

Cleavage from the resin was routinely performed using trifluoroacetic acid (TFA)/H₂O/triisopropyl silane (90/5/5, v/v/v/) for 3 h. Y₂R(1–45) was cleaved with TFA/thioanisole/ 3,6-dioxa-1,8-octanedithiol (DODT) (90/7/3, v/v/v) for 3 h. The peptides were then precipitated using 10 ml ice-cold diethyl ether. The precipitate was washed at least four times with 10 ml ice-cold diethyl ether, dried and dissolved in 20% acetonitrile (ACN) in H₂O for analytics and purification. Reversed-phase high-performance liquid chromatography (RP-HPLC) analysis and purification was carried out using a linear gradient of eluent B (0.08% TFA in ACN) in eluent A (0.1% TFA in H₂O). Peptides were purified to >95% on a Kinetex 5 μm XB-C18 100 Å column (Phenomenex, Torrance, USA). Purity of the peptides was determined by RP-HPLC on a Jupiter 4 μm Proteo 90 Å C12 (Phenomenex) and Aeris 3.6 μm 100 Å XB-C18 (Phenomenex) column. The identity of the peptides was confirmed by MALDI-TOF MS on an Ultraflex II (Bruker Daltonics, Billerica, USA).

### Y₂R production, purification, and refolding

A cysteine-deficient Y₂R[74] with a C-terminal 8x His tag was recombinantly expressed in *E. coli* as inclusion bodies using a fed-batch fermentation process[75]. Inclusion bodies were solubilized and purified in sodium dodecyl sulfate (SDS), and subsequently functionally reconstituted in isotropic DMPC/DHPC-c7 bicelles, as described before[40]. In brief, purification of the denatured Y₂ receptor was done by Metal Chelate Affinity Chromatography (IMAC) using a HisPrepTM FF 16/10 column (GE Healthcare). Purified Y₂R in 15 mM SDS, 50 mM sodium phosphate (NaP) pH 8, was diluted to 0.5 mg/ml and dialyzed against a degassed buffer (0.5 mM SDS, 50 mM NaP, 1 mM EDTA, 1 mM reduced glutathione (GSH), and 0.5 mM oxidized glutathione (GSSG)) at pH 8.5 for 48 h for functional disulfide formation. Preformed lipid bicelles (DMPC/DHPC-c7 (Avanti Polar Lipids, Alabaster, USA) in a molar ratio of 1/4) were incubated with the Y₂R at a molar ratio of 1/600/2400 receptor/DMPC/DHPC-c7, followed by three cycles of fast temperature changes from 42 °C to 0 °C.

### Solution NMR experiments

Solution NMR experiments of specifically ¹³C/¹⁵N labeled NPY in the absence and in the presence of the Y₂R N-terminal fragment were carried out on a Bruker Avance III 600 MHz NMR spectrometer using a standard 5 mm inverse triple resonance probe with z-gradient at a temperature of 27 °C. NPY was dissolved at a concentration of 400 μM in buffer (5 mM DHPC, 50 mM NaP at pH 7). The N-terminal Y₂R peptide was added at a 1:1 molar ratio. Fast phase-sensitive gradient-enhanced ¹H-¹⁵N HSQC experiments using WATERGATE (3-9-19) solvent suppression were carried out using 8 μs ¹H and 37 μs ¹⁵N π/2 pulses and 64 transients per $t_1$ increment. Spectra were analyzed using Topspin 4.4.1 (Bruker, Billerica, MA). Chemical shift perturbations (CSP) were calculated according to Eq. (1)

$$CSP\left(\Delta^1H, \Delta^{15}N\right) = \sqrt{\left(\Delta^1H\right)^2 + \left(\Delta^{15}N/5\right)^2} \qquad (1)$$

### XL-MS

**Cross-linking, enzymatic digestion, and LC-MS.** 1.37 μM bicelle-reconstituted full-length Y₂R or synthetic N-terminal peptide of Y₂R (aa 1–45) in 50 mM HEPES (pH 7.2) containing 1.5 mM DHPC were incubated for 1 h on ice in darkness with a triply diazirine-substituted photo-leucine [L17pL, L24pL, L30pL] NPY variant (final concentration 13.6 μM). For competition experiments, the receptor was additionally incubated with either 50× molar excess of the antagonist JNJ-31020028 (Selleck Europe) or 50× molar excess unlabeled NPY. Cross-linking was induced by UV-A irradiation (365 nm) with an LED lamp (ANUJ3000, Panasonic, Ottobrunn, Germany) at 30 J cm⁻². Samples were denatured in 10% SDS, 100 mM Tris-HCl (pH 7.5) and subjected to S-Trap enrichment (Protifi, Fairport, NY, USA). Afterwards, proteins were reduced, alkylated, and enzymatically digested with AspN and trypsin according to an existing protocol[76]. Samples were stored at − 20 °C before LC-MS analysis.

Peptide mixtures were reconstituted in an aqueous solution of 3% (v/v) ACN and 0.05% (v/v) TFA prior to LC-TIMS-MS/MS analysis on an UltiMate 3000 RSLC nano-HPLC system (Thermo Fisher Scientific, Waltham, MA, USA) coupled to a timsTOF Pro mass spectrometer (Bruker Daltonics, Bremen, Germany). Peptides were trapped and desalted on a C18 precolumn (Acclaim PepMap 100, 300 μm × 5 mm, 5 μm, 100 Å, Thermo Fisher Scientific, Waltham, MA, USA) with aqueous 0.1 % (v/v) TFA at a flow rate of 30 μL/min (precolumn temperature 50 °C). Using a flow of 300 nL/min, peptide mixtures were then eluted and separated on a self-packed Picofrit C18 column, 75 μm ID x 40 cm, Tip ID 15 μm (New Objective, Littleton, MA, USA), packed with Reprosil-Pur 120 C18-AQ, 3 μm, 120 Å material (Dr. Maisch GmbH, Ammerbuch, Germany) or a μPAC 50 C18 column (PharmaFluidics, Thermo Fisher Scientific, Germany). Linear gradients were from 3% to 50% B over 90 min, 50% to 85% B (over 5 min) and 85% B (5 min); solvent A: water containing 0.1% (v/v) formic acid and solvent B: acetonitrile containing 0.1% (v/v) formic acid, the separation column was kept at 40 °C. To align results obtained with different chromatographic conditions, unmodified peptides were used as indexed retention time (iRT) standards.

Data collection was performed in data-dependent acquisition (DDA) and data-independent acquisition (DIA) modes using parallel accumulation-serial fragmentation (PASEF) with 200 ms ramps for ion accumulation and separation. In DDA-PASEF, two mobility-dependent collision energy ramps were used: i) low: 59 eV at an inverse reduced mobility (1/$K_0$) of 1.6 V·s/cm² and 20 eV at 0.6 V·s/cm² and ii) standard: 95 eV at 1.60 V·s/cm² and 20 eV at 0.6 V·s/cm². Collision energies were linearly interpolated between these two 1/$K_0$ values and kept constant above or below. The target intensity per individual PASEF precursor was set to 100,000 with an intensity threshold of 1000. 10 PASEF MS/MS scans were triggered per acquisition cycle, corresponding to a cycle time of 2.47 s. Precursor ions in an *m/z* range between 100 and 1700 with charge states between 2–8 were selected for fragmentation. Active exclusion was enabled for 0.5 min (mass width 0.015 Th, 1/$K_0$ width 0.100 V·s/cm²). For DIA-PASEF, isolation schemes were adapted to the sample matrix based on cross-linked peptides identified in the DDA-PASEF data. DDA-PASEF data was processed with DataAnalysis (v5.3; Bruker Daltonics, Bremen, Germany)

### Identification of cross-linked peptides

Identification of cross-links was performed by a MeroX (v. 2.0.1.7, ref. [77]) search against a fasta file containing the amino acid sequences of triply diazirine-substituted [L17pL, L24pL, L30pL] NPY and the cysteine-deficient human Y₂R. The following settings were employed: semi-specific proteolytic cleavage: C-terminal at Lys and ArgR and N-terminal at Asp and Glu with up to 3 missed cleavages for each site and a maximum of 5 missed cleavages per peptide; peptide length of 4–25 amino acids; fixed modifications: carbamidomethylation of Cys; variable modifications: oxidation of Met; precursor and fragment mass tolerances were set to 10 ppm. For cross-link detection, photo-leucine was set as a new amino acid (C₆H₁₁NO) and cross-links were defined as loss of CH₄ (–16.031 u). Two searches were performed per data set, considering photo-leucine reactivity towards (i) only the acidic amino

acids Asp and Glu and the protein C-terminus and (ii) reactivity towards all 20 amino acids. Peptide spectral matches were initially filtered to a 10% intensity score, a minimum score cut-off of 30 and 1% FDR (false discovery rate). Lenient FDR filters were compensated by manual validation of all proposed cross-links. In addition, DDA-PASEF data were analyzed with Skyline[78] as described recently[43]. Specifically, LC-MS/MS data were processed and simplified into peak lists in Mascot generic format (.MGF), compatible with most database search engines. Subsequently, these files were used for the identification of cross-linked products by MeroX. MeroX results, converted to Proxl XML files (https://github.com/yeastrc/proxl-import-merox), alongside with MGF files, were used to create spectral libraries of cross-linked products in Skyline. In addition to XL-MS data from photo-leucine-labeled NPY, previously obtained cross-linking data using photo-methionine-labeled $Y_2R$ were used as a basis for building comprehensive spectral libraries. The compiled ion mobility library was integrated into the document for extracting ion chromatograms (EICs) of the first three isotopes of each precursor ion from raw DDA-PASEF files, using 10 ppm extraction windows. Using the spectral libraries as described allowed DIA-PASEF data to be processed by Skyline, resulting in a more sensitive detection of cross-linked peptides compared to a DDA workflow.

MS/MS datasets used for cross-link identifications were also used for the identification of non-cross-linked peptides. MS/MS peak lists were analyzed using MS-GF + (v 2023.01.12, ref. [79]) and MyriMatch (v 2.2.140, ref. [80]). The search was conducted using SearchGUI (v 4.2.14, ref. [81]). Protein identification was conducted against a concatenated target/decoy protein database containing the sequence of porcine NPY (UniProt ID: P01304), the in-house produced cysteine deficient $Y_2R$, porcine trypsin (UniProt ID: P00761), flavastacin (UniProt ID: Q47899), and their respective reverse decoy sequences. The identification settings were as follows: trypsin, semi-specific, with a maximum of two missed cleavages, precursor and fragment ion error tolerance of 10 ppm; carbamidomethylation of Cys as a fixed modification and oxidation of Met as a variable modification. Peptides and proteins were inferred from the spectrum identification results using PeptideShaker version 2.2.25 (ref. [82]). Peptide spectrum matches (PSMs), peptides and proteins were validated at a 1% FDR and results exported to.mzID format.

## DDA-PASEF and DIA-PASEF data analysis with Skyline
A detailed analysis pipeline has been reported recently[43]. All data have been deposited to ProteomeXchange with identifier PXD051865 (https://proteomecentral.proteomexchange.org/cgi/GetDataset?ID=PXD051865).

## Computational modeling
AlphaFold2[45] version 2.1 was used to generate an ensemble of $Y_2R$ models with the disordered NT in contact with the NPY peptide bound to the receptor's orthosteric pocket. Since AlphaFold2 is biased toward predicting the inactive state of class A GPCRs, some models exhibited a hybrid conformation, in which the NT interacted with the bound NPY peptide while the transmembrane (TM) helices remained in an inactive-like state. To select a fully active model that also displayed a partially folded NPY-NT complex, we chose the structure that best balanced two criteria: (i) the lowest root-mean-square deviation (RMSD) to the fully active cryo-EM structure of the $Y_2$ receptor in complex with NPY and $G_{i1}$ (PDB ID 7X9B) and (ii) the lowest average residue-residue distances for cross-linked residue pairs from mass spectrometry (MS) data. The selected model was further refined using RosettaCM[83], incorporating the cryo-EM structure as a template and applying cross-linking data as distance constraints (Supplementary Table 4). Specifically, the maximum allowed distance between $C_\beta$ atom pairs without an energetic penalty was set to

13 Å. The final model was chosen based on the minimization of the REF2015 Rosetta score[84], which included logarithmic energetic penalties proportional to the number and magnitude of constraint violations.

## Molecular dynamics simulation
The final model of the computational modeling was used to probe the conformational dynamics of the complex by MD simulations. The stabilizing palmitoylation site C8.60, as well as E8.61, were not resolved in the cryo-EM template structure. Therefore, the residues C8.60 and E8.61 were added manually using PYMOL (version 2.5.2, Schrödinger LLC). Finally, during the setup using the CHARMM-GUI Membrane builder[85], C8.60 was palmitoylated using the CYSP patch option. The N- termini of the $Y_2$ receptor and NPY were capped with the patch NTER from the CHARMM force field. Furthermore, the Charmm-GUI patch CT1 was used for the C-terminal end (E8.61) of the $Y_2R$ model. The patch CT2 was used for the C-terminus of NPY. The receptor cavities were filled with water molecules using *dowser*[86]. The amino acids were kept in their standard protonation state of the CHARMM36m force field[87], with the exception of the $Y_2$ receptor residues D2.50 and D3.49, which were protonated to emulate the active state of the receptor[88,89]. The model was embedded in a POPC bilayer with 358 molecules in the upper leaflet and 356 molecules in the lower leaflet. The embedded structure was inserted into a simulation box with a length of ~160 Å (x and y) and a height of ~160 Å (z), filled with ~97500 TIP3[90] water and two chloride ions (to neutralize the system) using published procedures[85] (Supplementary Table 5).

The simulations were run in the NPT ensemble at a temperature of 310.15 K and a pressure of 1.013 bar using GROMACS 2024.2. The CHARMM36m force field[87] was employed for lipids and proteins. Bonds involving hydrogen were constrained with LINCS[91] to allow a time step of 2 fs. Each system containing about 400,000 atoms was energy-minimized with the steepest descent algorithm and 1000 kJ mol⁻¹ nm⁻¹ as the threshold. The system was equilibrated in seven steps with descending restraints of protein side chains, backbone and on the POPC molecules, similar to the default CHARMM-GUI Membrane builder approach (Supplementary Table 6). The last equilibration step, however, was elongated from 200 ns to 500 ns to facilitate a smoother transition between the force field used for modeling as well as refinement and the CHARMM36m force field used for the MD simulations[92]. A total of 10 independent MD simulations were run for 1 μs each. Contact mapping was performed with *mdciao*[93], and the visualization with the software *VMD*[94] and *ChimeraX*[95]. The MD-trajectories are accessible under https://proteinformatics.uni-leipzig.de/mdsrv.html?load=file://_nt/y2r_nt.js.

## Plasmids
The plasmids encoding human $Y_2R$ with C-terminal fusion of enhanced yellow fluorescent protein (eYFP)[96] and the corresponding variant with an additional Nanoluciferase (Nluc)[25] at the receptor N-terminus for binding assays have been described previously. Site-directed mutagenesis of the $Y_2R$ sequence was performed using a modified Quik-Change strategy[97] using Phusion Polymerase (ThermoFisher, Waltham, MA, USA) with the primers listed in Supplementary Table 7. The plasmid encoding Nluc fused to the N-terminus of arrestin-3 was generated as described[98]. The chimeric G protein $G\alpha_{\Delta 6qi4myr}$ (as described[99]) was obtained from Evi Kostenis. The G-CASE sensors[100] were obtained from Addgene (Go1-CASE, #168123; Gi1-CASE #168120).

## Cell Culture
HEK293 cells (DSMZ, Heidelberg, Germany, acc. no 305) were maintained in Dulbecco's Modified Eagle Medium (DMEM)/F12 nutrient mix + 15% fetal bovine serum (FBS Gold; Sigma Aldrich, St. Louis, MO, USA) at 37 °C and 5% $CO_2$ in a humidified atmosphere and were used

between passages 10 and 20. The cells were regularly tested negative for mycoplasma contamination.

## Live-cell microscopy

Cellular localization of $Y_2R$ variants was examined by live-cell microscopy. HEK293 were seeded into 8-well µ-slides (Ibidi, Gräfelfing, Germany) and cultured to a confluence of 70%. Plasmids encoding a C-terminal fusion protein of $Y_2R$ and eYFP in the pVitro2 vector[101] or N-terminal variants thereof, generated by a modified QuikChange mutagenesis strategy[97], were then transiently transfected into the cells using Lipofectamine2000 (Invitrogen, Waltham, MA, USA) according to the manufacturer's protocol using 500 ng DNA per well. 24 h post-transfection, the cells were imaged in Opti-MEM medium (Thermo Fisher Scientific, Waltham, MA, USA) containing 2.5 µg/mL H33342 nuclear stain (Sigma-Aldrich, St. Louis, MO, USA) using an Axio Observer Z1 microscopic setup with ApoTome.2 (Zeiss, Oberkochen, Germany: 63 ×/1.4 oil objective, filter settings (ex/em): YFP 500(20) nm / 535(30) nm, H33342 365(20) nm / 420(30) nm). Identical acquisition times and image processing were applied for wild-type and receptor mutants. From the processed pictures, the plasma membrane fluorescence was quantified using automated object identification in Cell Profiler v4.2.8 (ref. [102]) with H33342-stained nuclei as primary objects (allowed diameter 80 x 150 px), which allowed identification of cells (propagation method, adaptive threshold with minimum cross-entropy) and plasma membrane (cells shrunken by 8 px) as secondary and tertiary objects, respectively. The median fluorescence intensity of the plasma membrane of transfected cells was recorded. For any $Y_2R$ variant, ≥40 cells from 3-4 independent experiments were quantified.

## Arrestin recruitment assays

Interactions of $Y_2R$ with arrestin-3 were measured by BRET, devising a C-terminal fusion of eYFP to the $Y_2R$ and an N-terminal fusion of Nluc to arrestin-3, as described[42]. HEK293 cells were seeded into 6-well plates and grown to 70% confluence. The cells were then transiently transfected with 30 ng plasmid encoding Nluc-arrestin-3 and 3970 ng of a plasmid encoding the respective $Y_2R$-eYFP variant using MetafectenePro (Biontex, München, Germany). The large excess of the fluorescent acceptor enables saturation of the Nluc-donor. One day post-transfection, the cells were re-seeded into white and black poly-D-lysine coated 96-WP at a density of 150,000 cells/well in technical triplicate using phenol-red-free medium. The next day, the cells in black plates were used for quantification of receptor expression. Full medium was exchanged to BRET buffer (Hank's balanced salt solution (HBSS) containing 20 mM HEPES, pH 7.4), and the eYFP fluorescence was determined by direct excitation at 485 (20) nm and 544 (25) nm emission in a Tecan Spark reader (Tecan, Männedorf, Switzerland). The cells in white plates were loaded with 4.2 µM coelenterazine H (Nanolight/Prolume, Pinetop, AZ, USA) in BRET buffer for 5 min at 37 °C, and stimulated with the indicated concentration of NPY by adding a 4x concentrated peptide stock. After 10 min at 37 °C, the BRET signal was read out using a Spark plate reader (Tecan, Männedorf, Switzerland, filter settings: 400–470 nm (luminescence); 535–650 nm (fluorescence); well-wise mode). In the kinetic measurements, peptide addition was performed by a built-in injector system of the reader, and the signal was read continuously for the time indicated. BRET values were corrected by subtracting the buffer controls. The means of three independent experiments were pooled and analyzed using GraphPadPrism9 (San Diego, CA, USA). Concentration-response-curves were fit to a three-parameter-based agonist (log) vs. response regression model, the observed rate constants $k_{obs}$ of arrestin recruitment in the kinetic measurements by one-phase association fit.

## Inositol phosphate accumulation assay

Activity of $Y_2R$-eYFP variants in the canonical $G_i$ pathway can be measured by re-routing the cellular response to the phospholipase C pathway using a chimeric $G_{iq4\Delta myr}$ protein as described[26,103]. Briefly, HEK293 were co-transfected with the respective receptor construct and $G_{iq4\Delta myr}$-chimera using MetafectenePro (Biontex, München, Germany) at 70% cell confluence and subsequently re-seeded into 384-well plates at a density of 20,000 cells/well. Approximately 36 h post-transfection, the cells were stimulated with NPY dilutions in HBSS containing 20 mM LiCl for 60 min at 37 °C. Produced cellular inositol monophosphate was quantified using the IP-One Gq kit (Cisbio/PerkinElmer, Waltham, MA, USA) and the HTRF signal was measured in a Spark plate reader (Tecan, Männedorf, Switzerland, filter settings (ex/em): 320(25) / 620(10) nm (donor) and 320(25) / 665(8.5) nm (acceptor)). The signal was normalized to wild-type $Y_2R$ (minimum/maximum response). The normalized means of the independent experiments were pooled and fit to a three-parameter-based agonist (log) vs. response non-linear regression (GraphPad Prism 9, San Diego, CA, USA).

## $G_{i1}$ and $G_o$ activation assay by BRET

To measure $Y_2R$ G protein signaling employing the native $G_{i/o}$ pathways, we used the G-CASE sensors[100] (plasmids obtained from ADDGENE #168120 and #168123), which carry internal fusions of Nluc in the Gα subunit and cpYFP in a single-chain variant of Gβγ and enable real-time measurement of G protein activation by BRET. Upon activation of the G protein by the receptor, the high BRET state of the G protein heterotrimer is reduced by the separation of the subunits. To this end, HEK293 were seeded into 6 WP. At approx. 70% cell confluence, the cells were transfected with 1200 ng plasmid encoding the respective receptor variant and 800 ng plasmid encoding the respective G-CASE sensor (total DNA per well 2000 ng) using MetafectenePro (Biontex, München, Germany). The next day, the cells were re-seeded into solid white 96 WP coated with poly-D-lysine at a density of 150,000 cells/well in technical triplicate and were grown for additional 24 h. The next day, the cells were loaded with 4.2 µM coelenterazine H (Nanolight/Prolume, Pinetop, AZ, USA) in 150 µl BRET buffer for 5 min at 37 °C, and stimulated with 50 µl of the indicated concentration of NPY by adding a 4x concentrated peptide stock (final volume/well: 200 µl). After 15 min at 37 °C, the BRET signal was measured using a Spark plate reader (Tecan, Männedorf, Switzerland, filter settings: 400–470 nm (luminescence); 535–650 nm (fluorescence); well-wise mode). In the kinetic measurements, peptide addition was performed by a built-in injector system of the reader and the signal was read continuously for the time indicated. Changes in the BRET signal were normalized to the baseline with the addition of just BRET buffer, and the normalized means were fit to a three-parameter-based agonist (log) vs. response regression model for the concentration-response-curves. The observed rate constants ($k_{obs}$) of G protein activation in the kinetic measurements were determined by one-phase dissociation fit.

## NanoBRET ligand binding assay

Membrane preparations of transiently transfected HEK293 cells expressing Nluc-$Y_2R$-eYFP or N-terminal variants thereof were used to measure the $K_D$ of NPY binding as described recently[26]. Briefly, for each data point, membranes containing 0.03 µg total protein were suspended in ice-cold BRET binding buffer (Hank's balanced salt solution (HBSS) containing 20 mM HEPES (pH 7.4), 0.1% bovine serum albumin (BSA)). The assay was performed in solid black 96-WP with a total assay volume of 100 µL in technical duplicate or triplicate. A dilution series of K18-Tamra-NPY was prepared as 10x stock in $H_2O$ + 0.1% BSA, 10 µl was added to the prepared membranes (90 µl) and incubated for 10 min at room temperature with gentle agitation. 10 µL of coelenterazine H (42 µM) in HBSS buffer were added to each well, and

BRET was measured using a Spark plate reader (Tecan, Männedorf, Switzerland, well-wise mode, filter settings: 430–470 nm (luminescence); 550–700 nm (fluorescence), integration time 500 ms). To determine unspecific binding, 100 μM of the $Y_2R$ antagonist JNJ-31020028 (Selleck Europe) were added to each well. To measure $k_{off}$, membranes were pre-incubated with 300 nM K18-Tamra-NPY for 10 min (in 90 μl), Coelenterazine H was added to a final concentration of 4.2 μM (10 μl), and the baseline signal was read. To start the dissociation measurement, the antagonist JNJ-31020028 was added to a final concentration of 50 μM (20 μL) using the built-in injector system of the plate reader and the BRET decay was measured for 20 min. Raw BRET was calculated as the ratio of fluorescence to luminescence, and the background value of the unspecific binding controls was subtracted to obtain the netBRET signal. For the analysis of concentration-response curves, the netBRET means of the independent experiments were pooled and fit to a biphasic non-linear regression model; kinetic ligand dissociation data were fit from single experiments by a built-in one-phase dissociation model (GraphPad Prism 9, San Diego, CA, USA).

### Reporting summary
Further information on research design is available in the Nature Portfolio Reporting Summary linked to this article.

## Data availability
The XL-MS generated in this study have been deposited in the ProteomeXchange database under accession code PXD051865. MD-trajectories generated in this study are accessible under https://proteinformatics.uni-leipzig.de/mdsrv.html?load=file://_nt/y2r_nt.js. The structural data used in this study are available in the PDB database under accession codes 7DDZ, 7X9B, and 7YON. Further information and requests for resources and reagents should be directed to and will be fulfilled by Anette Kaiser and Andrea Sinz. Source data are provided with this paper.

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

## Acknowledgements

Dirk Tänzler and Marvin Bremer are acknowledged for performing initial cross-linking experiments. We thank Ronald Weichelt and Barbara Klüver for excellent technical assistance in the functional characterization of $Y_2R$ variants, and are grateful to Daniel Huster for discussion and advice with NMR experiments and careful proofreading. Funding: This work was funded by the Deutsche Forschungsgemeinschaft (DFG, German Research Foundation), project number 421152132, SFB 1423, subprojects A03 (P.S.), A04 (R.F.), A07 (J.M.), B03 (A.K., A.S.), C01 (P.W.H.), Z03 (A.B.S.), Z04 (J.M., P.W.H.). AS acknowledges financial support by the DFG (RTG 2467, project number 391498659 "Intrinsically Disordered Proteins-Molecular Principles, Cellular Functions, and Diseases"; RTG2751 "InCuPanC", project number 449501615; INST 271/404-1 FUGG; INST 271/405-1 FUGG; INST 271/528-1 FUGG), the Federal Ministry for Economic Affairs and Energy (BMWi, ZIM project KK5096401SK0), the region of Saxony-Anhalt, and the Martin Luther University Halle-Wittenberg (Center for Structural Mass Spectrometry). Supported by the Open Access Publishing Fund of Leipzig University.

## Author contributions

A.K., A.S., and P.W.H. designed the study and the experimental approaches. A.K., J.C.R., A.B., M.P., K.D.L., D.S., R.M., R.F., and P.S. carried out experiments. A.K., J.C.R., A.B., M.P., K.D.L., C.Ia., D.S., C.Ih., R.F., A.B-S., P.S., J.M., P.W.H., and A.S. contributed to data analysis. A.K., J.C.R., A.B., M.P., D.S., and P.W.H. prepared figures. A.K. and A.S. wrote the manuscript with input from all co-authors. All authors read and approved the final manuscript.

## Funding

## Competing interests

The authors declare no competing interests.
