## [Transparent Peer Review file · Nature Communications]

Transient ligand contacts of the intrinsically disordered N-terminus of neuropeptide Y₂ receptor regulate arrestin-3 recruitment

Corresponding Author: Dr Anette Kaiser

Version 0:

Reviewer comments:

Reviewer #1

(Remarks to the Author)

In this work, Kaiser et al have proposed that the intrinsically disordered N-terminus (NT) of the neuropeptide Y (NPY) receptor Y₂ (Y₂R) transiently interact with NPY and regulate arrestin-3 recruitment, using crosslinking-mass spectrometry (XL-MS), distance-based modeling, and molecular dynamics (MD) simulations, as well as Gi signaling assays and arrestin recruitment BRET assays.

Majors

While XL-MS data show clear transient interactions between Y₂R NT and NPY, nanoBRET assay does not support such interactions (i.e., NT is not largely involved in NPY binding) and thus NT does not influence Gi signaling through Y₂R. The discussion is not really convincing about the role of NT in terms of its interaction with NPY and biological functions.

The MD simulations are very short (only 50 ns for 10 replicas each). To be honest, it is not even clear why MD simulations are needed in this study. Distance-based models already provide enough snapshots and interaction patterns. If the goal of the MD simulations prove the distance-based models and show diverse interaction patterns between NT and NPY, the authors should run longer simulations (at least 1 microsecond for each replica). The author can also run MD simulations with ligands for an "open" conformation.

Minors

1. In Figure 2, (B) can be a separate figure. (A) is not needed to be so big. How the RMSD was measured? Aligned by the receptor TMs?
2. No citation for the protein force field.
3. Line 585: the CYSP path -> the CYSP patch; how CYSP was added?
4. Line 589: Does the published procedures correspond to using CHARMM-GUI Membrane Builder?
5. In Table 1: (1) how the lipid restraints were applied? (2) How these equilibration steps are different from the CHARMM-GUI Membrane Builder? (3) Does this table really need to be in the main?

Reviewer #2

(Remarks to the Author)

The current study combines experimental and computational approaches including cross-linking MS, structural modeling and MD simulation, and receptor pharmacological assays, to study the structure-function relationship of the intrinsically disordered Y₂R NT. The key finding proposed by this study is a transient binding mode of the ligand NPY to the Y₂R NT that seems to selectively affect arrestin-3 (i.e. beta-arrestin 2) binding rather than Gi coupling.

One of the major weaknesses is the difficulty to rationalize this functional selectivity with the structural information gained from modeling and simulation. The structural models highlight the frequent contacts of acidic clusters in NT with various residues in NPY, yet experimentally the authors found "mutating the acidic patches to neutral residues or reverting the negative charge to positively charged stretches still allowed for high-affinity NPY binding and activation of the canonical Gi pathway." While NPY binding and Gi coupling do not depend on the Y₂R NT, it was found that beta-arrestin 2 recruitment to

the receptor could be more strongly affected by the NT mutations. For this unusual signaling selectivity, the author mainly described the result yet did not provide convincing structural explanation. The N terminus of GPCR is known to play a key role in mediating the kinetics of ligand access into the pocket. A missing critical link here is how specific NT mutations affect the kinetics of ligand binding rather than only K_d determination at an equilibrium state.

It is also unclear what is the contribution of cross-linking MS data to modeling the conformational ensemble of the receptor NT. This study employed cross-linking MS and structural modeling and simulation to propose “a transient interaction network between two acidic clusters in the Y2R NT with the central helix of NPY”. But most of the frequent interactions in Figure 3A seem to be self-evident based on the sequences. In other words, as a comparison, if you model the complex using Rosetta or AF2 (without using the cross-linking data) and run MD with the same procedure, what kind of dynamics and contacts would you see for the Y2R NT and NPY?

In addition, many essential details and results of the experiments are missing, which makes the conclusions less convincing.

1. Identification of photo-crosslinking peptides and sites is not trivial and easy to get false IDs. In the SI Table, confidence scores for peptide and site assignment, together with PSM numbers should be provided for reported cross-linked peptides. MS/MS spectra for important cross-links should be also provided with standard annotations. The authors labeled certain cross-link IDs with “below LOD”, yet the criteria for this manual check is not described. What is more concerning is that almost identical cross-link sites were found from the full length receptor vs the NT peptide alone, which is counterintuitive as it suggests the major ligand-binding motifs (TMDs and ECL2) have no impact on NT interaction with the ligand. This would need some explanation. And the XL-MS experiment needs more controls (e.g. blocking NPY binding with an antagonist, or using a receptor mutant incapable of ligand binding) to validate the cross-link identifications.

Furthermore, the deposited MS data inventory (shown in the attachment) is quite confusing and inconsistent with the manuscript. While it was mentioned in Results that this work labeled NPY with diazirine-containing photo-leucines, the raw data files are named with both pLeu and pMet, suggesting photo-Leu and photo-Met were both used? The authors acquire MS data in both DDA and DIA modes to but how these two different modes were applied to which samples or conditions and how DDA/DIA data were processed separately are completely missing in Methods. The data inventory should be clearly annotated with sample and acquisition conditions to allow re-use and re-analysis by others.

2. For the statement “We identified several clusters of interactions, in which the diazirine substituted leucine analogs in NPY at positions L24 and L30 interact with multiple sites in the Y2R NT”, it is not true to describe cross-links as interactions or contacts as they just indicate spatial proximity, not physical interactions.

3. The procedures for Rosetta modeling and MD simulation are too simplified. Figure S2 is meaningless with no details. Key steps for integrating cross-linking data into modeling should be described in Results, and a detailed workflow can be provided in SI. Other critical results about how many models generated, how to validate them, RMSD between models to the original structure, and internal variations of these models... all need to be described. For the MD part, how to select models as the starting point is also missing. The whole section of modeling and MD is difficult to follow and evaluate due to lack of essential details and clarity. Figure 3C is hard to read, it would be better to change the color code and show MD results in a format similar to high-quality publications.

4. “a turn-structure in the structurally well-resolved membrane-proximal part of the NT” can be rephrased, as it is a modeling result, not a true structure.

5. For receptor expression measurement, fluorescence microscopy is not quantitative or accurate. The standard flow cytometry can be done to measure receptor surface expression. Typically deleting NT would impair receptor trafficking and cell surface expression, yet the authors did not see such as an effect on the NT deletion mutant, please give some explanation.

6. Figure 4: K_d values for the low affinity state are missing. Also need SI tables to show K_d values and statistics from replicates. Here it is critical to assess how different mutants affect ligand binding kinetics (k_{on} , k_{off} rates) relative to wild-type.

7. For the beta arrestin 2 BRET assays, the E_{max} measurement heavily relies on receptor expression. So the expression levels should be first adjusted for all mutants to be comparable with the wild-type. Also need an SI table to summarize EC_{50} and E_{max} data from replicates.

8. For the NMR experiment, no methods are provided, and data presentation is incomplete. In addition to the delta ppm plot, the original spectra with annotation should be provided as a convention. The authors mentioned in Fig. S1 that the overall binding pattern of NPY to Y2R NT is different from the full-length Y2R-bound conformation, which is in fact contradictory to the cross-linking MS result (the same set of cross-links identified for NT and full-length).

9. In Discussion, “the NT kinetically limits ligand access and the on-rate is accelerated upon electrostatic perturbation” is not supported by data, need ligand kinetics measurement. Arrestin recruitment kinetics does not directly reflect ligand access to the binding pocket.

10. As structural models indicates a very high contact frequency between D42NT and K304ECL3, it would be informative to show the functional effect of mutating K304.

11. As for the full-length Y2R protein, its construct and expression condition should be briefly mentioned in Results. In this work, a cysteine-deficient Y2R was recombinantly expressed in *E. coli* as inclusion bodies, which is not a conventional procedure for GPCR expression. Cysteine deficiency will remove disulfides important for receptor stabilization, expression in bacteria rather than in insect cells will likely deprive the receptor of important PTMs esp. in N terminus. Could the authors explain why not expressing the receptor in insect cells as described in previous structural work? Also SEC data and SDS-PAGE data need to be shown to confirm the protein purity and quality.

12. For many critical experiments, only citing a reference in Methods is not enough. Specific details should be provided such as sample amount, concentration, buffer conditions, etc. How DDA and DIA data were processed with Bruker software and other packages (MeroX, proXL, and Skyline) should be described in sufficient details rather than citing a reference.

13. XL-MS experimental replicate information is missing in Methods.

Reviewer #3

(Remarks to the Author)

The manuscript by Kaiser et al. focuses on the role of N-terminus (NT) of the NPY Y2 receptor (Y2R) in ligand binding and signaling. Previous study from Dr. Beck-Sickinger's group, who is a co-author on the present manuscript, indicated that deletion of the Y2R-NT decreases ligand binding, signaling and internalization (PMID: 18845246). Here, the authors extend these observations using cross-linking mass spectrometry and computational modeling to identify specific areas of Y2R-NT interactions with NPY, which is a novel aspect of the study. Based on the mutational studies, they also conclude that NT modulates arrestin-3 binding to Y2R, but it does not affect ligand binding or G-protein activation. These findings raise some important issues that need to be addressed:

1) The lack of the effect on ligand binding and G protein signaling seems to be in conflict with previous studies. This was not addressed in the discussion. In Figure 4A, NT deletion or D42A-S43A-T44A mutations lead to 6- and 18-fold decrease in G-protein activity (EC50). Why is this called "a minor contribution"? Was any statistical analysis done to determine how consistent this effect was?

2) The above conclusions were made based on the artificial system utilizing chimeric Giq4Δmyr to measure G protein signaling. Showing the effect on the endogenous NPY receptor signaling pathways (e.g. cAMP levels upon forskolin stimulation) would strengthen and validate the above data.

3) The conclusion about the effect of the NT deletion on arrestin-3 binding were made solely based on BRET. This data should be confirmed using additional methods, detecting interactions with endogenous arrestin-3. Do the mutations introduced to the NT affect the receptor internalization?

4) Previous study (PMID: 18845246) suggested that the amino acid composition of the Y2R-NT does not affect its function. Here, the authors identified two acidic patches within NT that are responsible for binding NPY. Yet, their mutation to neutral asparagine or glutamine residues had no effect on arrestin-3 recruitment. Doesn't this mean that these acidic amino acids are not really essential?

5) Potential mechanisms by which changes in extracellular NT affect arrestin-3 binding without changes to the ligand binding should be discussed.

6) The authors mention using different algorithms for identifying intrinsically disordered regions of the Y2R. Presenting the results of at least one such analysis would be helpful.

Altogether, the presented findings are novel. However, it is essential to perform functional validation (signaling, internalization) based on the endogenous signaling pathways. Moreover, better discussion and data interpretation in the context of previous studies is required.

Version 1:

Reviewer comments:

Reviewer #2

(Remarks to the Author)

The authors have made great efforts to address critical comments raised by the referee, which have been mostly resolved. Only two points remained for further clarification:

(1) Since DIA-PASEF data acquisition is not widely adopted in XL-MS experiments and needs a complex data processing procedure, the criteria used for generating EICs from raw DIA-PASEF data in Skyline based on a reference library can be specified in Methods; (2) the distance restraints for specific cross-linked residue pairs used in computational modeling of Y2R's N-terminal in contact with NPY can be also provided as SI data.

Reviewer #3

(Remarks to the Author)

Thank you for the thorough revision of the manuscript. My previous concerns have been adequately addressed. However, there are some minor points that still need to be addressed:

1) Fig. 2B – description of the light blue regions of the receptor is missing (ECLs?).

- 2) Fig. 4E – are the differences in the times of Gi1 activation statistically significant?
- 3) Fig. S8 – the differences in the Y2R internalization should be quantified.
- 4) Line 278 – grammatical error – please correct.

Reviewer #4

(Remarks to the Author)

I was asked to review this in place of Reviewer 1. I can see that the authors have tried to address what reviewer 1 asked for with longer MD simulations. However, I think they missed the point that reviewer 1 was really asking about in terms of what the point of the MD simulations are actually for. The focus of the analysis and the conclusions are on the flexibility of the N-terminus, but this is fairly uninformative because this is already a) expected and b) implicitly present in the cross-linking (and NMR) data and the resultant constraint-derived models, as indeed succinctly encapsulated in Fig. S6.

What I was expecting to see from the extended MD simulations was an analysis of the contact times and nature of these. The authors refer to “contact rewiring” (an odd phrase) which presumably means that some contacts are not stable but its not clear which ones are stable? And do the experimentally derived cross-link distance fade over time or are they genuinely transient – ie. do they form/re-form within the 1 μ S MD simulations. Finally, its not clear to me how the interactions here lead to modulation of arrestin-3 effects? Most of the dramatic effects appear to come from the D42-S43-T44 region which is already ordered rather than from the remaining more N-terminal sections. I thus tend to think this connection to the transient nature of the interactions remains a bit speculative overall.

Some other minor points:

1. The authors should rationalize why L24_NPY to E16_NT was not seen in the complex.
2. On page 8, they say they “devised” the algorithms. I think the authors mean “employed”
3. Table S1 is very difficult to interpret for non-experts. It could use some annotation.
4. “Interestingly, within the Y2R NT, the two acidic patches E15–E20NT and D35–E39NT remain at a distance ≥ 7 Å most of the time and never approach closer than 5.3 Å to each other.” That two acidic patches don’t come close together is hardly surprising.

Version 2:

Reviewer comments:

Reviewer #2

(Remarks to the Author)

All previous comments have been well resolved by the authors, and the reviewer would recommend publication of the revised version.

Reviewer #3

(Remarks to the Author)

Thank you once again for making the changes in the manuscript. Here are some remaining comments:

1) Fig. 2B – description of the light blue regions of the receptor is missing (ECLs?).

Response of the authors:

We apologize for the confusion. The color code in panel B is the same as in panel A, with orange color for the more distal residues and pink color for the membrane- proximal regions. We have updated Figure 2 and the figure legend accordingly.

New comment:

I made a mistake in my previous review. I was referring to the light blue boxes in Figure 1B, not 2B. Please add the description along with other regions. My deepest apologies for the confusion.

3) Fig. S8 (currently S10B)– the differences in the Y2R internalization should be quantified.

Response of the authors:

Changes in the text:

In agreement with the BRET results, the Δ 2-41 and D42A-S43A-T44A variants, which show the weakest and lowest potency arrestin-3 recruitment, display delayed internalization with 58% (Δ 2-41) and 106% (D42A-S43A-T44A) of receptors, respectively, still residing in the plasma membrane after 10 min stimulation with 100 nM NPY, compared to 32% for wild-type Y2R (Fig. S10B).

New comment:

Thank you for the quantification. However, the above changes in the text do not accurately describe the new results. While the D42A-S43A-T44A variant indeed shows significant changes in internalization, the delay in the internalization of the $\Delta 2-41$ variant is not statistically significant as compared to the control. Moreover, the internalization rate of the $\Delta 2-41$ form is not different from other mutants that did not show significant changes in arrestin recruitment. How do authors reconcile these discrepancies?

Reviewer #4

(Remarks to the Author)

The authors have addressed my concerns satisfactorily. Happy to publish.

NCOMMS-24-44402

Transient ligand contacts of the intrinsically disordered N-terminus of neuropeptide Y₂ receptor regulate arrestin-3 recruitment

Response to Reviewer's comments

We are grateful to the Reviewers for the insightful comments on our study. We have used the constructive feedback and amended our manuscript according to the suggestions made.

In the following, we are addressing the Reviewer's comments point-by-point. We quote statements from the Reviewer in **bold face**. We refer to changes made in the manuscript in *yellow italics*.

Reviewer #1 (Remarks to the Author):

In this work, Kaiser et al have proposed that the intrinsically disordered N-terminus (NT) of the neuropeptide Y (NPY) receptor Y₂ (Y₂R) transiently interact with NPY and regulate arrestin-3 recruitment, using crosslinking-mass spectrometry (XL-MS), distance-based modeling, and molecular dynamics (MD) simulations, as well as G_i signaling assays and arrestin recruitment BRET assays.

Majors

While XL-MS data show clear transient interactions between Y₂R NT and NPY, nanoBRET assay does not support such interactions (i.e., NT is not largely involved in NPY binding) and thus NT does not influence G_i signaling through Y₂R. The discussion is not really convincing about the role of NT in terms of its interaction with NPY and biological functions.

We thank the reviewer for allowing us to clarify this point. We agree that the transient contacts between NPY and the Y₂R NT hardly affect the high-affinity state of the receptor in nanoBRET binding experiments or G_i signaling. This can be directly explained from the recruitment and activation of G proteins at the Y₂R.

The high-affinity state in NanoBRET binding experiments reflects the nucleotide-free ternary complex NPY-Y₂R-G_{i1}, while the low-affinity state is G protein-independent (Tang T. et al. Sci Adv. 2022;8(18):eabm1232). This means that G-protein-coupling and nucleotide release induce a large allosteric enhancement to the affinity of the peptide ligand, increasing its affinity by over 1000-fold from ~300 nM to about 0.15 nM (see also Fig. 5A of the current manuscript). This is probably mediated by a strong contraction of TM2 and TM6 compared to the antagonist-bound state (Tang T et al. Nat Commun. 2021;12(1):737), enabling tight interactions of many NPY residues with the Y₂R TM pocket and ECL2. Under these conditions, the extra gain in free energy from interactions with Y₂R NT is comparably small and the effect of point mutations is therefore not readily visible in the functional assays that report on the high-affinity state of Y₂R (high-affinity binding constant in NanoBRET binding and G protein activation). We did, however, measure reduced affinity in NanoBRET binding assays in the G-protein-independent low-affinity state when either large portions of the NT were deleted (Δ 2-41) or the Y₂R was

disconnected from the TM domain by the D42A-S43A-T44A mutation, underlining a functional contribution (novel Table S3).

In contrast, the functional contribution of the N-terminal contacts are more clearly visible with respect to arrestin recruitment. The allosteric enhancement of arrestin towards the ligand binding pocket is rather small, as reflected by the mid-nanomolar potency in arrestin-3 recruitment assays. Therefore, loss of transient contacts between Y₂R NT and NPY has larger effects. Moreover, the entire process of G protein dissociation, recruitment of GPCR-kinases, receptor phosphorylation and finally recruitment of arrestin-3 is rather slow and includes multiple steps. We have now included direct measurement of the kinetic ligand dissociation (k_{off}), which is inversely related to ligand residence time and has been shown to be important for arrestin recruitment. Indeed, k_{off} was accelerated for both, the $\Delta 2$ -41 and the charge reversal variants of the acidic patches, providing a basis for the reduced complex formation with arrestin-3. We have now included the k_{off} measurements and have updated Figures 4 and 5. Moreover, we significantly extended the discussion to make these points clear.

- Changes in the text:

Novel Figure 4:

Figure 4: Y₂R NT mutations differentially modulate G-protein activation and recruitment of arrestin-3 variants. A) Close-up view to membrane proximal Y₂R NT from cryo-EM (PDB 7X9B) highlighting interactions of D42-S43-T44^{NT}. B) Live cell fluorescence microscopy shows that Y₂R NT variants are

transported to the plasma membrane like the wild-type. Y_2R -eYFP variants are shown in yellow, cell nuclei are stained by H33342 and shown in blue, scale bar equals 10 μ m. All pictures were acquired with identical light exposure and picture processing. C) Quantification of cellular receptor expression is based on eYFP fluorescence in a plate reader; shown are the means and individual values of five independent experiments relative to wild-type Y_2R . D) Activity of Y_2R variants towards G_{i1} proteins as measured by a direct BRET assay (G_{i1} -CASE) 10 min after ligand stimulation. All variants show full activation, D42A-S43A-T44A has the strongest shift in potency, while all others show mild or no effects. E) Kinetics of G_{i1} activation after stimulation with 1 nM NPY is overall similar for most variants except Δ 2-41 and D42A-S43A-T44A, which have a tendency for faster apparent rate constants (k_{obs}). 95% CI of wild-type Y_2R k_{obs} is given as grey rectangle for comparison. F) Recruitment of arrestin-3 to Y_2R variants as measured by BRET 10 min after NPY stimulation. D42A-S43A-T44A and Δ 2-41 variants reduce recruitment of arrestin-3 to the receptor. Neutral (in green) and charge-inverted (in blue) variants of the acidic patch in Y_2R NT lead to distinct behavior, with only charge reversal impairing arrestin-3 recruitment. G) Kinetic analysis of arrestin-3 recruitment after stimulation with 100 nM/ 1 μ M NPY. Charge reversal in acidic patches 1 and 2 leads to faster initial recruitment, but also early signal decay. Bar plot shows quantification of the initial rate of arrestin-3 recruitment to receptor variants. Apparent rate constants (k_{obs}) are plotted in comparison to wild-type Y_2R -eYFP and its 95% confidence interval (grey rectangle). * $p < 0.05$, ** $p < 0.01$ in one-way ANOVA with Dunnett's post-hoc test. For color legend, please see panel E. Data in C–G are the mean \pm SEM of $n \geq 3$ independent experiments each performed in technical triplicate.

Novel Figure 5:

Figure 5: Ligand binding at Y_2R NT variants. A) Equilibrium binding between K18-Tamra-NPY and Nluc- Y_2R -eYFP variants. Charge reversal in the acidic patches has wild-type-like equilibrium binding properties, while D42A-S43A-T44A displays a significantly smaller BRET window and further reduced low-affinity binding compared to wild-type Y_2R . B) N-terminal deletion in Δ 2–41 Y_2R increases overall BRET window as expected from the reduced distance, and the low-affinity state has about two-fold reduced affinity compared to wild-type Y_2R . Please note that Y-axis scaling is different between panels A and B. C) Ligand dissociation from the low-affinity state (bound to 300 nM K18-Tamra-NPY; re-binding blocked by 50 μ M antagonist) is faster for Δ 2-41 and acidic patch charge reversal mutants. Data are the mean \pm SEM of $n=3$ (A, B) or $n=4$ (C) independent experiments, performed in technical duplicate or triplicate. Related to Fig. S9: Cellular localization and activity of the Nluc- Y_2R -eYFP construct used for NanoBRET binding.

Discussion:

p.20, l 357 ff.

Mutating the acidic patches around E15–E20^{NT} and D35–E39^{NT} to neutral asparagine or glutamine residues or reverting the negative charge to positively charged lysine stretches still allowed for high-affinity NPY binding and activation of the canonical G_i pathway. [...]

The lack of significant effects of Y₂R NT on high affinity binding and G protein activation can be explained by a very strong allosteric enhancement of ligand affinity by G_{i/o} proteins²⁶. In absence of G_i, the affinity of NPY is rather low with ~300 nM, but the NPY-Y₂R-G_i ternary complex displays a high-affinity interaction with sub-nanomolar K_D, which is also reflected in the high functional affinity in G protein activation assays. This strong gain in affinity is probably mediated by a contraction of TM2 and TM6 compared to the inactive Y₂R bound to an antagonist (pdb 7DDZ, ref²⁵), and enables high affinity interactions of ligand and receptor in the transmembrane pocket. Accordingly, minor changes in the extracellular contacts have negligible impact on the affinity of the overall G-protein-bound complex.

In contrast to the mild effects of G protein activation, the Y₂R NT strongly modulates the recruitment of arrestin-3 to the receptor. This interaction occurs with significantly lower potency of around 50 nM, and hence, does not seem to be allosterically stabilized by the intracellular partner and changes in the extracellular interactions should become more visible. This is consistent with the observed affinity reduction of the G-protein-independent low affinity state by two- to three-fold when deleting the flexible part of Y₂R's NT in the Δ2-41 variant or dislodging the entire NT by D42A-S43A-T44A mutation in the membrane proximal region (see below) (Fig. 5A,B, Tab S3). Moreover, also the potency to recruit arrestin-3 is further reduced for these variants compared to the wild-type Y₂R (Fig. 4D,F, Tab S2).

p.21 l.384 ff.

In contrast, substitution to positive charges in the Y₂R E15K–E20K^{NT} and Y₂R D35K–E39K^{NT} variants show a strongly reduced recruitment of arrestin-3. This apparently originates from reduced stability of the complex as reflected by a BRET decay already after 4–5 min at a sub-saturating ligand concentration of 100 nM, while the wild-type Y₂R-arrestin-3 complex remains stable for ≥ 15 min under the experimental conditions. This was underlined by increased k_{off} rates and hence reduced ligand residence time in ligand binding experiments for the charge reversal variants of the acidic patches and the Δ2-41 deletion even though we could not detect significant changes in equilibrium binding for the E15–E20K and D/E35–E39K variants (Fig. 5, Tab S3). We speculate that an increased k_{on} rate of the ligand at least partly compensates the increased off-rate, such that equilibrium binding is only minimally affected. This is supported by the observed increased apparent rates (k_{obs}) of arrestin-3 recruitment and G_{i1} activation for the Δ2-41 deletion, and partly for the charge reversal mutants (Fig. 4E,G, Tab S2).

The MD simulations are very short (only 50 ns for 10 replicas each). To be honest, it is not even clear why MD simulations are needed in this study. Distance-based models already provide enough snapshots and interaction patterns. If the goal of the MD simulations prove the distance-based models and show diverse interaction patterns between NT and NPY, the authors should run longer simulations (at least 1 microsecond for each replica). The author can also run MD simulations with ligands for an “open” conformation.

We conducted MD-simulations to confirm the distance-based models and monitor the nanosecond dynamics of the NPY-Y₂R complex. As suggested by the reviewer, we have now elongated these simulations to 1 microsecond per replica. Over the entire course of the simulations, contacts between NPY and Y₂R NT are present, and show a similar interaction pattern and frequent contact re-wiring as in the short simulations. Interestingly, however, during the longer simulation runs, the contact frequency to acidic patch 1 (E15-E16-E20) is somewhat reduced, while the contact frequency to acidic patch 2 (D35-E37-E39) is increased, which matches the experimental observation that mutation of acidic patch 2 is functionally even slightly more detrimental. We have included these longer simulations in the main text and have updated Figures 2 and 3.

- Changes in the text:

Figure 2:

Figure 2: Structural flexibility of the Y₂R NT in microsecond MD simulations. **A:** Violin plot of the backbone-RMSD values for each residue of the NT during the MD simulations, relative to the initial frame, grouped in sets of five. **B:** Example conformations sampled during MD simulations. Residue coloring matches the groups in the violin plot, with opacity increasing over simulation time.

Figure 3:

Figure 3: Residue Contacts between Y₂R NT and NPY. A: Flare plot showing the contact frequency between residues of Y₂R NT and NPY from nanosecond MD simulations. Contacts are defined as residues

within a distance of 4.5 Å (excluding direct neighbors) and are represented by black lines. Line opacity indicates contact frequencies with higher opacity reflecting more frequent contacts. The acidic patches E15-E20 and D35-E39 of the NT are highlighted. **B**: Visualization of overall contact frequency between residues of NPY and Y₂R NT. Contact frequencies are shown by red hue intensity, with higher frequencies indicated by darker shades. Residues with high contact frequencies are labeled for clarity. **C**: Distance plots for selected residue pairs M17^{Y₂R}-R25^{NPY}, D35^{Y₂R}-R25^{NPY}, and L40^{Y₂R}-R19^{NPY}. Different colors represent different simulation runs (run 1: blue to run 10: green). Corresponding snapshots at 0.4 μs are shown below the plots using the same color scheme.

p.9., l. 170 ff.

Given the flexibility of the NT, we performed 10 independent simulations of 1 μs each to sample the conformational space of the modeled Y₂R NT.

p.10, l. 192 ff.

In summary, the MD simulations reveal the disordered character of the Y₂R NT. Moreover, within the limitations of the relatively short simulation times, they suggest that the huge conformational space adopted by the NT is restricted by electrostatic forces, specifically by some frequently observed interactions between the flexible NT and the peptide ligand.

p.19, l.343 ff.

Overall, despite some contact re-wiring during the 1 μs long MD simulations, the experimentally derived cross-links remain enriched during the simulations, strongly supporting specific interactions between Y₂R NT and the central helix of NPY.

Minors

1. In Figure 2, (B) can be a separate figure. (A) is not needed to be so big. How the RMSD was measured? Aligned by the receptor TMs?

We have updated Figure 2 and made panel A smaller. For the RMSD measurement, the backbone of the receptor TM regions was aligned to the initial frame of the simulation, following equilibration. The RMSD of the NT backbone was then calculated using VMD, with the NT of the initial simulation frame as reference.

- Changes in the text:

Legend to Figure 2:

A: Violin plot of the backbone-RMSD values for each residue of the NT during the MD simulations, relative to the initial frame, grouped in sets of five.

2. No citation for the protein force field.

We have included citations of the CHARMM36m force fields in the methods section.

Huang, J., Rauscher, S., Nawrocki, G. *et al.* CHARMM36m: an improved force field for folded and intrinsically disordered proteins. *Nat Methods* **14**, 71–73 (2017). => (now ref. 87)

- Changes in the text:

p.30, l.633 ff.

The amino acids were kept in their standard protonation state of the CHARMM36m force field⁸⁷ ...

3. Line 585: the CYSP path -> the CYSP patch; how CYSP was added?

4. Line 589: Does the published procedures correspond to using CHARMM-GUI Membrane Builder?

The CYSP patch was added during setup creation utilizing the CHARMM-GUI Membrane Builder.

- Changes in the text:

p.30, l.641 ff.

Finally, during the setup using the CHARMM-GUI Membrane builder⁸⁵, C8.60 was palmitoylated using the CYSP patch option.

5. In Table 1: (1) how the lipid restraints were applied? (2) How these equilibration steps are different from the CHARMM-GUI Membrane Builder? (3) Does this table really need to be in the main?

As suggested, we have shifted table 1 with the equilibration steps of the MD into the supplementary material as novel Table S5. The equilibration protocol follows a similar approach to that provided by the CHARMM-GUI Membrane Builder, with the exception that equilibration step 7 was extended from 200 ns to 500 ns. This modification was made to facilitate a smoother transition from the model based on the Rosetta force field to the CHARMM36m force field.

- Changes in the text:

p.30, l.645 ff.

The system was equilibrated in seven steps with descending restraints of protein side chains, backbone and on the POPC molecules, similar to the default CHARMM-GUI Membrane builder approach (Table S5). The last equilibration step, however, was elongated from 200 ns to 500 ns to facilitate a smoother transition between the force field used for modeling as well as refinement and the CHARMM36m force field used for the MD simulations⁹⁷. A total of 10 independent MD simulation were run for 1 μ s each.

Reviewer #2 (Remarks to the Author):

The current study combines experimental and computational approaches including cross-linking MS, structural modeling and MD simulation, and receptor pharmacological assays, to study the structure-function relationship of the intrinsically disordered Y2R NT. The key finding proposed by this study is a transient binding mode of the ligand NPY to the Y2R NT that seems to selectively affect arrestin-3 (i.e. beta-arrestin 2) binding rather than Gi coupling.

One of the major weaknesses is the difficulty to rationalize this functional selectivity with the structural information gained from modeling and simulation. The structural models highlight the frequent contacts of acidic clusters in NT with various residues in NPY, yet experimentally the authors found “mutating the acidic patches to neutral residues or reverting the negative charge to positively charged stretches still allowed for high-affinity NPY binding and activation of the canonical Gi pathway.” While NPY binding and Gi coupling do not depend on the Y2R NT, it was found that beta-arrestin 2 recruitment to the receptor could be more strongly affected by the NT mutations. For this unusual signaling selectivity, the author mainly described the result yet did not provide convincing structural explanation. The N terminus of GPCR is known to play a key role in mediating the kinetics of ligand access into the pocket. A missing critical link here is how specific NT mutations affect the kinetics of ligand binding rather than only Kd determination at an equilibrium state.

We thank the reviewer for this insightful comment. Indeed, the differential effects of the transient contacts of the Y₂R NT to NPY upon G proteins and arrestins are closely related to the different recruitment/activation mechanisms of G proteins and arrestins to this receptor.

Since Reviewer 1 raised a very similar concern (Majors, first section), we would like to refer to our response above.

It is also unclear what is the contribution of cross-linking MS data to modeling the conformational ensemble of the receptor NT. This study employed cross-linking MS and structural modeling and simulation to propose “a transient interaction network between two acidic clusters in the Y2R NT with the central helix of NPY”. But most of the frequent interactions in Figure 3A seem to be self-evident based on the sequences. In other words, as a comparison, if you model the complex using Rosetta or AF2 (without using the cross-linking data) and run MD with the same procedure, what kind of dynamics and contacts would you see for the Y2R NT and NPY?

We thank the reviewer for allowing us to clarify this point. XL-MS is one of the few techniques that is able to experimentally capture transient ligand interactions and was therefore of invaluable importance for the present study. The interactions captured by XL-MS do not show direct amino acid contacts, but rather spatial proximity in the range of C_α-

C α distances around 13 Å. XL-MS using diazirine chemistry with short reaction times in the micro- to millisecond time range yields important insights into highly dynamic protein states. These cross-links guide subsequent computational modeling experiments, such as Rosetta modelling. Accordingly, the Rosetta models provide another layer of evidence, showing that the experimentally mapped patterns of proximity are geometrically indeed feasible and - within the limitations of the used scoring functions - energetically favorable in the context of the native structure.

The MD-simulations, a method that provides the time-resolved motion of proteins at atomic resolution, link structural “snapshots” as those provided by AF2 and Rosetta in space *and* time. MD tracks conformational changes by monitoring not only local minima (“snapshots”), but also the connecting short lived and in IDPs frequent transitions. MD is a physics-based method that reliably calculates potential interaction energies, in the time-steps of 2 picoseconds, here between about 100.000 atoms. This comes with an exceptionally high computational cost: To reach the 1 microsecond timescale, 5×10^8 steps, with appr. 4×10^7 pairwise interactions per step, so about 2×10^{16} calculations in total is required, in each of the ten microsecond MD simulations. To sample the complete conformational space here, several order of magnitudes longer timescales would be required. Recently, we have used an adaptive sampling strategy to predict the relevant conformational space of a flexible receptor C-terminus, based on contact-distance restraints provided by smFRET data (Heng J et al. Nat Commun. 2023;14(1):2005.). Here, we sampled the relevant conformational space around the interactions mapped by the XL-MS, represented by the AF2 and Rosetta models.

A more complete sampling of the conformational spaces as suggested by the reviewer, despite its attractiveness, is limited by computational resources and beyond the scope of the current manuscript, which is mainly driven by the XL-MS approach that provides the necessary restraints to model the conformational space of the highly flexible N-terminus. Similarly, Rosetta relies on a stochastic Monte Carlo algorithm, which is computationally expensive. To reduce sampling time and constrain the structural search space, we adopted a strategy that involves starting from a complex structure predicted by AlphaFold2 and guiding the search using experimental constraints. This approach was necessary because AlphaFold2 alone failed to accurately predict the interactions between NPY and the N-terminal (NT) region of Y₂R (see Fig. R1 below). While the transmembrane pocket and its interaction with NPY closely resemble the experimentally determined structure, the Y₂R NT appears elongated with very low pLDDT values. This outcome is not surprising, as AlphaFold2 was trained primarily on structured proteins, aiming to predict a single conformation rather than a structural ensemble, which would be more suitable for intrinsically disordered regions (IDRs) (Ruff KM, Pappu RV. J Mol Biol. 2021;433(20):167208).

Indeed, although AlphaFold2-multimer has demonstrated a moderate ability to generate plausible structures adopted by IDRs upon binding, its predictive quality declines for heterogeneous fuzzy interactions, often resulting in random atomic coordinates (Omidi A et al. Proc Natl Acad Sci U S A. 2024;121(44):e2406407121). Nevertheless, the predicted structures provide a valuable starting point for generating an ensemble consistent with experimental distance constraints using physics-based methods (Brotzakis ZF et al. Nat Commun. 2025;16(1):1632).

Figure R1: A) Model of full length Y₂R in complex with NPY generated by AlphaFold-multimer v3 without any experimental constraints (pastel color; representative models from 3 different seeds), overlaid with the cryo-EM structure of NPY bound to Y₂R (grey, including Gai1β1γ2 heterotrimer). While the positioning of the ligand in the TM binding pocket is very accurate, the Y₂R NT remains in an extended conformation with very low pLDDT scores. B) Model of full length Y₂R as deposited in the AlphaFold Structure database (AF-P49146-F1-v4) and colored by pLDDT score. While the TM region is predicted with very high confidence (very high pLDDT, dark blue), the N-terminus cannot be confidently predicted and remains in an elongated conformation (very low pLDDT, orange).

In addition, many essential details and results of the experiments are missing, which makes the conclusions less convincing.

1. Identification of photo-crosslinking peptides and sites is not trivial and easy to get false IDs. In the SI Table, confidence scores for peptide and site assignment, together with PSM numbers should be provided for reported cross-linked peptides. MSMS spectra for important cross-links should be also provided with standard annotations. The authors labeled certain cross-link IDs with “below LOD”, yet the criteria for this manual check is not described.

We agree with the reviewer that providing raw data and accurate methodological descriptions is important for the scientific community. We therefore deposited the MeroX

results on the panorama server including confidence scores for peptide and site assignment, together with PSM numbers.

Moreover, we now provide the most important MS/MS data in a novel Figure S3, while an overview of all contacts can be found in Table S1.

We set our criteria for our very stringent manual checking as follows: In order to avoid a false assignment of cross-links, we excluded all cross-linked peptides, for which an unambiguous identification was not possible due to low-confident detection of either the precursor or fragment ions. These signals were manually excluded ensuring that only high-quality XL-MS data are considered for building our models. Thus, we replaced “below LOD” to “undetected” in Table S1.

- Changes in the text:

p.5 l.199 ff.

Raw data of annotated mass spectra are found in ProteomeXchange with identifier PXD051865 and select MS/MS spectra of the cross-linked fragments are presented in Figure S3.

Legend to Table S1:

** LOD = Limit of detection*

Novel Figure S3A-H:

A File: D:\Skyline..._allAAsIcombined_allAAs_bicelle Score: 106
 Scan: File:"tims..._1/K0=0.918, #10385-10492" Peptide α [VEQYGQTTTR] (E2)
 theor. Mass (M + H⁺): 2030.993 (>AGSINZI, from 19 to 30)
 Precursor Mass (M + H⁺): 2030.996 Peptide β [YYSAIR] (I5)
 Deviation 1.34 ppm (>AGSINZI, from 20 to 26)
 m/z: 677.6701 Crosslinker: Photo-Leu-PTM
 Charge: +3

B File: D:\Skyline..._allAAsIcombined_allAAs_bicelle Score: 69
 Scan: File:"tims..._1/K0=1.053, #14386-14582" Peptide α [DPEPELII] (E3)
 theor. Mass (M + H⁺): 1567.764 (>AGSINZI, from 35 to 42)
 Precursor Mass (M + H⁺): 1567.768 Peptide β [YYSAIR] (I5)
 Deviation 2.5 ppm (>AGSINZI, from 20 to 26)
 m/z: 784.38753 Crosslinker: Photo-Leu-PTM
 Charge: +2

C

File: D:\Skyline..._allAAstcombined_allAAst_bicelle
 Scan: File "fms..._1f0-0-995_#11116-11125"
 theor. Mass (M + H+): 2288.179
 Precursor Mass (M + H+): 2288.179
 Deviation: 0.36 ppm
 m/z: 753.35766
 Charge: -3

Score: 81
 Peptide: α [VEQYGGQTTPR] (E2)
 (+AGSINZ), from 15 to 30
 Peptide: β [HYINILTR] (S)
 (+AGSINZ), from 26 to 34
 Crosslinker: Photo-Leu-PTM

D

File: D:\Skyline..._allAAstcombined_allAAst_bicelle
 Scan: File "fms..._1f0-0-122_#9264-9466"
 theor. Mass (M + H+): 1846.893
 Precursor Mass (M + H+): 1846.9
 Deviation: 3.82 ppm
 m/z: 923.95366
 Charge: +2

Score: 71
 Peptide: α [HYINILTR] (S)
 (+AGSINZ), from 26 to 34
 Peptide: β [DENGIVTE] (E2)
 (+AGSINZ), from 9 to 16
 Crosslinker: Photo-Leu-PTM

E

File: D:\Skyline...e-Y2R-NPY_XL_DDA_01_RD2_1_41 Score: 61
 Scan: File "fims..._1k0=1.043_#9737-9747" Peptide α [DENQTV] (E7)
 theor. Mass (M + H⁺): 1589.708 (>AGSINZI, from 9 to 16)
 Precursor Mass (M + H⁺): 1589.704 Peptide β [YYSAIR] (I5)
 Deviation -2.42 ppm (>AGSINZI, from 20 to 26)
 m/z: 795.3562 Crosslinker: Photo-Leu-PTM
 Charge: +2

**F**

File: D:\Skyline...ermicombined_DE_C-term_bicelle Score: 51
 Scan: File "fims..._1k0=0.750_#10733-10756" Peptide α [YYSAIR] (I5)
 theor. Mass (M + H⁺): 1500.712 (>AGSINZI, from 20 to 26)
 Precursor Mass (M + H⁺): 1500.709 Peptide β [WPGEER] (E4)
 Deviation -1.93 ppm (>AGSINZI, from 207 to 213)
 m/z: 500.90781 Crosslinker: Photo-Leu-PTM
 Charge: +3

G File: D:\Skyline ... allMskcombined_allAAs_bicelle
 Scan: File "fims ... 149=0.822, #9575-9692"
 theor. Mass (M + H⁺): 1757.897
 Precursor Mass (M + H⁺): 1757.898
 Deviation 0.79 ppm
 m/z: 586.6393
 Charge: +3

Score: 61
 Peptide α [HYINIIIR] (E5)
 (>AGSINZ], from 26 to 34)
 Peptide β [WFGEEK] (E5)
 (>AGSINZ], from 207 to 213)
 Crosslinker: Photo-Leu-PTM

H File: D:\Skyline ... e-Y2R-NPY_XL_DDA_01_RD2_1_41@Score: 53
 Scan: File "fims ... 149=1.128, #12808-12830"
 theor. Mass (M + H⁺): 1741.887
 Precursor Mass (M + H⁺): 1741.888
 Deviation 0.49 ppm
 m/z: 871.4475
 Charge: +2

Peptide α [HYINIIIR] (E5)
 (>AGSINZ], from 26 to 34)
 Peptide β [GPIGAEAD] (E6)
 (>AGSINZ], from 2 to 10)
 Crosslinker: Photo-Leu-PTM

Figure S3: Selected fragment ion mass spectra of Y₂R-NPY cross-links. Precursor ions are shown in green, b- and y-type ions of cross-linked peptides are presented in blue and red, cleavage of the cross-linker is shown in yellow. A) VEQYGPQTTPR (Y₂R aa19–30)-YYSAIR (NPY aa 20–26), B) DPEPELID (Y₂R aa35–43)-YYSAIR (NPY aa20–26), C) VEQYGPQTTPR (Y₂R aa19–30)-YYSAIR (NPY aa20–26), D) DENQTVE (Y₂R aa9–16)-HYINIITR (NPY aa26–34), E) DENQTVE (Y₂R aa9–16)-YYSAIR (NPY aa20–26), F) WPGEEK (Y₂R aa207–213)-YYSAIR (NPY aa20–26), G) WPGEEK (Y₂R aa207–213)-HYINIITR (NPY aa26–34), H) GPIGAEAD (Y₂R aa2–10)-HYINIITR (NPY aa26–34).

What is more concerning is that almost identical cross-link sites were found from the full length receptor vs the NT peptide alone, which is counterintuitive as it suggests the major ligand-binding motifs (TMDs and ECL2) have no impact on NT interaction with the ligand. This would need some explanation. And the XL-MS experiment needs more controls (e.g. blocking NPY binding with an antagonist, or using a receptor mutant incapable of ligand binding) to validate the cross-link identifications.

As reflected by our structural models, the major interaction sites between the Y₂R NT and NPY do not overlap with the reported interactions of NPY in the TM area. Therefore, it is feasible that similar interaction patterns occur in samples containing the isolated Y₂R NT and the full length receptor, respectively.

Nonetheless, there might be some modulation by the TMD and ECL2 in the full length receptor that remain invisible in the XL-MS experiments. It is important to note that from the intensities of signals in the mass spectra it is impossible to quantify the cross-links as the ionization efficiencies between different cross-linked products are different. There might be cross-links that were not detected due to insufficient ionization, lack of elution in the LC separation or too small peptides. For these cases, we cannot exclude a spatial proximity that we were not able to detect in our experiments.

To complement the XL-MS data, we also performed NMR binding experiments between (isotopically labeled) NPY and the isolated Y₂R NT (former Fig. S1, now Fig. S4). This technique is more biased towards the highly mobile conformations as the thermal energy can drive molecules apart anytime, and the method has high sensitivity for short-range interactions. These experiments confirmed interactions between NPY and Y₂R NT that are strongest within the central helical portion of NPY. Compared to NMR experiments with full length Y₂R, the pattern of chemical shifts appeared different, indicating a modulation of NPY binding by the TMD. This is reflected most dramatically for R35 in NPY's C-terminus, which is tightly engaged in full length Y₂R, while its changes in chemical shift are much weaker when only the isolated Y₂R NT is present.

With regards to the technical controls: We performed additional XL experiments to carefully validate the specificity of the crosslinks by using excess unlabeled NPY and the specific antagonist JNJ-31020028 to block specific binding of the diazirine-labeled NPY. Under both conditions, cross-links of pL^{17,24,30}-NPY were essentially abolished, validating the specificity of our approach.

- Changes in the text:

p.6, l 131 ff:

... E9^{NT}, E10^{NT}, and Y22^{NT} were detected. To probe the specificity of the cross-links, we performed competition experiments using excess of either the Y₂R antagonist JNJ-31020028 (ref44) or unlabeled NPY. Both unlabeled ligands essentially abolished all cross-links and hence validate the specificity of the photo-reactions. The corresponding mass spectra are also available in the ProteomeXchange accession.

p.26, l.527 ff:

For competition experiments, the receptor was additionally incubated with either 50x molar excess of the antagonist JNJ-31020028 (Selleck Europe) or 50x molar excess unlabeled NPY.

For changes regarding the complementary NMR experiments, please refer to point 8 of our response below.

Furthermore, the deposited MS data inventory (shown in the attachment) is quite confusing and inconsistent with the manuscript. While it was mentioned in Results that this work labeled NPY with diazirine-containing photo-leucines, the raw data files are named with both pLeu and pMet, suggesting photo-Leu and photo-Met were both used? The authors acquire MS data in both DDA and DIA modes to but how these two different modes were applied to which samples or conditions and how DDA/DIA data were processed separately are completely missing in Methods. The data inventory should be clearly annotated with sample and acquisition conditions to allow re-use and re-analysis by others.

We apologize for the confusion and renamed the deposited MS data more appropriately. To label the NPY ligand, only photoLeucin was used. However, our precursor detection was based on building comprehensive spectral libraries. To this end, we also included cross-linking data using photo-methionine-labeled Y₂R.

With regards to the DDA and DIA modes: Using the spectral libraries created from DDA data enabled us to process the DIA-PASEF data by Skyline, resulting in a more sensitive detection of cross-linked peptides compared to a DDA workflow (Rojas Echeverri J C et al. *Analytical Chemistry* 96.19 (2024): 7373-7379; ref43). We clarified this now in the Methods section.

- Changes in the text:

p.28, l.579 ff.

Additionally, DDA-PASEF data were analyzed with Skyline⁷⁸ as described recently⁴³. Briefly, MeroX cross-linking results were converted into ProXL format. The ProXL and MGF files were used as input for creating spectral libraries with Skyline including retention times of the cross-linked products. In addition to XL-MS data from photo-leucine-labeled NPY, previously obtained cross-linking data using photo-methionine-labeled Y₂R were used as basis for building comprehensive spectral libraries. Using the spectral libraries as described allowed DIA-PASEF data to be processed by Skyline, resulting in a more sensitive detection of cross-linked peptides compared to a DDA workflow.

2. For the statement “We identified several clusters of interactions, in which the diazirine substituted leucine analogs in NPY at positions L24 and L30 interact with multiple sites in the Y2R NT”, it is not true to describe cross-links as interactions or contacts as they just indicate spacial proximity, not physical interactions.

The reviewer is absolutely right, we have corrected this to **clusters of proximity** throughout the manuscript.

3. The procedures for Rosetta modeling and MD simulation are too simplified. Figure S2 is meaningless with no details. Key steps for integrating cross-linking data into modeling should be described in Results, and a detailed workflow can be provided in SI. Other critical results about how many models generated, how to validate them, RMSD between models to the original structure, and internal variations of these models... all need to be described. For the MD part, how to select models as the starting point is also missing. The whole section of modeling and MD is difficult to follow and evaluate due to lack of essential details and clarity. Figure 3C is hard to read, it would be better to change the color code and show MD results in a format similar to high-quality publications.

As suggested by the reviewer, we extended our methods section to provide a more detailed description of the in silico part. Figure 3 has been updated with longer simulation time and a new color-code.

- Changes in the text:

p.29, l.609 ff.

Computational modeling

AlphaFold2⁴⁵ version 2.1 was used to generate an ensemble of Y₂R models with the disordered NT in contact with the NPY peptide bound to the receptor's orthosteric pocket. Since AlphaFold2 is biased toward predicting the inactive state of class A GPCRs, some models exhibited a hybrid conformation, in which the NT interacted with the bound NPY peptide while the transmembrane (TM) helices remained in an inactive-like state. To select a fully active model that also displayed a partially folded NPY-NT complex, we chose the structure that best balanced two criteria: (i) the lowest root-mean-square deviation (RMSD) to the fully active cryo-EM structure of the Y₂ receptor in complex with NPY and G_{i1} (PDB ID 7X9B) and (ii) the lowest average residue-residue distances for cross-linked residue pairs from mass spectrometry (MS) data. The selected model was further refined using RosettaCM⁸⁸, incorporating the cryo-EM structure as a template and applying cross-linking data as distance constraints. Specifically, the maximum allowed distance between C_β atom pairs without an energetic penalty was set to 13 Å. The final model was chosen based on the minimization of the REF2015 Rosetta score⁸⁹, which included logarithmic energetic penalties proportional to the number and magnitude of constraint violations.

Molecular dynamics Simulation

The final model of the computational modeling was used to probe the conformational dynamics of the complex by MD simulations. The stabilizing palmitoylation site C8.60 as well as E8.61, were not resolved in the cryo-EM template structure. Therefore, the residues C8.60 and E8.61 were added manually using PYMOL (version 2.5.2, Schrödinger LLC). Finally, during the setup using the CHARMM-GUI Membrane builder⁸⁵, C8.60 was palmitoylated using the CYSP patch option. The N- termini of the Y₂ receptor and NPY was capped with the patch NTER from the CHARMM force field. Furthermore, the Charmm-GUI patch CT1 was used for the C-terminal end (E8.61) of the Y₂R model. The patch CT2 was used for the C-Terminus of NPY.

Updated Figure 3:

indicates contact frequencies with higher opacity reflecting more frequent contacts. The acidic patches E15-E20 and D35-E39 of the NT are highlighted. **B**: Visualization of overall contact frequency between residues of NPY and Y₂R NT. Contact frequencies are shown by red hue intensity, with higher frequencies indicated by darker shades. Residues with high contact frequencies are labeled for clarity. **C**: Distance plots for selected residue pairs M17^{Y₂R}-R25^{NPY}, D35^{Y₂R}-R25^{NPY}, and L40^{Y₂R}-R19^{NPY}. Different colors represent different simulation runs (run 1: blue to run 10: green). Corresponding snapshots at 0.4 μs are shown below the plots using the same color scheme.

4. “a turn-structure in the structurally well-resolved membrane-proximal part of the NT” can be rephrased, as it is a modeling result, not a true structure.

This sentence refers to a local structure of the amino acids D42-S43-T44^{NT} in the membrane proximal Y₂R NT. This local structure is structurally resolved in both the inactive (PDB 7ddz) and active (PDB 7X9B, 7YON) high-resolution structures of Y₂R. Therefore, we can be very confident about this local structure. To make this clearer, we inserted a direct reference to the pdb identifiers:

- Changes in the text:

p. 12; L202 ff:

Furthermore, we mutated a stretch of three polar residues D42-S43-T44^{NT} to alanine, which appear to stabilize a turn-structure in the structurally well-resolved membrane-proximal part of the NT (PDB 7DDZ, 7X9B, 7YON, ref 25–27) and mediate contacts to the ECL3 (Fig. 4A).

5. For receptor expression measurement, fluorescence microscopy is not quantitative or accurate. The standard flow cytometry can be done to measure receptor surface expression. Typically deleting NT would impair receptor trafficking and cell surface expression, yet the authors did not see such as an effect on the NT deletion mutant, please give some explanation.

To investigate the function of the Y₂R NT in a state that is as native as possible, we chose to use Y₂R expression plasmids devoid of any N-terminal affinity tag. Unfortunately, according to our experience there are no antibodies available that enable specific detection of Y₂R in eukaryotic cells, making it impossible to use ELISA or FACS as a readout for cell surface expression of these constructs. Instead, we relied on the C-terminal YFP tag and measured its expression in a plate reader. To ensure that the receptors are correctly folded and exported to the plasma membrane, and hence, the eYFP signal corresponds to surface-receptors, we performed confocal fluorescence microscopy using identical exposure time and contrast settings to enable proper comparison between the constructs. To back up the expression measurements by the plate reader, we now also quantified cell surface fluorescence directly from the confocal images (40-50 cells from 4 independent experiments). The results are quite comparable and underline a somewhat weaker expression of the charge reversal mutant of acidic

patch 2 (D35-E37-E39K; see Figure R2 below), while the other constructs are similar to wild type Y₂R. Thus, we are confident that all receptor variants of this study are expressed at levels comparable to the wild type receptor that enable a quantitative comparison. We extended the Methods sections to explain both receptor quantification strategies. Modulation of receptor expression by N-terminal residues has been reported for many GPCRs. However, the Y₂R seems to be an unusual receptor in this respect, which is reflected in several aspects. Firstly, no endogenous N-terminal signal peptide has been identified, consequently mutations in Y₂R NT will not interfere with this sequence, and the receptor expression seems to be quite robust by itself. Secondly, Y₂R is apparently not glycosylated in its N-terminus, which could affect receptor expression or localization of N-terminal mutants. There is a consensus sequence for N-glycosylation at N11^{NT}, however, there is no evidence for N-glycosylation of the receptor at least in heterologous HEK293 culture. We performed Western Blot analyses with lysates of transfected HEK293 cells. For both the wild-type Y₂R and a N11A mutant bearing a FLAG tag at their C-termini, there was a single sharp band at ~50 kDa corresponding to the unglycosylated receptor. There was no band shift with addition of PNGaseF for either of this constructs (see Figure R3 below). In agreement with this, a N11Q receptor variant does not affect binding or signaling (Lindner D et al. Cell Signal. 2009;21(1):61-68). The same study also looked at N-terminal deletions and receptor chimeras with other NPY receptors and did not report reduced expression as long as the membrane-proximal stalk starting at D42 was present (“ΔN+8”). In summary, while it is rather unusual that N-terminal mutations do not affect Y₂R expression or localization, we are confident that this is an intrinsic property of the receptor and not an artefact of our study.

Figure R2: Quantification of receptor surface expression of Y₂R-eYFP variants based on confocal microscopy. Results show the mean fluorescence intensity in plasma membranes of 40-50 single

cells from 4 independent experiments. Statistical differences were assessed by one-way ANOVA followed by Dunnett's post-hoc test against WT Y₂R. * p<0.05, ** p< 0.01.

Figure R3: Western Blot analysis of potential N-glycosylation at N11 of Y₂R. Lysates of transiently transfected HEK293 cells (25 µg total protein) were analyzed with or without prior treatment with PNGase F. A sharp band at ~50 kDa corresponding to Y₂R-FLAG is detected in all samples and does not change with PNGase F treatment. Additional bands at 43 and 30 kDa are from added PNGase F enzyme.

- Changes in the text:

Methods, p.32, l. 676 ff:

Identical acquisition times and image processing were applied for wild type and receptor mutants. From the processed pictures, the plasma membrane fluorescence was quantified using automated object identification in Cell Profiler v4.2.8 (ref85), with Hoechst33342-stained nuclei as primary objects (allowed diameter 80x150 px), which allowed identification of cells (propagation method, adaptive threshold with minimum cross-entropy) and plasma membrane (cells shrunken by 8 px) as secondary and tertiary objects, respectively. The median fluorescence intensity of the plasma membrane of transfected cells was recorded. For every Y₂R variant, ≥ 40 cells from 3-4 independent experiments were quantified.

p.32, l.691 ff:

One day post transfection, the cells were re-seeded into white and black poly-D-lysine coated 96-WP at a density of 150,000 cells/well in technical triplicate using phenol-red-free medium. The next day, the cells in black plates were used for quantification of receptor expression, and the eYFP fluorescence was determined by direct excitation at 485 (20) nm and 544 (25) nm emission in a Tecan Spark reader (Tecan, Männedorf,

Switzerland).

6. Figure 4: K_d values for the low affinity state are missing. Also need SI tables to show K_d values and statistics from replicates. Here it is critical to assess how different mutants affect ligand binding kinetics (k_{on}, k_{off} rates) relative to wild-type.

We thank the reviewer for bringing up this issue. Indeed, deletion of the Y₂R NT decreases the low-affinity binding constant, while single mutations in the acidic patches do not (Figure 5). This is very clear from both the 95% CI, but also single replicate data, and underlines a functional contribution of the Y₂R NT. In addition, we measured k_{off} rates for the different Y₂R NT variants. As described in more detail above, deletion of the Y₂R, but also charge reversal in the acidic patches increases k_{off} by two-to three-fold, providing an explanation for decreased recruitment of arrestin-3. All of these data are now provided in a novel SI Table S3, and also shown in Figure 5.

- Changes in the text:

Table S3: Binding properties of Y₂R NT variants. All values are given as mean (95% CI). The K_D, high of D42A-S43A-T44A has a very wide 95% CI and is therefore shown in italics. Statistical significance in each signaling pathway (row) was tested in a 1-way-ANOVA with Dunnett's post-hoc test against wild-type Y₂R. * p<0.05, ** p<0.01, *** p<0.001. n.d. not determinable due to small measuring window.

	WT Niuc-Y ₂ R-eYFP	Δ2-41	D42A-S43A-T44A	E15-20K	D/E35-39K
BRET _{max}	0.0292 (0.0273 to 0.0311)	0.6520 **** (0.5907 to 0.7133)	0.0060 (0.0040 to 0.0080)	0.0307 (0.0243 to 0.0371)	0.0270 (0.025 to 0.0287)
Fraction high affinity	0.0617 (0.0352 to 0.0881)	0.0412 (0.0357 to 0.0467)	0.1507 ** (0.0649 to 0.2365)	0.0783 (0.0705 to 0.0862)	0.0957 (0 to 0.1938)
Log K _{D, high}	-9.57 (-10.25 to -8.90)	-9.64 (-9.94 to -9.33)	-10.20 (-13.50 to -6.90)	-9.46 (-10.48 to -8.44)	-9.65 (-10.64 to -8.66)
[K _{D, high} replicates]	-9.88, -9.40, -9.43	-9.51, -9.76, -9.64	-9.81, -11.68, -9.11	-9.24, -9.21, -9.93	-9.38, -9.46, -10.11
Log K _{D, low}	-6.54 (-6.59 to -6.49)	-6.18 *** (-6.24 to -6.12)	-6.12 *** (-6.47 to -5.77)	-6.56 (-6.65 to -6.47)	-6.62 (-6.80 to -6.44)
[K _{D, low} replicates]	-6.52, -6.56, -6.54	-6.16, -6.18, -6.21	-6.11, -6.26, -5.98	-6.58, -6.52, -6.58	-6.53, -6.67, -6.65
k _{off, fast} / s ⁻¹	0.052 (0.009 to 0.095)	0.149 * (0.027 to 0.272)	n.d.	0.123 (0.078 to 0.168)	0.105 (0.054 to 0.156)

Figure 5:

Figure 5: Ligand binding at Y_2R NT variants. A) Equilibrium binding between K18-Tamra-NPY and Nluc- Y_2R -eYFP variants. Charge reversal in the acidic patches has wild-type-like equilibrium binding properties, while D42A-S43A-T44A displays a significantly smaller BRET window and further reduced low-affinity binding compared to wild-type Y_2R . B) N-terminal deletion in Δ2-41 Y_2R increases overall BRET window as expected from the reduced distance, and the low-affinity state has about two-fold reduced affinity compared to wild-type Y_2R . Please note that Y-axis scaling is different between panels A and B. C) Ligand dissociation from the low-affinity state (bound to 300 nM K18-Tamra-NPY; re-binding blocked by 50 μM antagonist) is faster for Δ2-41 and acidic patch charge reversal mutants. Data are the mean ± SEM of n=3 (A, B) or n=4 (C) independent experiments, performed in technical duplicate or triplicate. Related to Fig. S9: Cellular localization and activity of the Nluc- Y_2R -eYFP construct used for NanoBRET binding.

7. For the beta arrestin 2 BRET assays, the Emax measurement heavily relies on receptor expression. So the expression levels should be first adjusted for all mutants to be comparable with the wild-type. Also need an SI table to summarize EC50 and Emax data from replicates.

With regard to the BRET assays: Our test conditions ensure a very high excess of acceptor fluorophore over donor luminophore (3970 ng plasmid DNA over 30 ng). Under this condition of donor saturation, the BRET ratio tolerates fluctuations in the acceptor amount at least up to 50% reduction, which is the observed expression range of the Y_2R N-terminal variants. We have validated this by titrating the amount of WT Y_2R -eYFP to 50% of the original expression (by reducing the DNA amount, validated by fluorescence measurements) and got the same BRET ratios (see Figure R4 below). Moreover, also the observed initial rate of arrestin-3 recruitment is not sensitive to these changes in the expression level of receptor nor to reduction in the expression of Nluc-arrestin3 (Figure R4 below and updated Figure 4 in the manuscript).

Thus, we are confident that the substantial changes in the maximal BRET ratio or the interaction kinetics of some of the Y_2R variants reflect reduced stabilization of the NPY- Y_2R -arrestin-3 complex and are not a result of different expression levels.

Figure R4: Variation in receptor expression does not significantly affect maximal BRET signal or EC₅₀ (left). Moreover, also the observed rate of arrestin3-recruitment is stable under these conditions.

Moreover, we have now collected all numerical data of signaling assays (G protein and arrestin-3) in a novel Table S2.

- Changes in the text:

Table S2: Signaling of Y₂R NT variants. All values are given as mean (95% CI). Statistical significance in each signaling pathway (row) was tested in a 1-way-ANOVA with Dunnett's post-hoc test against wild-type Y₂R. * p<0.05, ** p<0.01, *** p<0.001, **** p<0.0001.

		WT Y ₂ R-eYFP	Δ2-41	D42A-S43A-T44A	E15-20K	D/E35-39K	E15-20Q	D/E35-39N/Q
G _{i1}	%BRET _{max}	-23.0 (-23.7 to -22.2)	-22.4 (-23.6 to -21.2)	-21.7 (-22.9 to -20.5)	-23.3 (-24.5 to -22.1)	-22.3 (-23.2 to -21.4)	-21.6 (-22.8 to -20.5)	-21.6 (-22.7 to -20.5)
	logEC ₅₀	-9.35 (-9.42 to -9.28)	-8.82 **** (-8.94 to -8.70)	-8.264 **** (-8.39 to -8.14)	-9.38 (-9.50 to -9.27)	-9.04 ** (-9.14 to -8.94)	-9.54 (-9.67 to -9.27)	-9.37 (-9.49 to -9.26)
	k _{obs} at 1 nM NPY / s ⁻¹	0.0102 (0.0068 to 0.0137)	0.0230 (0.0085 to 0.0374)	0.0196 (0 to 0.0401)	0.0135 (0.0102 to 0.0168)	0.0110 (0.0055 to 0.0165)	0.0122 (0.0082 to 0.0162)	0.0107 (0.0066 to 0.0148)
G _{oA}	%BRET _{max}	-39.7 (-40.7 to -38.8)	-40.0 (-40.9 to -39.0)	-36.9 (-38.1 to -35.8)	-38.5 (-39.6 to -37.3)	-38.6 (-40.5 to -36.8)	-37.5 (-38.6 to -36.3)	-38.0 (-38.9 to -37.2)
	logEC ₅₀	-9.39 (-9.44 to -9.33)	-8.74 **** (-8.80 to -8.67)	-7.97 **** (-8.04 to -7.90)	-9.38 (-9.45 to -9.31)	-9.15 **** (-9.25 to -9.04)	-9.50 (-9.58 to -9.43)	-9.59 *** (-9.64 to -9.53)
G _{qi}	E _{max} / %	100 (96.3 to 103.6)	97.6 (90.3 to 104.8)	93.6 (84.5 to 102.6)	86.7 (83.4 to 90.0)	77.6 ** (70.0 to 85.1)	73.4 *** (66.4 to 80.3)	83.8 * (75.9 to 91.7)

	logEC ₅₀	-10.27 (-10.39 to -10.15)	-9.50 *** (-9.71 to -9.29)	-9.01 **** (-9.21 to -8.82)	-10.16 (-10.25 to -10.06)	-9.98 (-10.20 to -9.76)	-10.30 (-10.52 to -10.09)	-10.46 (-10.71 to -10.21)
Arr3	BRET _{max}	0.201 (0.194 to 0.209)	0.150 *** (0.143 to 0.156)	0.088 **** (0.074 to 0.103)	0.155 *** (0.143 to 0.168)	0.099 **** (0.091 to 0.108)	0.236 ** (0.222 to 0.250)	0.218 (0.199 to 0.237)
	logEC ₅₀	-7.26 (-7.34 to -7.19)	-6.54 **** (-6.62 to -6.45)	-6.01 **** (-6.27 to -5.74)	-7.35 (-7.50 to -7.20)	-7.23 (-7.40 to -7.05)	-7.37 (-7.48 to -7.25)	-7.42 (-7.59 to -7.24)
	k _{obs} at 100 nM NPY / s ⁻¹	0.0060 (0.0042 to 0.0078)	0.0106 * (0.0053 to 0.0159)	0.0085 at 1 μM NPY (0.0006 to 0.0164)	0.0120 ** (0.0087 to 0.0152)	0.0114 ** (0.0098 to 0.0130)	0.0089 (0.0064 to 0.0114)	0.0092 (0.0053 to 0.0131)

Figure 4:

Figure 4: Y₂R NT mutations differentially modulate G-protein activation and recruitment of arrestin-3 variants. *A*) Close-up view to membrane proximal Y₂R NT from cryo-EM (PDB 7X9B) highlighting interactions of D42-S43-T44^{NT}. *B*) Live cell fluorescence microscopy shows that Y₂R NT variants are transported to the plasma membrane like the wild-type. Y₂R-eYFP variants are shown in yellow, cell nuclei are stained by H33342 and shown in blue, scale bar equals 10 μm. All pictures were acquired with identical light exposure and picture processing. *C*) Quantification of cellular receptor expression is based on eYFP fluorescence in a plate reader; shown are the

means and individual values of five independent experiments relative to wild-type Y₂R. D) Activity of Y₂R variants towards G_{i1} proteins as measured by a direct BRET assay (G_{i1}-CASE) 10 min after ligand stimulation. All variants show full activation, D42A-S43A-T44A has the strongest shift in potency, while all others show mild or no effects. E) Kinetics of G_{i1} activation after stimulation with 1 nM NPY is overall similar for most variants except Δ2-41 and D42A-S43A-T44A, which have a tendency for faster apparent rate constants (k_{obs}). 95% CI of wild-type Y₂R k_{obs} is given as grey rectangle for comparison. F) Recruitment of arrestin-3 to Y₂R variants as measured by BRET 10 min after NPY stimulation. D42A-S43A-T44A and Δ2-41 variants reduce recruitment of arrestin-3 to the receptor. Neutral (in green) and charge-inverted (in blue) variants of the acidic patch in Y₂R NT lead to distinct behavior, with only charge reversal impairing arrestin-3 recruitment. G) Kinetic analysis of arrestin-3 recruitment after stimulation with 100 nM/ 1 μM NPY. Charge reversal in acidic patches 1 and 2 leads to faster initial recruitment, but also early signal decay. Bar plot shows quantification of the initial rate of arrestin-3 recruitment to receptor variants. Apparent rate constants (k_{obs}) are plotted in comparison to wild-type Y₂R-eYFP and its 95% confidence interval (grey rectangle). * $p < 0.05$, ** $p < 0.01$ in one-way ANOVA with Dunnett's post-hoc test. For color legend, please see panel E. Data in C–G are the mean ± SEM of $n \geq 3$ independent experiments each performed in technical triplicate

8. For the NMR experiment, no methods are provided, and data presentation is incomplete. In addition to the delta ppm plot, the original spectra with annotation should be provided as a convention. The authors mentioned in Fig. S1 that the overall binding pattern of NPY to Y₂R NT is different from the full-length Y₂R-bound conformation, which is in fact contradictory to the cross-linking MS result (the same set of cross-links identified for NT and full-length).

We added all original NMR spectra to the SI and provided the assignment. We had to correct a misassignment in the previous data, now the values are correct and introduced into a novel SI Figure 4. We also added a brief experimental section.

With regard to the discrepancy between NMR and cross-linking MS, we would like to emphasize that these techniques are complementary rather than contradictory. In the MS, once a cross link is formed, the two molecules stick together. Therefore, cross-linking MS may prefer the more “defined” conformations of the complex structure out of a much larger ensemble of structures. In this sense, the XL-MS can be biased towards the per-connected/structured conformations. In NMR, once an interaction between two sites is formed, thermal energy drives the molecules apart again, so the NMR results are rather biased towards the highly mobile and less structured conformations of the same ensemble. Hence, NMR is better able to pick up changes in the interaction pattern between NPY and Y₂R NT in isolation compared to the context of the full-length receptor. In the latter scenario, the C-terminal end of NPY inserts into the binding pocket of full Y₂R with various contacts with residues from helices 2, 3, 5 and 6 of the receptor. These important residues restrain the peptide and lead to tight binding. Interestingly, also in our previous NMR study using full-length Y₂R (Kaiser A, Müller P, Zellmann T, et al. *Angew Chem Int Ed Engl.* 2015;54(25):7446-7449), L¹⁷ of NPY displayed strong chemical shifts, which cannot be readily explained by the recent cryo-EM structure, as this residue lacks sufficient proximity to any structurally resolved part of the receptor. However, our current results show that L17^{NPY} (in synthetic NPY variants, mimicking the endogenous M17^{NPY})

is taking part in the interaction network with the Y₂R NT that was not taken into account previously.

In a scenario with the isolated Y₂R NT, the binding of NPY detected by NMR is much weaker as judged by the low chemical shifts. This is agreement with earlier studies by the Zerbe group that detected only weak binding of NPY to isolated N-termini of NPY receptors (Zou C et al. J Pept Sci. 2009;15(3):184-191). Thus, in absence of the stabilizing contacts by the TM bundle, it is not surprising that NMR shows a different interaction pattern compared to the full-length receptor.

- Changes in the text:

Page 7, l. 142 ff.:

Furthermore, we monitored binding of site-specifically ¹³C/¹⁵N-labeled NPY variants²⁸ to the Y₂R NT peptide by NMR spectroscopy (Fig. S4). NMR is complementary to the XL-MS technique as it is biased towards the very mobile conformations that can be driven apart by thermal energy anytime, and the method has high sensitivity for short-range interactions. Chemical shift changes are induced by conformational changes in the peptide complex upon binding which alter the magnetic environment of a given nucleus. Indeed, weak changes in the chemical shifts of the labeled amino acids E15^{NPY}, S22^{NPY}, and R35^{NPY} of NPY are seen upon binding to Y₂R NT, which confirm interactions between NPY and Y₂R NT that are strongest within the central helical portion of NPY.

p.25, l. 511 ff.:

~~NMR experiments with specifically ¹³C/¹⁵N-labeled NPY were conducted as previously described²⁸.~~

Solution NMR experiments

Solution NMR experiments of specifically ¹³C/¹⁵N labeled NPY in the absence and in the presence of the Y₂R N-terminal fragment were carried out on a Bruker Avance III 600 MHz NMR spectrometer using a standard 5 mm inverse triple resonance probe with z-gradient at a temperature of 27°C. NPY was dissolved at a concentration of 400 μM in buffer (5 mM DHPC, 50 mM NaP at pH 7). The N-terminal Y₂R peptide was added at a 1:1 molar ratio. Fast phase-sensitive gradient enhanced ¹H-¹⁵N HSQC experiments using WATERGATE (3-9-19) solvent suppression were carried out using 8 μs ¹H and 37 μs ¹⁵N π/2 pulses and 64 transients per t₁ increment. Chemical shift perturbations (CSP) were calculated according to

$$CSP(\Delta_{\blacksquare}^1H, \Delta_{\blacksquare}^{15}N) = \sqrt{(\Delta_{\blacksquare}^1H)^2 + (\Delta_{\blacksquare}^{15}N/5)^2}.$$

Figure S4:

(purple). E15^{NPY}, S22^{NPY} and R35^{NPY} show biggest changes in chemical shifts indicating sites of interaction. The interaction pattern of NPY with the N-terminal peptide of Y₂R is different from the full-length Y₂R-bound conformation (grey, confidence interval as dotted line), taken from ref²⁸. B) ¹H/¹⁵N correlation spectra of the three isotopically labeled NPY variants used for interaction studies with synthetic Y₂R NT peptide (black: peptide alone, red: bound to Y₂R NT).

9. In Discussion, “the NT kinetically limits ligand access and the on-rate is accelerated upon electrostatic perturbation” is not supported by data, need ligand kinetics measurement. Arrestin recruitment kinetics does not directly reflect ligand access to the binding pocket.

We conducted additional experiments to provide more evidence that the Y₂R NT modifies Y₂R signaling by alteration of kinetic properties of ligand binding. As explained above, measurement of k_{off} showed a two to three-fold accelerated ligand dissociation, which consistent with the observed reduction of arrestin-3 recruitment as its recruitment to the receptor is a rather slow, multi-step process. Since we did not detect significant in equilibrium binding for the E15–E20K and D/E35–E39K variants, we speculate that an increased k_{on} rate of the ligand at least partly compensates the increased off-rate, such that equilibrium binding is only minimally affected. This is supported by the observed increased apparent rates (k_{obs}) of arrestin-3 recruitment and G_{i1} activation (new data) for the Δ2-41 deletion, and partly for the charge reversal mutants. Unfortunately, the small measuring window in the binding experiments currently precludes reliable direct measurement of k_{on} (k_{obs}).

We have now rephrased this section, and emphasized that accerelerated k_{on} currently remains a speculation solely based on k_{obs} for G_{i1} activation and arrestin-3 recruitment.

- Changes in the text:

p.21, l.393 ff:

We speculate that an increased k_{on} rate of the ligand at least partly compensates the increased off-rate, such that equilibrium binding is only minimally affected. This is supported by the observed increased apparent rates (k_{obs}) of arrestin-3 recruitment and G_{i1} activation for the Δ2-41 deletion, and partly for the charge reversal mutants (Fig. 4E,G, Tab S2).

10. As structural models indicates a very high contact frequency between D42NT and K304ECL3, it would be informative to show the functional effect of mutating K304.

Indeed, not only our MD simulations but also the cryoEM structures of Y₂R (pdb 7X9B, 7YON) show a contact of the stretch of residues D42-S43-T44^{NT} in Y₂R NT to the stretch K301^{7.29} - E302^{7.30} – K304^{7.32} in the upper TM7/ECL3 (the side chain of Y303^{7.31} is facing into the binding pocket and is therefore not considered). As suggested by the reviewer,

we have mutated these three residues to alanine to reduce their potential for polar interactions via their side chains. K301^{7.29}A was wild-type-like with respect to plasma membrane expression, G_{i/o} protein activation and recruitment of arrestin-3. K304^{7.32}A was exported to the plasma membrane, but showed overall reduced expression by about ~50%. While the potency of this mutant to activate G_{i/o} proteins was like WT Y₂R, recruitment of arrestin-3 to Y₂R-K304^{7.32}A was decreased to 66% compared to WT Y₂R. In addition, E302^{7.30}A, which folded completely normally and was expressed at similar levels to WT Y₂R, displayed about 4-fold reduced potency to activate G_i proteins as well as 4-fold reduced potency to recruit arrestin3, with a BRET_{max} of 65% compared to WT Y₂R. Accordingly, E302^{7.30} and K304^{7.32} have overall similar effects on Y₂R as their counterparts in the membrane-proximal NT, supporting a functional role for this contact. We also tried to combine the three mutants K301^{7.29}A-E302^{7.30}A-K304^{7.32}A to fully inhibit side-chain mediated contacts to the membrane-proximal NT similar to the D42A-S43A-T44A variant in the NT, however, this triple mutant had severe folding deficits and was retained intracellularly, hence precluding further quantitative investigation of this variant. We have included this new data into the manuscript and a new SI Figure S10.

- Changes in the text:

p.22, l.427 ff:

We confirmed that these severe functional effects originate from contacts to ECL3/TM7 by mutagenesis of the interacting residues (Fig. S10, Tab. S4). The point mutation E302^{7.30}A reduced the potency to activate G_{i1} by three-fold and reduced recruitment of arrestin-3 to 65% with four-fold reduced potency, thus resembling the effects of the mutation in the N-terminal side. Similarly, K304^{7.32}A reduced arrestin-3 recruitment to 66%. While those single mutants remained relatively mild in their effects (similar to single exchanges in the membrane proximal NT, cf. SI Fig. 9C), introducing multiple alanine exchanges in this stretch in ECL3/TM7 severely decreased receptor expression (Fig. S10B,C), thus precluding complete inhibition of hydrogen bonding from this side.

New Figure S10:

Figure S10: Functional characterization of Y_2R ECL3 residues in contact with the membrane-proximal NT. A) Close-up view to interactions of ECL3 with membrane proximal Y_2R NT from the cryo-EM structure (PDB 7X9B) B) Live cell fluorescence microscopy of single and combination mutants in ECL3. While the single mutants largely show wild-type-like expression, the double mutant K301^{7.29}A-K304^{7.32}A has reduced plasma membrane expression, and the triple mutant K301^{7.29}A-E302^{7.30}A-K304^{7.32}A is retained intracellularly. Y_2R -eYFP variants are shown in yellow, cell nuclei are stained by H33342 and shown in blue, scale bar equals 10 μ m. All pictures were acquired with identical light exposure and picture processing. C) Quantification of receptors in the plasma membrane based on microscopy experiments as shown in B from $N \geq 40$ cells in three independent experiments, relative to wild-type Y_2R . Plasma membrane expression was not determinable (n.d.) in the triple variant. D) Activity of Y_2R variants towards G_{i1} proteins as measured by a direct BRET assay (G_{i1} -CASE) 10 min after ligand stimulation. E302^{7.30}A displays a three-fold shift in potency despite wild-type-like expression. Concentration-response of double mutant K301^{7.29}A-K304^{7.32}A and triple mutant K301^{7.29}A-E302^{7.30}A-K304^{7.32}A is likely affected by their very low membrane expression (cf. B and C). E) Recruitment of arrestin-3 to Y_2R variants as measured by BRET 10 min after NPY stimulation. E302^{7.30}A displays a four-fold shift in potency and 35% reduced BRET_{max} despite wild-type-like expression. Concentration-response of double mutant K301^{7.29}A-K304^{7.32}A and triple mutant K301^{7.29}A-E302^{7.30}A-K304^{7.32}A is likely affected by their very low membrane expression (cf. B and C). Data in C–E are the mean \pm SEM of $n \geq 3$ independent experiments each performed in technical triplicate.

11. As for the full-length Y_2R protein, its construct and expression condition should be briefly mentioned in Results. In this work, a cysteine-deficient Y_2R was recombinantly expressed in *E. coli* as inclusion bodies, which is not a conventional procedure for GPCR expression. Cysteine deficiency will remove disulfides important for receptor stabilization, expression in bacteria rather than in insect cells will likely deprive the receptor of important PTMs esp. in N terminus. Could the authors explain why not expressing the receptor in insect cells as described in previous structural work? Also SEC data and SDS-PAGE data need to be shown to confirm the protein purity and quality.

We have added a few sentences about the Y₂R construct and its expression in the *Results*. Also, the *Methods* section now contains more information on the respective procedures, and we present purification and functionality of the in vitro folded Y₂R in a novel SI Figure S2.

We agree that most work in the structural biology field is done on recombinant receptors expressed in insect cells. However, quite a few labs also have produced functional GPCRs in *E. coli*, for instance the Opella group (e.g., Park SH, et al. *Nature*. 2012;491(7426):779-783), Grisshammer group (e.g., Daniels DA, Sohal AK, Rees S, Grisshammer R. *Anal Biochem*. 2002;305(2):214-226), Gawrisch group (e.g., Yeliseev A, Gawrisch K. *Methods Enzymol*. 2017;593:387-403), Kiefer group (e.g., Park SH, et al. *J Am Chem Soc*. 2006;128(23):7402-7403), Baneres group (e.g., Damian M et al. *Nat Commun*. 2021;12(1):3938), Plückthun group (e.g., Zhang M et al. *Nat Struct Mol Biol*. 2021;28(3):258-267) and others. Although some of these groups also switched to insect cells, *E. coli* expression and refolding is still a valid method if done properly. The problem with insect cultures and NMR applications is the high cost that are necessary for isotopic labeling of GPCRs. Because of low expression yields, large volumes of costly media are required which can only be afforded by a small number of groups. On the other hand, we and other groups have invested a lot of effort to optimize refolding and functionality of the receptors.

PTMs are not generated in *E. coli* indeed, but for the Y receptors, at least the palmitoylation at the end of the cytoplasmic helix 8 does not seem to be functionally important as C-terminal deletion mutant (Δ 342) folds normally and activates G proteins like the wild type (Walther C et al. *J Biol Chem*. 2010;285(53):41578-41590). Along the same line, the Cys-deficient Y₂R (including the C342A mutation) folds and signals normally in transfected HEK293 cells (Witte K et al. et al. *Biol Chem*. 2013;394(8):1045-1056). Cys-deficient receptors are also expressed in insect cells for EPR applications to provide a simple and straightforward strategy for labeling with spin probes (see work by Hubbell, for instance *Cell*. 2019; 176: 468). Moreover, we have no indication for an N-glycosylation in the NT of Y₂R at N11 (see our response to point 5 and corresponding Figure R3 above).

- Changes in the text:

Add to main text, p.5, line 106 ff.:

...employed lipid-reconstituted Y₂R. To this end, a cysteine-deficient Y₂R, which only contained the two Cys residues that form the disulfide bridge (C¹²³ and Cys²⁰³), was recombinantly expressed in *E. coli* as inclusion bodies using high density fed-batch fermentation⁴⁰. Inclusion bodies were solubilized in SDS, folded in vitro, and reconstituted into DMPC/DHPC bicelles. Receptor functionality after refolding was confirmed using a fluorescence-based ligand binding assay (Figure S2). This Y₂R in vitro system has provided **important** molecular insights...

Add to main text, p. 25, line 498 ff.:

A cysteine-deficient Y2R⁷⁴ with a C-terminal 8x His-tag was recombinantly expressed in *E. coli* as inclusion bodies using a fed-batch fermentation process. Inclusion bodies were solubilized and purified in sodium dodecyl sulfate (SDS), and subsequently functionally reconstituted in isotropic DMPC/DHPC-c7 bicelles, as described before⁷⁶. In brief, purification of the denatured Y2 receptor was done by Metal Chelate Affinity Chromatography (IMAC) using a HisPrep™ FF 16/10 column (GE Healthcare). Purified Y2R in 15 mM SDS, 50 mM sodium phosphate (NaP) pH 8 was diluted to 0.5 mg/ml and dialyzed against a degassed buffer (0.5 mM SDS, 50 mM NaP, 1 mM EDTA, 1 mM reduced glutathione (GSH), and 0.5 mM oxidized glutathione (GSSG)) at pH 8.5 for 48 h for functional disulfide formation. Preformed lipid bicelles (DMPC/DHPC-c7 (Avanti Polar Lipids, Alabaster, USA) in a molar ratio of 1/4) were incubated with the Y2R at a molar ratio of 1/600/2400 receptor/DMPC/DHPC-c7, followed by three cycles of fast temperature changes from 42°C to 0°C.

New SI Fig. 2:

Figure S2: Characterization of lipid-reconstituted cysteine-deficient Y₂R (Y₂R-Δ6). A) SDS-PAGE of purified protein. B) Fluorescence polarization measurements at a fixed ligand concentration of 50 nM Tetramethylrhodamine-NPY (Tamra-NPY) confirms specific ligand binding of Y₂R-loaded bicelles.

12. For many critical experiments, only citing a reference in Methods is not enough. Specific details should be provided such as sample amount, concentration, buffer conditions, etc. How DDA and DIA data were processed with Bruker software and other packages (MeroX, proXL, and Skyline) should be described in sufficient details rather than citing a reference.

We now included a very detailed supplementary methods part to enable adaptation of our workflow by others.

- Change in the text:

SI, Supplementary Methods:

Cross-linking

Aliquots (100 μL) containing 1.37 μM bicelle-refolded Y_2R or N-terminal peptide of Y_2R (NT- Y_2R) solution in 50 mM aqueous 2-[4-(2-hydroxyethyl)-piperazine-1-yl]ethanesulfonic acid (HEPES), pH 7.2, and 1.5 mM 1,2- diheptanoyl-sn-glycero-3-phosphocholine (DHPC) were incubated for 1 hour on ice and darkness with a triply diazirine-substituted NPY variant (final concentration of 13.6 μM). Cross-linking was induced by UV-A irradiation with a LED lamp (ANUJ6186, Panasonic) for 30 s for a total of 30 joules. For each cross-linking system triplicate cross-linking reactions were prepared and one negative control (NC) where no NPY was added; 3 x NPY- Y_2R + 1 x Y_2R NC and 3 x NPY-NT- Y_2R + NT- Y_2R NC. All samples were prepared using Suspension-Trapping (S-Trap™; Protifi) assisted protein digestion. Each replicate was mixed with 100 μL solution of 10% (w/v) sodium dodecyl sulfate (SDS) in 100 mM aqueous tris(hydroxymethyl)aminomethane hydrochloride (Tris-HCl), pH 7.5. The samples were reduced by adding 8.7 μL of a Tris(2-carboxyethyl)phosphine hydrochloride (TCEP) stock solution (120 mM in deionized water (dH₂O), final concentration 5 mM) and incubating at 55 °C for 15 min. Reduced cysteines were alkylated by adding 8.7 μL of iodoacetamide (IAA) stock solution (500 mM in dH₂O, final concentration 20 mM) and allowing reaction to proceed for 10 min at room temperature (RT) in darkness. The solution was acidified with 21.7 μL of aqueous 27.5% (v/v) phosphoric acid (final concentration ~2.5%) and vortexed. Prior to loading into S-Trap columns, each replicate was mixed with 1435 μL of binding/washing solution (100 mM Tris-HCl, pH 7.5, in 90% (v/v) Methanol). Samples were loaded sequentially in 150 μL aliquots and centrifuged at 4,000 x g for 30 s until all sample was transferred to the S-Trap columns. Additional 150 μL of binding/washing solution were added and passed-through by centrifugation at 4,000 x g for 30 s three times. After the last washing step, the S-Trap columns were centrifuged again at 4,000 x g for 1 min. Then, each S-Trap column was transferred to a clean Eppendorf Protein LoBind 2 mL tubes. To the top of each S-Trap column, 1 μg of AspN dissolved in 20 μL of aqueous 50 mM ammonium bicarbonate (ABC) was added to reach a 100:1 (protein:enzyme) ratio. The samples were transferred to a wet chamber at 37 °C and allowed the digestion to proceed overnight. Second enzymatic digestion was performed by adding 4 μg of Trypsin dissolved in 20 μL of aqueous 50 mM ABC to reach a 25:1 (protein:enzyme) ratio. After second digestion proceeded for 14 h, 40 μL of 50 mM ABC solution was added to each sample and centrifuged at 4,000 x g for 1 min. Afterwards, 40 μL of aqueous 0.2% (v/v) formic acid were added and centrifuged at 4,000 x g for 1 min. Last elution step was performed by adding 40 μL of aqueous 50% (v/v) acetonitrile (ACN) and centrifugation at 4,000 x g for 1 min. The peptide solutions were dried under vacuum and stored at -20 °C.

LC-TIMS-ToF-MS/MS Data Collection

Dried peptides were reconstituted by first adding with 5 μ L of aqueous 30% (v/v) ACN with 0.05% (v/v) trifluoroacetic acid (TFA), vortexing and short-spinned. The mixture was diluted with 45 μ L of aqueous 0.05% TFA before injections to a final ACN concentration of 3% (v/v). For each analysis, 20% of the total sample was loaded on column corresponding to an estimated 20 μ g of digest (based on initial protein content). Peptides were separated on an UltiMate 3000 RSLC nano-HPLC system (Thermo Fisher Scientific) that was coupled to a timsTOF Pro mass spectrometer (Bruker Daltonics). Peptides were trapped on a C18 column (precolumn Acclaim PepMap 100, 300 μ m \times 5 mm, 5 μ m, 100 \AA) (Thermo Fisher Scientific) and separated on a μ PAC 50 column (PharmaFluidics) or a self-packed Picofrit (New Objective) nanospray emitter (360 μ m ID \times 75 μ m ID \times 150 mm L, 15 μ m Tip ID) packed with a C18-stationary phase (3.0 μ m, 120 \AA) (Dr. Maisch GmbH). After sample load, the precolumn was washed for 15 minutes with aqueous 0.1% (v/v) TFA at a flow rate of 30 μ L/min and a pre-column temperature 50°C. Peptide elution and separation on the μ PAC column was performed with a linear 90 min water–acetonitrile (ACN) gradient from 3% to 50% B where A is aqueous 0.1% (v/v) formic acid and B is 0.1% (v/v) formic acid in ACN. Additionally, a flow gradient was employed ranging from 900 to 600 nL/min. The column was washed at a flow rate of 600 nL/min with the following gradient: 50% to 85% B ACN (5 min), 85% B (5 min), 85% B to 3% B ACN (5 min), 3% B (15 min). Only cross-linking samples generated from the bicelles embedded with Y₂R (Y₂R-bicelle) were analyzed with the μ PAC column. All subsequent samples were analyzed with the self-packed C18 column. On the self-packed C18 column, peptides were eluted and separated using a linear gradient from 3% to 50% B (with solvent B: 0.1% (v/v) formic acid in ACN) with a constant flow rate of 300 nL/min over 90 min, 50% to 85% B (1 min) and 85% B (9 min). The separation column is kept at 40°C using an external column heater (Sonation GmbH). To align results obtained from different chromatographic conditions, endogenous unmodified indexed retention time (iRT) peptides identified in all samples were used to create a retention time calculator to account for retention time shifts.

After chromatographic separation the peptides were ionized through electrospray ionization (ESI) with a capillary voltage of 1500 V and facilitated drying with N₂ gas at 180 °C and flow rate of 3.0 L/min. The generated ions were then analyzed by trapped ion mobility spectrometry (TIMS) in a dual trapped ion mobility cell setup prior to tandem mass spectrometry (MS/MS) detection. The TIMS-MS/MS data were acquired both in DDA-PASEF and DIA-PASEF mode. Ion accumulation and ramp time was performed in 200 ms. Two mobility-dependent collision energy ramps were used: 1) 59 eV at an inversed reduced mobility ($1/K_0$) of 1.6 V·s/cm² and 20 eV at 0.60 V·s/cm² and 2) 95 eV at 1.60 V·s/cm² and 20 eV at 0.60 V·s/cm². Collision energies were linearly interpolated between these two $1/K_0$ values and kept constant above or below the set values, respectively. For DDA-PASEF experiments, the target intensity per individual PASEF precursor was set to 100 000 with an intensity threshold of 1000. 10 PASEF MS/MS scans were triggered per acquisition cycle (2.47 s). Precursor ions in an m/z range between 100 and 1700 with charge states $\geq 2+$ and $\leq 8+$ were selected for fragmentation. Active exclusion was enabled

for 0.5 min (mass width 0.015 Th, $1/K_0$ width 0.100 V·s/cm²) with early re-targeting if the precursor intensity had a 4x improvement in precursor intensity. For DIA-PASEF, isolation schemes were adapted to the sample matrix based on cross-linked peptides identified in the DDA-PASEF data. The DIA-PASEF isolation scheme defined with timsControl (v 3.0.21) DIA-PASEF Window Editor function is provided in Table S6. After acquisition, DDA-PASEF data was processed with DataAnalysis (v5.3; Bruker Daltonik GmbH) to generate fragment ion spectra peak lists in mascot generic format text files (i.e., .mgf). Collected fragment ion spectra were combined if their precursor ions were collected within a 0.75 min window, the monoisotopic m/z of the targeted precursors was within 0.015 m/z , and the precursors $1/K_0$ were within 0.025 V·s·cm⁻².

Data Validation and Processing

MS identification of cross-linked peptides

Identification of cross-links was performed with MeroX (v. 2.0.1.7) (Götze et al., 2019) from the .mgf files generated with DataAnalysis. The annotation of Y₂R-NPY cross-links was performed with the following settings: semi-specific proteolytic cleavage: C-terminal at Lys and Arg (trypsin) and N-terminal at Asp and Glu (Asp-N) with up to 3 missed cleavages for each and a maximum of 5 total missed cleavages; peptide lengths of 4 to 25 amino acids; PTMs: alkylation of Cys by iodoacetamide (fixed), oxidation of Met; an artificial amino acid was defined in MeroX as "l" to consider (2S)-2-Amino-3-(3-methyl-3H-diazirin-3-yl)propanoic acid (photo-leucine; pLeu) with an elemental composition of C₆H₁₁NO and, for downstream analysis with proXL and Skyline, the cross-linker was defined as the mass shift with respect to Leucine (-CH₄); cross-linker specificity: site 1 was set to consider only NPY peptides containing "l" (i.e. pLeu) and site 2 was set to Asp, Glu, and the C-terminus for search set considering only acidic sites reactivity and site 2 was set to all 20 proteinogenic amino acids for search set considering reactivity to all amino acids; cross-link specific fragments (XL-fragments): evidence of two XL-fragments (Pep-N2: -CH₄ and Pep-N2+H₂O: +O - CH₂) at site 1 were considered as optional and at site 2 XL-fragments where the whole cross-linker was missing (i.e., Δ mass = 0) was considered as optional; search algorithm: Quadratic mode with a minimal peptide score of ≥ 0 ; a-, b-, and y-ion series were considered; precursor mass accuracy: 10 ppm; fragment ion mass accuracy: 10 ppm; 10% intensity as prescore cut-off; 1% false discovery rate (FDR) cut-off, and minimum score cut-off: 0.

MeroX results from each search set were combined and the combined results were exported as .csv files. Using an in-house written R script, a peptide and precursor-specific ion mobility library was extracted from grouped MeroX results. For each of the matched precursor ions, the mean $1/K_0$ and range were calculated from all the $1/K_0$ reported for all cross-link spectra matches (XSMs). These values were formatted into a Skyline ion mobility spectral library and exported as .csv files.

The same reformatted .mgf files used for XL-peptides identification were used for the identification of non-cross-linked peptides. Peak lists obtained from MS/MS spectra were identified using MS-GF+ version Release (v 2023.01.12) (Kim & Pevzner, 2014) and MyriMatch (v 2.2.140) (Tabb et al., 2007). The search was conducted using SearchGUI version (v 4.2.14) (Barsnes & Vaudel, 2018). Protein identification was conducted against a concatenated target/decoy protein database containing the mature version of porcine NPY (UniProt ID: P01304), the in-house produced cysteine deficient Y₂R, porcine trypsin (UniProt ID: P00761), flavastacin (UniProt ID: Q47899), and their respective reverse decoy sequences generated in SearchGUI. The identification settings were as follows: Trypsin, Semi-Specific, with a maximum of 2 missed cleavages 10.0 ppm as MS1 and 10.0 ppm as MS2 tolerances; fixed modifications: alkylation of Cys (+57.021464 Da), variable modifications: Oxidation of M (+15.994915 Da). Peptides and proteins were inferred from the spectrum identification results using PeptideShaker version 2.2.25 (Vaudel et al., 2015). Peptide Spectrum Matches (PSMs), peptides and proteins were validated at a 1.0% False Discovery Rate (FDR) estimated using the decoy hit distribution. All validation thresholds and algorithms specific settings are listed in the “SearchEngine parameters” and, together with the PeptideShaker project and .mzID results, can be obtained in https://panoramaweb.org/XL-MS_Y2R-NPY_photoLeu.url

Peptide ID import into Skyline and validation

Detailed analysis pipeline has been reported recently⁴³, but here we describe a summary of the validation procedure. MeroX results were converted with proXL. The .pep.xml output in combination with the .mgf files were used to generate XL-peptides spectral libraries in Skyline with a maximal 0.05 q-value cutoff. Spectral libraries for unmodified peptides were generated using PeptideShaker’s results using a Peptide confidence score of 0.99. Unmodified peptides from Y₂R and the enzymatic background that could be detected in all replicates were chosen as endogenous iRT standards. All peptides annotated to Y₂R and cross-linked peptides were imported into the Skyline document. The parsed ion mobility library was added to the document prior to generating extracted ion chromatograms (EICs) of the first three isotopes of each precursor ion from raw DDA-PASEF files; a TOF resolving power of 60,000 and ion mobility resolving power of 40 were used. Only XL-peptides that had passed the initial manual annotation in MeroX, were detected in all sample preparation replicates, and did not show similar peptide chromatographic features in the negative control samples were kept as true XL-peptides. These peptides, with their curated retention time integration windows, were then indexed into a retention time calculator within Skyline using endogenous iRT standards as reference points. The iRTs of these XL-peptides, the ion mobility library generated from DDA-PASEF data, and the top-10 fragment ions (intensity-wise) of each precursor ion were then used to guide generation of EICs from raw DIA-PASEF data in Skyline.

13. XL-MS experimental replicate information is missing in Methods.

All experiments were conducted at least three times independently.

- Changes in the text:

Legend to Figure 1:

Cross-linking experiments were conducted at least three times independently. Related to Table S1, listing all observed cross-links.

Reviewer #3 (Remarks to the Author):

The manuscript by Kaiser et al. focuses on the role of N-terminus (NT) of the NPY Y2 receptor (Y2R) in ligand binding and signaling. Previous study from Dr. Beck-Sickinger's group, who is a co-author on the present manuscript, indicated that deletion of the Y2R-NT decreases ligand binding, signaling and internalization (PMID: 18845246). Here, the authors extend these observations using cross-linking mass spectrometry and computational modeling to identify specific areas of Y2R-NT interactions with NPY, which is a novel aspect of the study. Based on the mutational studies, they also conclude that NT modulates arrestin-3 binding to Y2R, but it does not affect ligand binding or G-protein activation. These findings raise some important issues that need to be addressed:

1) The lack of the effect on ligand binding and G protein signaling seems to be in conflict with previous studies. This was not addressed in the discussion. In Figure 4A, NT deletion or D42A-S43A-T44A mutations lead to 6- and 18-fold decrease in G-protein activity (EC₅₀). Why is this called "a minor contribution"? Was any statistical analysis done to determine how consistent this effect was?

We thank the reviewer for allowing us to clarify this point. Our construct $\Delta 2-41$ is equivalent to "Y2 ($\Delta N+8$)" in the previous study (Lindner D et al. Cell Signal. 2009;21(1):61-68; PMID 18845246). We chose this deletion to be able to distinguish the role of the distal NT from the D42-S43-T44 patch.

In agreement with the mentioned previous study, the effect of the $\Delta 2-41/\Delta N+8$ variant on G protein signaling was rather mild (our study: 3-6-fold depending on the read-out; Lindner et al. 2-fold), and a similar shift showed in our quantitative arrestin3-recruitment assay, which was not yet available in 2009. The microscopy experiments in the study by Lindner et al. used quite high ligand concentrations of 1 μ M, which is still ~ 4 -fold above the EC₅₀ of this construct to recruit arrestin3. Thus, it is not surprising that the effects on arrestin3 recruitment (or receptor internalization) were not readily visible. However, using a ligand concentration of 100 nM, we can clearly see delayed receptor internalization for $\Delta 2-41/\Delta N+8$ compared to the wild-type receptor in live cell microscopy experiments (see also our response to point 3 below).

In contrast, our DST mutant functionally rather resembles the previous ΔN construct, which also lacks the polar anchor to TM7/ECL3. While Lindner et al. found 80-fold shifted EC₅₀ to activate G proteins, our construct displayed 18-fold shifted EC₅₀, which can be explained by still carrying the distal NT, even though it is displaced.

Thus, our data match the previous dataset well and provide further quantitative and structural insights into the function of Y2R NT. With respect to the wording, we changed the text to make clearer distinctions between the $\Delta 2-41$ and D42A-S43A-T44A variants, and indicate that the functional effects of $\Delta 2-41$ are small to moderate, while D42-S43-T44^{NT} is critical for function.

As suggested by the reviewer, we have now put together supplementary tables that include statistical analysis of all signaling assays. Indeed, the effects were highly reproducible and the differences in EC₅₀ and E_{max} are statistically highly significant.

- Changes in the text:

p.12, l. 213 ff:

In line with these results, deletion of the flexible part of Y_2R in the $\Delta 2-41$ variant also displayed a moderate three to four-fold decreased EC_{50} , while the D42A-S43A-T44A variant was twelve-fold less potent to activate G_{i1} -proteins. We measured very similar effects for the activation of the neuronal G_{oA} subtype in the same BRET-based setting, and in a classic second messenger assays using a chimeric G_{qi} protein that is very well established for NPY receptors^{26,47,48} (Tab. S2, Fig. S7). This suggests that the structurally well-resolved membrane proximal region around D42-S43-T44^{NT} is critical for high-potency activation of G proteins, while the more distal areas including the acidic patches shows smaller effects on G protein activation.

p.20, l. 357 ff:

Mutating the acidic patches around E15-E20^{NT} and D35-E39^{NT} to neutral asparagine or glutamine residues or reverting the negative charge to positively charged lysine stretches still allowed for high-affinity NPY binding and activation of the canonical G_i pathway. This is also reflected by the $\Delta 2-41$ variant, which lacks the flexible part of Y_2R NT including both acidic patches, and only displays a moderately three- to six-fold reduced potency in different $G_{i/o}$ -based readouts (Fig. 4D, Fig. S7 and Tab S2) in agreement with a previous study⁴⁷ using the same deletion (termed ' Y_2R ($\Delta N+8$)', 2-fold reduced EC_{50} compared to wild-type Y_2R).

p.22, l.412 ff:

In addition to the role of the dynamic interaction network in the disordered NT of Y_2R , our data also show an important role of the more ordered membrane-proximal part of the NT for the integrity of the Y_2R binding pocket. The polar stretch D42-S43-T44^{NT}, which was left intact in the $\Delta 2-41$ mutant, proved essential for receptor functionality, ...

Table S2: Signaling of Y_2R NT variants. All values are given as mean (95% CI). Statistical significance in each signaling pathway (row) was tested in a 1-way-ANOVA with Dunnett's post-hoc test against wild-type Y_2R . * $p < 0.05$, ** $p < 0.01$, *** $p < 0.001$, **** $p < 0.0001$.

		WT Y_2R - eYFP	$\Delta 2-41$	D42A-S43A- T44A	E15-20K	D/E35-39K	E15-20Q	D/E35- 39N/Q
G_{i1}	%BRET _{max}	-23.0 (-23.7 to -22.2)	-22.4 (-23.6 to -21.2)	-21.7 (-22.9 to -20.5)	-23.3 (-24.5 to -22.1)	-22.3 (-23.2 to -21.4)	-21.6 (-22.8 to -20.5)	-21.6 (-22.7 to -20.5)
	logEC ₅₀	-9.35 (-9.42 to -9.28)	-8.82 **** (-8.94 to -8.70)	-8.264 **** (-8.39 to -8.14)	-9.38 (-9.50 to -9.27)	-9.04 ** (-9.14 to -8.94)	-9.54 (-9.67 to -9.27)	-9.37 (-9.49 to -9.26)
	k _{obs} at 1 nM NPY / s ⁻¹	0.0102 (0.0068 to 0.0137)	0.0230 (0.0085 to 0.0374)	0.0196 (0 to 0.0401)	0.0135 (0.0102 to 0.0168)	0.0110 (0.0055 to 0.0165)	0.0122 (0.0082 to 0.0162)	0.0107 (0.0066 to 0.0148)
G_{oA}	%BRET _{max}	-39.7 (-40.7 to -38.8)	-40.0 (-40.9 to -39.0)	-36.9 (-38.1 to -35.8)	-38.5 (-39.6 to -37.3)	-38.6 (-40.5 to -36.8)	-37.5 (-38.6 to -36.3)	-38.0 (-38.9 to -37.2)
	logEC ₅₀	-9.39 (-9.44 to -9.33)	-8.74 **** (-8.80 to -8.67)	-7.97 **** (-8.04 to -7.90)	-9.38 (-9.45 to -9.31)	-9.15 **** (-9.25 to -9.04)	-9.50 (-9.58 to -9.43)	-9.59 *** (-9.64 to -9.53)
G_{qi}	E _{max} / %	100 (96.3 to 103.6)	97.6 (90.3 to 104.8)	93.6 (84.5 to 102.6)	86.7 (83.4 to 90.0)	77.6 ** (70.0 to 85.1)	73.4 *** (66.4 to 80.3)	83.8 * (75.9 to 91.7)

	logEC ₅₀	-10.27 (-10.39 to -10.15)	-9.50 *** (-9.71 to -9.29)	-9.01 **** (-9.21 to -8.82)	-10.16 (-10.25 to -10.06)	-9.98 (-10.20 to -9.76)	-10.30 (-10.52 to -10.09)	-10.46 (-10.71 to -10.21)
Arr3	BRET _{max}	0.201 (0.194 to 0.209)	0.150 *** (0.143 to 0.156)	0.088 **** (0.074 to 0.103)	0.155 *** (0.143 to 0.168)	0.099 **** (0.091 to 0.108)	0.236 ** (0.222 to 0.250)	0.218 (0.199 to 0.237)
	logEC ₅₀	-7.26 (-7.34 to -7.19)	-6.54 **** (-6.62 to -6.45)	-6.01 **** (-6.27 to -5.74)	-7.35 (-7.50 to -7.20)	-7.23 (-7.40 to -7.05)	-7.37 (-7.48 to -7.25)	-7.42 (-7.59 to -7.24)
	k _{obs} at 100 nM NPY / s ⁻¹	0.0060 (0.0042 to 0.0078)	0.0106 * (0.0053 to 0.0159)	0.0085 at 1 μM NPY (0.0006 to 0.0164)	0.0120 ** (0.0087 to 0.0152)	0.0114 ** (0.0098 to 0.0130)	0.0089 (0.0064 to 0.0114)	0.0092 (0.0053 to 0.0131)

2) The above conclusions were made based on the artificial system utilizing chimeric G_{qi}Δmyr to measure G protein signaling. Showing the effect on the endogenous NPY receptor signaling pathways (e.g. cAMP levels upon forskolin stimulation) would strengthen and validate the above data.

This point is well taken. Even though the G_{qi} chimera is a very well established technique in particular for the signaling of NPY receptors used in many publications, it might be prone to higher signal amplification due to the further downstream measurement especially when compared to the direct recruitment assay for arrestin-3. We therefore repeated the G protein assays using the novel BRET-based G-CASE sensors established in the Schulte Lab (Schihada H et al. Sci Signal. 2021;14(699):eabf1653). This allowed us to monitor the activation of the native G_{oA} and G_{i1} pathways while reducing the stimulation time to 15 min or even less. The results were highly similar to our previous results utilizing the G_{qi} chimera at 60 min stimulation time. All N-terminally modified Y₂R variants retained the ability to fully activate G_{i1} and G_{oA}, and the EC₅₀ shifts were comparable. For example, the EC₅₀ shifts for the DST variant ranged from 12-fold (G_{i1}) to 28-fold (G_o), which is fully consistent with the data for the G_{qi} chimera (18-fold shifted EC₅₀). As seen before, the charge modification in acidic patch 1 and 2 does not change G protein activation substantially.

Thus, the novel G-CASE data validate the mild to moderate effects of Y₂R N-terminal variants towards G protein signaling and further underline the differential behavior towards the recruitment of arrestin-3. The activation data for G_{i1} are now included in Figure 4, replacing the IP Assays using the G_{qi} chimera, which have now been moved to a novel Supplementary Figure S7, alongside with the activation data for G_{oA} measured by the G-CASE BRET sensor.

- Changes in the text:

p.12, l.208 ff:

We first investigated the signaling properties of Y₂R NT variants towards the canonical G_i-pathway (Fig. 4 D,E) and recruitment of arrestin-3 (Fig. 4 F,G). Wild-type Y₂R activated G_{i1} in a direct BRET-based readout with an EC₅₀ of 0.4 nM. This was hardly affected by mutations in the acidic clusters, with the exception of the charge reversal at D/E35-39K

(acidic patch 2), which displayed a very subtle, but statistically significant two-fold shifted EC_{50} value (Tab. S2). In line with these results, deletion of the flexible part of Y_2R in the $\Delta 2-41$ variant also displayed a moderate three to four-fold decreased EC_{50} , while the $D42A-S43A-T44A$ variant was twelve-fold less potent to activate G_{i1} -proteins. We measured very similar effects for the activation of the neuronal G_{oA} subtype in the same BRET-based setting, and in a classic second messenger assays using a chimeric G_{q1} protein that is very well established for NPY receptors^{26,47,48} (Tab. S2, Fig. S7).

Updated Figure 4:

Figure 4: Y_2R NT mutations differentially modulate G-protein activation and recruitment of arrestin-3. A) Close-up view to membrane proximal Y_2R NT from cryo-EM (PDB 7X9B) highlighting interactions of $D42-S43-T44^{NT}$. B) Live cell fluorescence microscopy shows that Y_2R NT variants are transported to the plasma membrane like the wild-type. Y_2R -eYFP variants are shown in yellow, cell nuclei are stained by H33342 and shown in blue, scale bar equals 10 μm . All pictures were acquired with identical light exposure and picture processing. C) Quantification of cellular receptor expression is based on eYFP fluorescence in a plate reader; shown are the means and individual values of five independent experiments relative to wild-type Y_2R . D) Activity of Y_2R variants towards G_{i1} proteins as measured by a direct BRET assay (G_{i1} -CASE) 10 min after ligand stimulation. All variants show full activation, $D42A-S43A-T44A$ has the strongest shift in potency, while all others show mild or no effects. E) Kinetics of G_{i1} activation after stimulation with

1 nM NPY is overall similar for most variants except $\Delta 2-41$ and D42A-S43A-T44A, which have a tendency for faster apparent rate constants (k_{obs}). 95% CI of wild-type Y_2R k_{obs} is given as grey rectangle for comparison. F) Recruitment of arrestin-3 to Y_2R variants as measured by BRET 10 min after NPY stimulation. D42A-S43A-T44A and $\Delta 2-41$ variants reduce recruitment of arrestin-3 to the receptor. Neutral (in green) and charge-inverted (in blue) variants of the acidic patch in Y_2R NT lead to distinct behavior, with only charge reversal impairing arrestin-3 recruitment. G) Kinetic analysis of arrestin-3 recruitment after stimulation with 100 nM/ 1 μ M NPY. Charge reversal in acidic patches 1 and 2 leads to faster initial recruitment, but also early signal decay. Bar plot shows quantification of the initial rate of arrestin-3 recruitment to receptor variants. Apparent rate constants (k_{obs}) are plotted in comparison to wild-type Y_2R -eYFP and its 95% confidence interval (grey rectangle). * $p < 0.05$, ** $p < 0.01$ in one-way ANOVA with Dunnett's post-hoc test. For color legend, please see panel E. Data in C–G are the mean \pm SEM of $n \geq 3$ independent experiments each performed in technical triplicate.

New SI Figure 7:

Figure S7: G protein activation of Y_2R NT variants measured in alternative experimental settings provide highly similar outcomes. A) G protein activation was re-routed to the phospholipase C pathway by co-transfection of a chimeric $G_{iq\Delta 4myr}$, and signal was read out by quantifying cellular inositol phosphate level after 60 min NPY stimulation. B) G protein activation measured by a G-CASE BRET sensor⁸⁷ after 15 min of NPY stimulation.

3) The conclusion about the effect of the NT deletion on arrestin-3 binding were made solely based on BRET. This data should be confirmed using additional methods, detecting interactions with endogenous arrestin-3. Do the mutations introduced to the NT affect the receptor internalization?

The adaptation of BRET techniques to quantify arrestin recruitment to GPCRs provides very high accuracy, such that effects can also be quantitatively compared with changes to the G protein activation. In this regard, the addition of diverse fusion proteins to the N-terminus of arrestin is well established and functionally very well tolerated (e.g., Zheng C, et al. In-Cell Arrestin-Receptor Interaction Assays. Curr Protoc. 2023;3(10):e890. doi:10.1002/cpz1.890). Moreover, we devise an experimental set-up which uses fusion of

the small Nanoluciferase to the NT of arrestin-3, which is then transfected as energy donor in very small amounts (10 ng plasmid DNA per 10 cm² cell area). We therefore do not expect any artifacts due to the tagging or significant overexpression of arrestin-3.

To complement the dataset, we have performed additional internalization experiments in HEK293 cells with endogenous expression of native arrestin-3 and transient overexpression of Y₂R-eYFP or its N-terminal variants. To be better able to distinguish the internalization of different mutants, we chose to stimulate the cells with 100 nM NPY, which is closer to the EC₅₀ of the Y₂R-arrestin3 interaction and should therefore be sensitive to changes. Differences of the mutants to recruit arrestin-3 in the BRET experiments also showed in the internalization setting. As shown Figure S8, wild-type Y₂R internalizes rapidly, with clear vesicular structures and reduction of cell surface receptors showing already after 10 min minutes. Ligand-induced receptor internalization was essentially complete after 30 min. As expected, Y₂R mutants with less potent and overall reduced arrestin-interactions in the BRET experiments also internalized more slowly with clear residual membrane localization after 10 min, in the rank order D42A-S43A-T44A > Δ2-41 > E15-20K = D35-39K. After 30-60 min stimulation with 100 nM NPY, all receptor variants were completely internalized, in agreement with the previous study by Lindner et al. (2009) that used 1 μM NPY. This is probably enabled by the turnover in the receptor-arrestin interaction. In addition, we would like to note that ligand-dependent internalization of the Y₂R is not exclusively driven by arrestin-interactions (Walther C et al. J Biol Chem. 2010;285(53):41578-41590.) and can also be conveyed by arrestin-independent mechanisms. It is well possible that this arrestin-independent component contributes to the observed near-complete receptor internalization of the N-terminal Y₂R variants with reduced arrestin-3 interaction.

- Changes in the text:

p.13, l.230 ff:

Arrestin-3 recruitment was further reduced to 44% in the D42A-S43A-T44A variant and required 18-fold higher NPY concentration, underlining a strong functional contribution of the membrane-proximal region. We validated the reduced recruitment of arrestin-3 by monitoring the internalization of all receptor variants upon NPY stimulation by live cell fluorescence microscopy (Fig. S8), which uses unmodified arrestin at endogenous expression levels. Ligand-induced internalization of Y₂R is generally arrestin-3 dependent, although also arrestin-independent pathways can contribute.⁴⁹ In agreement with the BRET results, the Δ2-41 and D42A-S43A-T44A variants, which show the weakest and lowest potency arrestin-3 recruitment, display delayed internalization.

Novel Figure S8:

Figure S8: NPY-induced internalization of Y₂R NT variants in live HEK293 cells. HEK293 cells were transiently transfected with Y₂R-eYFP variants (yellow) and imaged before and after stimulation with 100 nM NPY over 60 min. Nuclei were stained with H333342, scale bar equals 10 μm. While wild-type Y₂R internalized in endosomal structures already after 10 min ligand stimulation and the plasma membrane is hardly visible, Δ2-41 and D42A-S43A-T44A display delayed internalization in agreement with their weaker recruitment of arrestin-3.

4) Previous study (PMID: 18845246) suggested that the amino acid composition of the Y₂R-NT does not affect its function. Here, the authors identified two acidic patches within NT that are responsible for binding NPY. Yet, their mutation to neutral asparagine or glutamine residues had no effect on arrestin-3 recruitment. Doesn't this mean that these acidic amino acids are not really essential?

The Y₂R NT is an intrinsically disordered region, capable of many different context-dependent interactions. Accordingly, the interactions between NPY and the Y₂R NT are highly dynamic in nature and re-wire frequently. In contrast to structurally well-defined interactions, for example of the C-terminal hexapeptide of NPY with the Y₂R transmembrane area, mutagenesis of a single residue is therefore not expected to completely disrupt the interactions. Thus, it is not surprising that the interactions that form between, for example, acidic patch 2 and the NPY stretch Y21-H26 tolerate exchange of acidic residues to other polar amino acids, in particular since the observed interactions are not always salt bridges, but often rather hydrogen bonds or even van-der-Waals interactions. However, the overall sequence length and amino acid composition is essential for enabling the observed contacts, which is also nicely reflected in receptor chimeras from the mentioned previous study (Lindner D et al. Cell Signal. 2009;21(1):61-68; PMID 18845246). Therein, the authors created two chimeras based on Y₂R, which

either carried the N-terminus of the Y₁R or the Y₅R (in addition to the deletion mutants discussed above). Although these N-terminal chimeras were only tested for G protein activation, which seems to be generally less sensitive to N-terminal mutations, the Y₂R(Y₁R-NT) chimera was about 9-fold less potently activated by NPY, which is similar to the effect of the Y₂R_Δ2-41/ΔN+8 deletion variant (6-fold reduced potency in a comparable readout using chimeric Giq) and underlines a functional role for the receptor N-terminus. In contrast, a chimera carrying the NT of the Y₅R was functionally tolerated very well. This differential behavior of the chimeras correlates with their N-terminal sequences (see Figure R5 below). Similar to Y₂R, Y₅R contains two regions with enriched acidic amino acids, one membrane-proximal, a second located more distally. In contrast, the Y₁R NT is overall much shorter and lacks a second acidic patch in the membrane distal region. We therefore speculate that the receptor NT takes over subtype-specific functions. This is supported by the high homology of a given receptor subtype over evolution, even in distantly related non-tetrapod species, such as coelacanth or shark (holocephali).

Figure R5: Alignment of Y₁R, Y₂R and Y₅R NT. Even in distantly related non-tetrapod species, such as coelacanth or shark (holocephali) the NT retain high homology within a given receptor subtype. In Y₁R and Y₂R, the membrane-proximal residues that tether the NT to ECL3 are conserved (Y₁R: C33^{NT}; Y₂R: D42-T44^{NT}) as marked by triangles, for the Y₅R we currently lack a high-resolution structure. Y₂R and Y₅R are enriched in acidic amino acids, which are similarly spaced in a membrane-proximal and distal region, while the Y₁R NT only has a membrane-proximal acidic region (dashed red boxes).

5) Potential mechanisms by which changes in extracellular NT affect arrestin-3 binding without changes to the ligand binding should be discussed.

The differential behavior of Y₂R NT variants towards ligand binding and G protein activation on the one hand, and recruitment of arrestin-3 on the other hand is intimately linked to the different activation/recruitment mechanisms of G proteins and arrestins to the Y₂R.

Since Reviewer 1 raised a very similar concern (Majors, first section), we would like to refer to our response above.

6) The authors mention using different algorithms for identifying intrinsically disordered regions of the Y2R. Presenting the results of at least one such analysis would be helpful.

A number of web-server-based algorithms for the prediction of intrinsically disordered regions are publicly available, which rank the probability of disorder between 0 (highly ordered) and 1 (disordered). We have now introduced a new supplementary Figure S1 that shows the prediction of the commonly used IUPred2A algorithm, alongside with predictions of fIDPnn and ESpritz-D, the latter two of which have been ranked among the top prediction tools in the recent critical assessments of protein intrinsic disorder, CAID1 and CAID2 (Necci M et al. Nat Methods. 2021 May;18(5):472-481; Conte AD et al. Proteins. 2023 Dec;91(12):1925-1934). All three algorithms predict disorder over the entire stretch of the Y₂R NT.

- Changes in the text:

Page 5, l. 102f.: ... such as peptide recognition or transducer coupling. *This is consistent with bioinformatic predictions by IUPred2A³⁷, fIDPnn³⁸ or ESpritz³⁹ that consistently show a high probability of intrinsic disorder in the Y₂R NT (Figure S1)*

Figure S1: Prediction of Y₂R NT disorder by different algorithms. Residues with disorder propensity have a score near 1, while scores <0.5 are likely ordered. Predictions were made using the online servers of IUPred2A [ref³⁷] (<https://iupred2a.elte.hu/>), fIDPnn [ref³⁸] (<http://biomine.cs.vcu.edu/servers/fIDPnn/>) and Espritz-D [ref³⁹] (<http://old.protein.bio.unipd.it/espritz/>).

Altogether, the presented findings are novel. However, it is essential to perform functional validation (signaling, internalization) based on the endogenous signaling pathways. Moreover, better discussion and data interpretation in the context of previous studies is required.

We thank the reviewer for the kind appreciation of our work and hope that our additional experiments and amendments to the manuscript make our study more compelling.

NCOMMS-24-44402

Transient ligand contacts of the intrinsically disordered N-terminus of neuropeptide Y₂ receptor regulate arrestin-3 recruitment

Response to Reviewer's comments

We are grateful to the Reviewers for the insightful comments on our study. We have used the constructive feedback and amended our manuscript according to the suggestions made.

In the following, we are addressing the Reviewer's comments point-by-point. We quote statements from the Reviewer in **bold face**. We refer to changes made in the manuscript in *yellow italics*.

Reviewer #2 (Remarks to the Author):

The authors have made great efforts to address critical comments raised by the referee, which have been mostly resolved. Only two points remained for further clarification:

We thank the reviewer for the kind appreciation of our work.

(1) Since DIA-PASEF data acquisition is not widely adopted in XL-MS experiments and needs a complex data processing procedure, the criteria used for generating EICs from raw DIA-PASEF data in Skyline based on a reference library can be specified in Methods

We extended the paragraph "Identification of cross-linked peptides" in the Methods section and added the following sentences on XL-MS data analysis at the end of paragraph (lines 590 ff):

Additionally, DDA-PASEF data were analyzed with Skyline(ref78) as described recently. Specifically, LC-MS/MS data were processed and simplified into peak lists in Mascot generic format (.MGF), compatible with most database search engines. Subsequently, these files were used for identification of cross-linked products by MeroX. MeroX results, converted to Proxl XML files (<https://github.com/yeastrc/proxl-import-merox>), alongside with MGF files, were used to create spectral libraries of cross-linked products in Skyline. In addition to XL-MS data from photo-leucine-labeled NPY, previously obtained cross-linking data using photo-methionine-labeled Y2R were used as basis for building comprehensive spectral libraries. The compiled ion mobility library was integrated into the document for extracting ion chromatograms (EICs) of the first three isotopes of each precursor ion from raw DDA-PASEF files, using 10 ppm extraction windows. Using the spectral libraries as described allowed DIA-PASEF data to be processed by Skyline, resulting in a more sensitive detection of cross-linked peptides compared to a DDA workflow.

(2) the distance restraints for specific cross-linked residue pairs used in computational modeling of Y2R's N-terminal in contact with NPY can be also

provided as SI data.

As per suggestion of the reviewer, we introduced a new Table S5 listing all constraints used for computational modeling.

Table S5. Rosetta distance constraints between the Y₂ receptor (chain A) and the NPY peptide (chain B).

Constraint Type	Atom Type	Residue Number and chain ID (Receptor)	Atom Type	Residue Number and chain ID (NPY)	Constraint Function	Target Distance	Force Constant	Flatness
AtomPair	CB	20A	CB	24B	FLAT_HARMONIC	13	1	2
AtomPair	CB	39A	CB	24B	FLAT_HARMONIC	13	1	2
AtomPair	CB	16A	CB	30B	FLAT_HARMONIC	13	1	2
AtomPair	CB	20A	CB	30B	FLAT_HARMONIC	13	1	2
AtomPair	CB	37A	CB	30B	FLAT_HARMONIC	13	1	2
AtomPair	CB	39A	CB	30B	FLAT_HARMONIC	13	1	2
AtomPair	CB	211A	CB	30B	FLAT_HARMONIC	13	1	2

Reviewer #3 (Remarks to the Author):

Thank you for the thorough revision of the manuscript. My previous concerns have been adequately addressed. However, there are some minor points that still need to be addressed:

1) Fig. 2B – description of the light blue regions of the receptor is missing (ECLs?).

We apologize for the confusion. The color code in panel B is the same as in panel A, with orange color for the more distal residues and pink color for the membrane-proximal regions. We have updated Figure 2 and the figure legend accordingly.

Figure 2: Structural flexibility of the Y₂R NT in microsecond MD simulations. **A:** Violin plot of the backbone-RMSD values for each residue of the NT during the MD simulations, relative to the initial frame, grouped in sets of five. **B:** Example conformations sampled during MD simulations. Residue coloring matches the groups in the violin plot with a color gradient from orange to pink from distal to membrane-proximal regions and opacity increasing over simulation time.

2) Fig. 4E – are the differences in the times of G_i1 activation statistically significant?

The differences in the kinetics of G_i activation are only a trend, and not statistically significant. The p-values are 0.0620 for Δ2-41 and 0.2908 for the D42A-S43A-T44A variant. We added the specific p values in panel 4E and indicated that this is a trend only in the figure legend and in the main text (lines 223 ff.: *Interestingly, however, there was a trend towards faster G_{i1} activation in the Δ2-41 and D42A-S43A-T44A mutant (Fig. 4E), which might indicate faster ligand access.*)

Updated Fig. 4(E):

3) Fig. S8 – the differences in the Y2R internalization should be quantified.

As suggested by the reviewer, we quantified the receptor amount in the plasma membrane before and after 10 min stimulation with 100 nM NPY, and added this as new panel B to Fig. S10 (renumbered due to two new supplemental Figures for the MD analysis). In the later time points, the plasma membrane is not clearly identifiable for most of the variants, which precluded quantitative analysis. After 10 min stimulation with 100 nM NPY, 32% of wild type Y₂R still reside in the plasma membrane. In agreement with visual inspection and the BRET results, the D42A-S43A-T44A variant does not internalize at a NPY concentration of 100 nM and completely remains in the plasma membrane (106%). The deletion variant Δ2-41 shows an intermediate effect with 58 % of the receptors still being present in the plasma membrane. Similarly, mutations in acidic patch 1 and 2 slightly reduce receptor internalization (46-55% receptors in plasma membrane).

Updated Fig. S10(B) (formerly Fig. S8):

B Internalization after 10 min
100 nM NPY

Changes in the text:

I. 240 ff.

In agreement with the BRET results, the $\Delta 2$ -41 and D42A-S43A-T44A variants, which show the weakest and lowest potency arrestin-3 recruitment, display delayed internalization with 58% ($\Delta 2$ -41) and 106% (D42A-S43A-T44A) of receptors, respectively, still residing in the plasma membrane after 10 min stimulation with 100 nM NPY, compared to 32% for wild-type Y_2R (Fig. S10B).

4) Line 278 – grammatical error – please correct.

We rephrased this section to (I.282 ff.):

When only the acidic patches were mutated (E15–E20K and D35–E39K), **we did not detect** alterations in NPY binding at equilibrium. **There were neither changes in the high- or low-affinity state nor in BRET window compared to wild-type Y₂R, suggesting unchanged affinity at equilibrium and overall highly similar binding orientation (Fig. 5A, Tab. S3).**

Reviewer #4 (Remarks to the Author):

I was asked to review this in place of Reviewer 1. I can see that the authors have tried to address what reviewer 1 asked for with longer MD simulations. However, I think they missed the point that reviewer 1 was really asking about in terms of what the point of the MD simulations are actually for. The focus of the analysis and the conclusions are on the flexibility of the N-terminus, but this is fairly uninformative because this is already a) expected and b) implicitly present in the cross-linking (and NMR) data and the resultant constraint-derived models, as indeed succinctly encapsulated in Fig. S6.

What I was expecting to see from the extended MD simulations was an analysis of the contact times and nature of these. The authors refer to “contact rewiring” (an odd phrase) which presumably means that some contacts are not stable but its not clear which ones are stable? And do the experimentally derived cross-link distance fade over time or are they genuinely transient – ie. do they form/re-form within the 1 uS MD simulations.

First of all we like to thank the reviewer for their critical evaluation of our manuscript and their helpful suggestions that helped to broaden and further improve our MD analysis. As suggested by the reviewer we now added more explicit information on the dynamic nature of contacts. These data complement our previous analysis of residue contacts between Y₂R NT and NPY, which was summarized in Fig. 3. While Fig. 3C exemplarily shows the time-resolved development of some critical contacts, Fig. 3A summarizes the contact frequency in a flare plot that inherently depicts which contacts are stable and which are not, but does not show the time-resolved development of contacts.

To meet the requirements of the reviewer, we now added the time-resolved contact plot of all relevant pairs as an appendix to our response. Moreover, we added a new supplemental figure that show the twenty time-resolved contact plot with contacts > 28% of simulation time, demonstrating its dynamic nature (Fig. S7). Only four contacts with the structured receptor region remain formed during at least 50% of the entire simulation time: T44^{NT}-N50^{1.32} (83%), D42^{NT}-K304^{7.32} (71%), K45^{NT}-Q50^{1.32} (52%), S43^{NT}-V49^{1.31} (52%). All the other contacts are highly dynamic and interchange at high rates, with few exceptions where a contact persists throughout a single run. Independent of individual life times, contacts can still be highly frequent. This becomes obvious in the flare-plot, because of higher opacity of these contacts e.g. engaging acidic patch 1 and 2 (Fig. 3A).

The residues determined by cross-linking fluctuate around the threshold distance used for modeling the NT, fluctuate higher distances or seem to fade out (Fig. S8). In agreement with the MD analysis of native contacts of the corresponding region, the distance of L24 and L30 to the membrane-proximal receptor region is closest and mostly well defined (e.g. L24^{NPY}-E37^{Y2R}, L24^{NPY}-E39^{Y2R}, L30^{NPY}-E37^{Y2R}, L30^{NPY}-E39^{Y2R}). The remaining pairs sample larger distances. From the latter, the distances of some residue pairs remain close to the threshold (e.g. L30^{NPY}-D35^{Y2R}, L30^{NPY}-E31^{Y2R}). Others tend to fade out in most repeats and keep in close distance only in some simulations (e. g. L24^{NPY}-E15^{Y2R}, L24^{NPY}-E16^{Y2R}, L24^{NPY}-E20^{Y2R}, L30^{NPY}-E15^{Y2R}, L30^{NPY}-E16^{Y2R}, L30^{NPY}-E20^{Y2R}, L30^{NPY}-E22^{Y2R}). Finally, in other simulations, there is no obvious tendency to form contacts (L30^{NPY}-E7^{Y2R}, L30^{NPY}-D9^{Y2R}, L30^{NPY}-E10^{Y2R}). Accordingly, while some contacts fade out during the sampled simulation time, others are genuinely transient, with some of the contacts appear preserved.

In summary, in addition to the information already provided by our previous version on contact frequencies and life times, we now complemented this analysis by providing explicit data on all contact pairs with contact frequencies above ~30 %. We hope these additional data convince the reviewer that the focus of our analysis and conclusions is not solely on the flexibility of the N-terminus, but also provides informative insights into the nature of its contacts with the NPY.

New Figures S7 and S8:

Fig. S7: Time traces of the NT interface contacts. The plots show the minimal distance over time between non-hydrogen atoms of the residue pairs indicated in each graph. Residues were considered to be in contact if the distance was equal to or less than 4.5 Å (indicated by the black dotted line). For better visibility, the time traces were smoothed by averaging over ten frames per point. The original,

unsmoothed time trace is shown in the same color with higher transparency. Only residue pairs that were in contact for at least 28% of the overall simulation time are shown in this figure.

Figure S8:

ADistance to $L24^{NPY}$ 
B

Distance to L30^{NPY}

Fig. S8: C_{β} - C_{β} Distance plots between NPY residues L24 (A)/L30 (B) and Y2R NT residues identified by the crosslinking experiment. The plots show the minimal distance over time between the C_{β} atoms of the residue pairs indicated in each graph. Residues were considered to be able to crosslink if the C_{β} - C_{β} distance ≤ 13 Å (indicated by the black dotted line). For better visibility, the time traces were smoothed by averaging over ten frames per point. The original, unsmoothed time trace is shown in the same color with higher transparency.

Please see attached file "Appendix_MD-pdf" for a list of all relevant contacts.

Changes in the text:

I. 178 ff.

Residues from the proximal region are frequently in contact distance of 4.5 Å with residues from the 7TM-region, with T44^{NT}-N50^{1.32} (83%), D42^{NT}-K304^{7.32} (71%), K45^{NT}-Q50^{1.32} (52%), S43^{NT}-V49^{1.31} (52%) accounting for the most frequent contacts (Fig. S7). In agreement with the MD analysis of native contacts, L24 and L30 remain in putative cross-linking distance to the membrane-proximal receptor region (e.g. L24^{NPY}-E37^{Y2R}, L24^{NPY}-E39^{Y2R}, L30^{NPY}-E37^{Y2R}, L30^{NPY}-E39^{Y2R}) (Fig. S8).

Finally, its not clear to me how the interactions here lead to modulation of arrestin-3 effects? Most of the dramatic effects appear to come from the D42-S43-T44 region which is already ordered rather than from the remaining more N-terminal sections. I thus tend to think this connection to the transient nature of the interactions remains a bit speculative overall.

In the present manuscript, we have identified two functionally important regions in the Y₂R N-terminus. On the one hand, the ordered membrane proximal stretch D42-S43-T44 is important for the architecture of the peptide binding pocket and similarly affects G_i activation and recruitment of arrestin-3. The mutation in the membrane-proximal area alters the relative orientation of the entire Y₂R NT towards the peptide ligand as shown by the severely reduced netBRET in ligand binding experiments. This changed geometry will also affect the distal, disordered region, most likely reducing the efficiency of the peptide interactions. Thus, the functional effects measured for the D43A-S43A-T44A variant should be interpreted as the sum of direct effects from changes in the membrane proximal area plus impaired interactions of the disordered distal Y₂R NT.

In addition to this important structural constraint in the membrane proximal area, the more distal, disordered part of the N-terminus displays a multitude of short-lived interactions with several residues in the central helix of NPY. We interrogated the functional relevance of this region using the Δ2-41 variant, which leaves the membrane proximal D42-S43-T44 intact, therefore this mutant reports on the relevance of the disordered part of Y₂R NT only. Still, the functional effects in terms of arrestin-3 recruitment are highly significant. This was supported by a ~ three-fold reduction in ligand residence time of the Δ2-41 variant, providing a mechanistic link to the reduced recruitment of arrestin-3.

Changes in the text:

I.424 ff.:

In the D42A-S43A-T44A variant, these contacts are lost and the NT has overall increased distance to the TM binding pocket as reflected by a five-fold reduced window in nanoBRET-based ligand binding assays. **We note that modification of this region alters the relative orientation of the entire N-terminus and will also affect conformation and interactions of the disordered distal part. Therefore, the functional effects should be interpreted as the sum of direct effects from changes in the membrane proximal area plus impairment of the interactions of the disordered distal Y₂R NT.**

Some other minor points:

1. The authors should rationalize why L24_NPY to E16_NT was not seen in the complex.

For building the initial model of full length Y₂R with NPY as shown in Fig. S6, we used seven distance constraints as listed in the new SI Table 5. These included all measured cross-links of L30^{NPY} (E16^{NT}, E20^{NT}, E37^{NT}, E39^{NT}, E211^{NT}), and two cross-links of L24^{NPY} to E20^{NT} and E37^{NT}. Although we did not include a constraint between L24^{NPY} and E16^{NT}, the model still positions these regions in reasonable proximity, such that their interaction is picked up during the MD simulations, where we see contacts of M17^{NT} to L24^{NPY} and Y27^{NPY}. We would like to note here that the MD simulations show mixed types of interactions with polar and non-polar contributions, while the cross-linking reaction of diazirines (chemically) prefers acidic side chains. It is therefore not surprising that the contact points shift to neighboring residues as the bias towards acidic side chains is taken away.

2. On page 8, they say they “devised” the algorithms. I think the authors mean “employed”

We changed the wording to “employed” in this section.

I. 160:

We **employed** the AlphaFold2⁴⁵ and Rosetta⁴⁶ algorithms to create 3D-structural models of full-length Y₂R in the presence of NPY.

3. Table S1 is very difficult to interpret for non-experts. It could use some annotation.

We updated Table S1 and added a legend to every column.

Legend:

Column A: region in Y2R in which cross-link is detected

Column B: indicates if a fragment has been detected in the cross-linking experiment using full length Y2R folded into lipid bicelles

Column C: indicates if a fragment has been detected in the cross-linking experiment using a synthetic peptide covering the Y2R NT sequence

Column D: cross-linked peptide (sequence); @ shows exact cross-linked position e.g., @ 7 (position 7 of DENQTVE), 5 (position 5 of YYSApLR) one-letter amino acid code, pL= photo-leucine

Column E: cross-linked amino acid in Y2R

Column F: position of photo-leucine (pL) in NPY

Column G: score MeroX software. Please see methods for assignment and Götze, M (2019, Anal Chem)

Column H: mass-to-charge ratio of XL peptides

Column I: charge of XL peptides

Column J: mass of the found protonated molecule ion, in atomic mass units

Column K: calculated mass, in atomic mass units

Column L: retention time of cross-linked peptide in chromatogram

Column M: deviation between found and calculated mass in parts per million (ppm)

Column N: sequence of found fragment 1 in one letter code

Column P/Q: start/end position of found peptide fragment 1 within Y2R or NPY

Column R: sequence of found fragment 2 in one letter code

Column T/U: start/end position of found peptide fragment 2 within Y2R or NPY

Column V/W: cross-linked position within peptide fragment 1/2

Column X: automated file output by MeroX allows assignment of the spectrum (and file path) of the cross-linked peptide; Scan of the cross-linked peptide; name of the file; Scan number; precursor collision energy in eV, retention time in minutes; ion mobility

4. “Interestingly, within the Y2R NT, the two acidic patches E15–E20NT and D35–E39NT remain at a distance ≥ 7 Å most of the time and never approach closer than 5.3 Å to each other.” That two acidic patches don’t come close together is hardly surprising.

We rephrased this section to:

II. 183 ff:

“Interestingly, Within the Y₂R NT, the two acidic patches E15–E20^{NT} and D35–E39^{NT} remain at a distance ≥ 7 Å most of the time and never approach closer than 5.3 Å to each other. Accordingly, we speculate that the negatively charged patches from these acidic residues - due to their repulsive forces – might keep the ligand binding pocket of the Y₂R NT in an ‘open’ conformation.

NCOMMS-24-44402C

Transient ligand contacts of the intrinsically disordered N-terminus of neuropeptide Y₂ receptor regulate arrestin-3 recruitment

Response to Reviewer's comments

We are grateful to the Reviewers for the insightful comments on our study. We have used the constructive feedback and amended our manuscript according to the suggestions made.

In the following, we are addressing the Reviewer's comments point-by-point. We quote statements from the Reviewer in **bold face**. We refer to changes made in the manuscript in *yellow italics*.

REVIEWER COMMENTS:

Reviewer #2 (Remarks to the Author):

All previous comments have been well resolved by the authors, and the reviewer would recommend publication of the revised version.

Reviewer #3 (Remarks to the Author):

Thank you once again for making the changes in the manuscript. Here are some remaining comments:

1) Fig. 2B – description of the light blue regions of the receptor is missing (ECLs?).

Response of the authors:

We apologize for the confusion. The color code in panel B is the same as in panel A, with orange color for the more distal residues and pink color for the membrane-proximal regions. We have updated Figure 2 and the figure legend accordingly.

New comment:

I made a mistake in my previous review. I was referring to the light blue boxes in Figure 1B, not 2B. Please add the description along with other regions. My deepest apologies for the confusion.

In Figure 1B, the light blue color refers to the extracellular loops, while the darker blue represents the NT. We added this legend and updated Figure 1 accordingly:

Figure 1: XL-MS between NPY and Y_2R . A) Scheme of photoaffinity labeling. NPY containing photo-leucines at positions 17, 24, and 30 was cross-linked to bicelle-reconstituted Y_2R preparations by UV-A light. The cross-linked complex is enzymatically digested, peptides are analyzed by liquid chromatography trapped ion mobility spectrometry tandem mass spectrometry (LC-TIMS-MS/MS) using the MeroX software. B) Overview of XL-MS results. Each cross-link is represented by a green line. Inset: The majority of cross-links were found between L24^{NPY} and L30^{NPY} to Y_2R NT. Cross-linking experiments were conducted at least three times independently. Related to Table S1, listing all observed cross-links.

3) Fig. S8 (currently S10B)– the differences in the Y_2R internalization should be quantified.

Response of the authors:

Changes in the text:

In agreement with the BRET results, the $\Delta 2-41$ and D42A-S43A-T44A variants, which show the weakest and lowest potency arrestin-3 recruitment, display

delayed internalization with 58% ($\Delta 2-41$) and 106% (D42A-S43A-T44A) of receptors, respectively, still residing in the plasma membrane after 10 min stimulation with 100 nM NPY, compared to 32% for wild-type Y2R (Fig. S10B).

New comment:

Thank you for the quantification. However, the above changes in the text do not accurately describe the new results. While the D42A-S43A-T44A variant indeed shows significant changes in internalization, the delay in the internalization of the $\Delta 2-41$ variant is not statistically significant as compared to the control. Moreover, the internalization rate of the $\Delta 2-41$ form is not different from other mutants that did not show significant changes in arrestin recruitment. How do authors reconcile these discrepancies?

The reviewer is right, the reduction of receptor internalization of the $\Delta 2-41$ variant does not reach statistical significance in our experiments. We attribute this to two main factors. On the one hand, arrestin-dependent and arrestin-independent pathways similarly contribute, in quantitative terms, to the internalization of Y2R (Walther C et al., J Biol Chem. 2010;285(53):41578-41590). Thus, both types of internalization overlay in this of experiment and are therefore expected to weaken the effects of the arrestin-dependent pathway. In addition, the experimental deviations/errors are much larger in this type of analysis compared to BRET, thus precluding definite conclusions for small signal windows.

We have toned down the respective statements in the results section and added that arrestin-independent pathways limit accuracy of this type of analysis.

Changes in the text:

p. 14, l. 241 ff.

As a limiting factor, the internalization of Y₂R contains arrestin-3-dependent and –independent components,⁴⁹ which overlay in this type of analysis and are therefore expected to weaken the apparent effects of the arrestin-dependent pathway. Nonetheless, in agreement with the BRET results, the D42A-S43A-T44A variant, which shows the weakest and lowest potency in arrestin-3 recruitment, displayed severely delayed internalization with ~100% (86-133%) of receptors still residing in the plasma membrane after 10 min stimulation with 100 nM NPY, compared to 32% for wild-type Y₂R (Fig. S10B). Internalization of the $\Delta 2-41$ variant was also reduced with 58% (50-71%) receptors residing in the membrane after 10 min, although this remained a trend that did not reach statistical significance.

Reviewer #4 (Remarks to the Author):

The authors have addressed my concerns satisfactorily. Happy to publish.